# Pangea: A Fully Open Multilingual Multimodal LLM for 39 Languages

**Xiang Yue,**[*] **Yueqi Song,**[*] **Akari Asai,**
**Seungone Kim, Jean de Dieu Nyandwi, Simran Khanuja, Anjali Kantharuban,**
**Lintang Sutawika, Sathyanarayanan Ramamoorthy, Graham Neubig**
{xyue2,yueqis,gneubig}@cs.cmu.edu

Carnegie Mellon University

https://neulab.github.io/Pangea/

## Abstract

Despite recent advances in multimodal large language models (MLLMs), their development has predominantly focused on English- and western-centric datasets and tasks, leaving most of the world's languages and diverse cultural contexts underrepresented. This paper introduces Pangea, a multilingual multimodal LLM trained on PangeaIns, a diverse 6M instruction dataset spanning 39 languages. PangeaIns features: 1) high-quality English instructions, 2) carefully machine-translated instructions, and 3) culturally relevant multimodal tasks to ensure cross-cultural coverage. To rigorously assess models' capabilities, we introduce PangeaBench, a holistic evaluation suite encompassing 14 datasets covering 47 languages. Results show that Pangea significantly outperforms existing open-source models in multilingual settings and diverse cultural contexts. Ablation studies further reveal the importance of English data proportions, language popularity, and the number of multimodal training samples on overall performance. We fully open-source our data, code, and trained checkpoints, to facilitate the development of inclusive and robust multilingual MLLMs, promoting equity and accessibility across a broader linguistic and cultural spectrum.

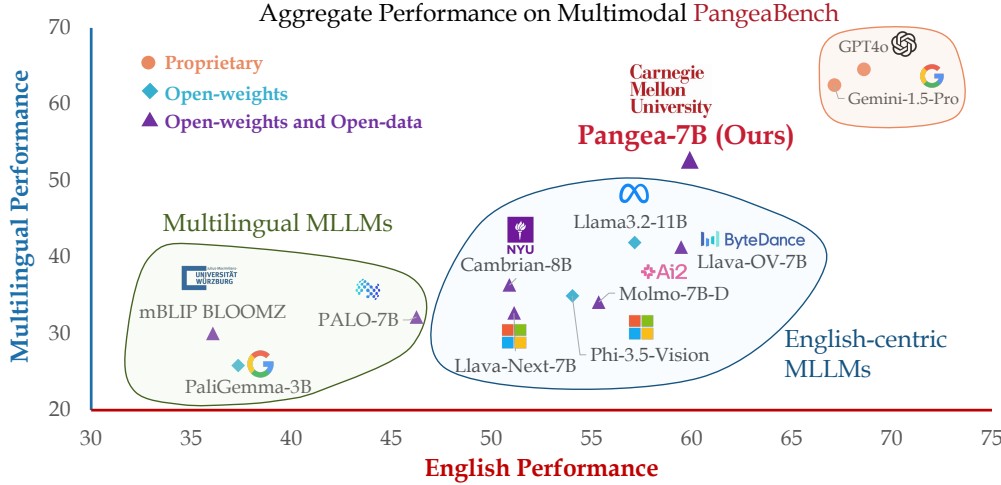

Figure 1: Overview of the aggregate performance of various multimodal LLMs on PangeaBench. Our Pangea-7B demonstrates comparable performance to SoTA open-source models in English settings, while significantly outperforming them in multilingual scenarios.

---

[*]Equal Contributions.

# 1 INTRODUCTION

Multimodal large language models (MLLMs) (Liu et al., 2023b; 2024; Dubey et al., 2024; Deitke et al., 2024; OpenAI, 2024b; Team et al., 2023) have demonstrated impressive capabilities in tasks such as image captioning, visual question answering, and complex reasoning (Yue et al., 2024a;b). Despite this rapid progress in their reasoning ability, a critical flaw persists: *the overwhelming focus on English- and western-centric training and evaluation datasets* (Liu et al., 2021; Song et al., 2023).

This homogeneity results in a lack of representation for the vast majority of the world's languages and diverse cultural contexts (Yu et al., 2022). Consequently, models predominately trained on such data exhibit: (a) diminished performance in multilingual settings (Blasi et al., 2022) with poor tokenization further leading to higher inference costs (Ahia et al., 2023); (b) generate outputs misaligned with the socio-cultural norms of underrepresented languages (AlKhamissi et al., 2024); and (c) lack the ability to recognize objects from geographically diverse regions (Ramaswamy et al., 2024) or rare objects belonging to the long-tail (Gupta et al., 2019). With the increased adoption of these models into real-world applications across the globe, there's an urgent need to develop multilingual MLLMs that equitably serve a diverse set of users. Few efforts have been made to develop multilingual MLLMs (Geigle et al., 2024b; Rasheed et al., 2025), however, their performance still exhibits inequalities across languages and lacks evaluation of cultural understanding.

In this paper, we address how to train and evaluate culturally inclusive multilingual MLLMs, using limited open-source resources, tackling four major challenges (Yu et al., 2022): **1) Data scarcity:** high-quality multilingual multimodal data is scarce, especially in low-resource languages, making it difficult to create large-scale training data; **2) Cultural nuances:** visual interpretations are context-dependent and vary across cultures (Ramaswamy et al., 2023; Khanuja et al., 2024); **3) Catastrophic forgetting:** training on many languages or modalities often results in suboptimal performance on some subsets and require careful balancing; **4) Evaluation complexity:** substantial resources and expertise are required to accurately measure performance across languages and cultures.

To tackle these challenges, we introduce PANGEA, an open-source multilingual MLLM designed to bridge linguistic and cultural gaps in visual understanding tasks. PANGEA is trained on PANGEAINS (Figure 2), a high-quality multilingual multimodal instruction tuning dataset comprising 6 million samples in 39 typologically diverse languages. PANGEAINS combines existing open-source resources with newly created instructions focused on multicultural understanding. We curate high-quality English instructions, carefully translate and adapt them for multilingual contexts. To address Western-centric biases in visual representations, we source images from LAION-Multi (Schuhmann et al., 2022), which includes images from various countries and captions in multiple languages. However, LAION-Multi contains images that are not culturally representative of the country's speaking population, and the associated alt text is often short, noisy, and lacks sufficient detail. To combat these issues, we develop a multicultural multilingual multimodal instruction generation pipeline, leveraging an LLM (Dubey et al., 2024) to score and filter images based on cultural informativeness. We then enhance the remaining data by generating detailed descriptions and creating complex instructions that combine culturally relevant tasks with general multilingual scenarios. This approach improves the model's cultural understanding while maintaining robust multilingual performance.

To evaluate PANGEA's capabilities, we present PANGEABENCH, a comprehensive multilingual and multimodal evaluation suite comprising five multimodal and three text-based tasks across 14 datasets in 47 languages. PANGEABENCH assesses MLLMs' performance on open-domain multimodal chat, image captioning, cultural understanding, multimodal reasoning, and text-only tasks including question answering and complex math reasoning. A key highlight of PANGEABENCH is the introduction of xChat, a human-crafted benchmark designed to evaluate open-ended, information-seeking multimodal conversations. xChat employs a fine-grained evaluation pipeline where human annotators annotate both reference answers and scoring rubrics for each query. An LLM then uses these rubrics to score the model's predictions on a 1-5 scale. This approach offers a more precise assessment of MLLM performance, addressing limitations of coarse LLM-as-Judge methods (Zheng et al., 2023). Additionally, we introduce xMMMU, a translated version of MMMU (Yue et al., 2024a), testing college-level multimodal reasoning across seven languages. Together, these components provide a robust, nuanced evaluation of PANGEA's cross-lingual and cross-cultural capabilities.

Our results demonstrate PANGEA's abilities in both English and multilingual scenarios, significantly outperforming existing open-source MLLMs on PANGEABENCH, surpassing the best open MLLMs

Figure 2: Statistics of PANGEAINS, comprising 6M multimodal instructions in 39 languages. The distribution of multilingual training data shows the percent of instances for each language among the *multilingual* instructions. PANGEAINS includes general instructions, document and chart question answering, captioning, domain-specific, culturally relevant, and text-only instructions.

by 0.4% on English tasks and 10.9% on multilingual tasks on average. Notably, PANGEA excels in multilingual and multicultural understanding, evidenced by its performance on xChat, CVQA, and MaRVL benchmarks. PANGEA also matches or outperforms state-of-the-art proprietary LLMs, namely Gemini-1.5-Pro and GPT4o, on several tasks such as XGQA. However, some performance gaps remain in multimodal chat and complex reasoning, shedding light on the need for further improvements in open MLLMs. We discuss key insights from trainingPANGEA, including the scaling effect of instructions, the role of English data, the impact of language-specific training proportions, and preliminary methods to improve multilingual OCR. We fully open-source PANGEAINS, PANGE-ABENCH, PANGEA-7B, and code, to advance culturally inclusive MLLMs across diverse languages.

## 2    PANGEAINS: MULTILINGUAL MULTIMODAL INSTRUCTION TUNING

Creating a truly multilingual, multicultural MLLM presents unique challenges. We developed PANGEAINS, a diverse and high-quality instruction tuning dataset. Comprising 6 million samples in 39 languages, PANGEAINS was curated with a focus on linguistic and cultural diversity. We implemented three key strategies to ensure comprehensive coverage, each addressing the specific hurdles encountered in multilingual multimodal learning. Figure 2 shows the distribution of PANGEAINS.

### 2.1    MACHINE TRANSLATED INSTRUCTIONS

To address the scarcity of human-annotated multilingual multimodal data, we primarily adopt machine translation as a practical and scalable solution to extend data beyond English. While human annotation is ideal, it is resource-intensive and impractical to cover a wide range of languages.

**Constructing a High-quality Pool of English Instructions from Existing Sources.** We first collect a high-quality set of English multimodal instructions, which serve as the foundation for translation into other languages. These instructions span a wide range of visual understanding tasks, including general visual instructions and conversations (Tong et al., 2024; Liu et al., 2024), visual reasoning, captioning, and chart question answering (Masry et al., 2022). Besides, we also added text-only high-quality English instructions, covering general instructions (Teknium, 2023), code (Zheng et al., 2024), and math (Li et al., 2024c). Figure 2 shows the statistics of our translated datasets. By leveraging existing English instructions, we ensured comprehensive coverage of visual interpretation and text instruction following tasks in English, preparing a pool of high-quality data for translation.

**Translation Model Selection.** To expand the English instructions to other languages, we initially experimented with strong open-source machine translation models, such as NLLB-3B (NLLB Team, 2024). However, we found that these models struggled with complex instruction-following scenar-

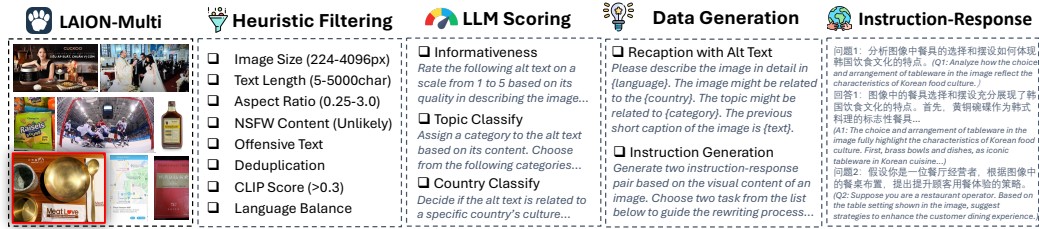

Figure 3: Overview of multicultural understanding instructions data generation pipeline.

ios and context-switching tasks, particularly in specialized domains like code generation and mathematical reasoning. For example, in code-related tasks, the model failed to recognize and correctly translate programming language keywords, significantly reducing the quality of the instructions. Based on these limitations, we shifted to using the proprietary Gemini 1.5 Pro model, which shows slightly better performance in small-scale human evaluations compared with GPT4o.

**Post-Processing Translated Data.** Even with high-quality translations, inconsistencies arose. To resolve issues such as mismatched conversation turns or missing candidates in multiple-choice questions, we developed a post-processing pipeline. This pipeline automatically corrected these errors or directly dropped the examples, ensuring that all translated instructions remained consistent.

Overall, Gemini 1.5 Pro's translation seems satisfactory, providing a fast, cost-effective alternative to human annotation, especially for scaling across languages. However, we acknowledge that machine translation still has limitations, particularly in handling nuanced contexts and cultural subtleties.

## 2.2 MULTICULTURAL UNDERSTANDING INSTRUCTIONS

While machine translation enables scaling across multiple languages, data translated from English is still Anglo-centric in coverage of cultural concepts (Yu et al., 2022). To address this, we developed a pipeline focused on creating instructions for multicultural understanding. Both visual and textual elements can convey deep cultural significance, and our goal is to design a dataset that allows models to not only recognize these nuances but also respond appropriately across various cultural contexts. The pipeline of creating multicultural understanding instructions is shown in Figure 3.

**Curation of Culturally Diverse Images.** To ensure that our dataset captures a wide array of cultural contexts, we began by sampling 10 million images from the LAION-Multi dataset (Schuhmann et al., 2022), which includes images and short alt texts from diverse languages and regions. A filtering process was proposed to guarantee both the quality and cultural relevance of the images.

- *Heuristic Filtering:* We implemented automatic filtering based on several key criteria: Image Size, Aspect Ratio, Text Length, NSFW content, Offensive Text, Deduplication, and CLIP Score (used to assess the alignment between the image and its textual description). This helped remove low-quality or inappropriate images and ensured the remaining dataset adhered to quality standards.

- *LLM Scoring:* To further refine the dataset, we employed the Llama-3.1-8B-Instruct model (Dubey et al., 2024) to evaluate the quality, subjects, and cultural relevance of the accompanying text descriptions (alt text) for each image. The model was instructed to perform the following tasks: 1) **Evaluate Text Quality:** The alt text was rated on a scale from 1 to 5 based on how well it described the corresponding image, assuming the model could not access the image itself. Alt text scoring below 4 was removed. 2) **Subject Classification:** The model assigned a subject or category to the alt text based on its content. 3) **Country/Region Classification:** The model determined whether the alt text was closely related to a specific country's culture. Images classified as "no specific country" (approximately 60% of the dataset) were excluded to ensure we focused on culturally identifiable content. The full LLM scoring prompt is included in Appendix B.

- *Avoiding Overrepresentation:* To maintain a balanced representation, we downsampled images from frequently occurring subjects, such as objects, materials, and clothing, to avoid skewing the dataset toward specific topics or regions. Then, we conducted an accessibility check, removing 30% of the remaining samples due to image download or other issues. Ultimately, we curated a final set of 1M high-quality, culturally specific images, forming the foundation of PANGEAINS.

**Captioning Images with Different Languages.** To provide context and enhance the model's ability to interpret images, we regenerated more detailed captions using Gemini 1.5 Pro based on high-quality alt texts. In this step, each image was accompanied by a caption written in the language corresponding to its cultural origin. However, our approach was not just about using a capable model. The alt text played a critical role in enriching the data, as it often contained culturally specific and contextually important information that would otherwise be absent from the images alone. For example, in Figure 8, with high-quality alt text, models can incorporate details such as *"President and CEO of The Walt Disney Company"* and *"a model of Shanghai Disneyland,"* adding significant context that may not be immediately evident from the image. This additional layer of information helps the model generate captions that better capture the cultural and contextual nuances.

**Generating Multilingual and Cross-Cultural Instructions.**

After recaptioning, we generated multilingual instructions based on the detailed captions with Gemini 1.5 Pro. Instead of only prompting the model to generate random instructions, we did a careful prompt engineering where we first came up with 13 task types (e.g., Information Seeking, Coding & Debugging, Critical Reasoning, Cultural Interpretation, etc.). Then for each image, up to two QA pairs were created, representing different instruction types to ensure a diverse set of interactions. This approach ensures that the model not only recognizes these visual elements but also responds appropriately across varied linguistic and different instruction contexts. The captioning and instruction generation prompts are included in Appendix B.

## 2.3 CURATING EXISTING MULTILINGUAL INSTRUCTIONS

To further enrich PANGEAINS, we conducted an extensive survey of available multilingual multimodal literature and datasets, including those hosted on HuggingFace. As a result, we incorporated several high-quality, open-source datasets into PANGEAINS. These include Chinese ALLaVA-4V (Chen et al., 2024), Viet Document and OCR QA (Doan et al., 2024), Llava Chinese (LinkSoul-AI, 2023), Llava Medical Chinese Instruction (BUAA, 2023), LLaVA-Japanese-Instruct (Toshi456, 2023), MTVQA (Tang et al., 2024), Japanese STAIR Captions (Yoshikawa et al., 2017), Russian GQA (Belopolskih & Spirin, 2024), French Doc-VQA (Sonagu & Sola, 2024), and French Table-VQA (Agonnoude & Delestre, 2024). Each of these datasets brings unique linguistic and cultural perspectives to the mix, covering a wide range of languages and task types.

## 2.4 DATASET STATISTICS

By combining these three methods, we created PANGEAINS, a comprehensive dataset addressing major challenges in building multilingual MLLMs: data scarcity, linguistic diversity, and cultural nuance. Its balanced language and task distribution supports the development of more sophisticated LLMs that can handle complex visual and textual content in a multilingual, multicultural context.

**Language and Task Distribution:** PANGEAINS features an extensive and balanced distribution of languages, tasks, and cultural contexts (as shown in Figure 2). We empirically keep the final language ratio of English to Multilingual as 40%:60% as we found a significant portion of English data plays an important role in cross-lingual transfer. See more discussions about the ratio in section 5 and Figure 6. The inclusion of diverse multimodal instructions ensures that the model develops a deeper understanding of varied linguistic and cultural environments. Examples of training samples from different languages and categories are provided in Appendix H. The comprehensive nature of PANGEAINS lays a solid foundation for training PANGEA, enabling it to become a truly multilingual, multicultural multimodal LLM, capable of understanding and interacting effectively with users from diverse linguistic and cultural backgrounds.

## 3 PANGEABENCH: EVALUATION OF MULTILINGUAL MULTIMODAL MODELS

## 3.1 OVERVIEW OF PANGEABENCH

To assess the capabilities of PANGEA across a variety of languages, cultures, and task types, we have developed PANGEABENCH, a comprehensive multilingual and multimodal evaluation suite. PANGEABENCH integrates diverse benchmarks that encompass both multimodal and text-only

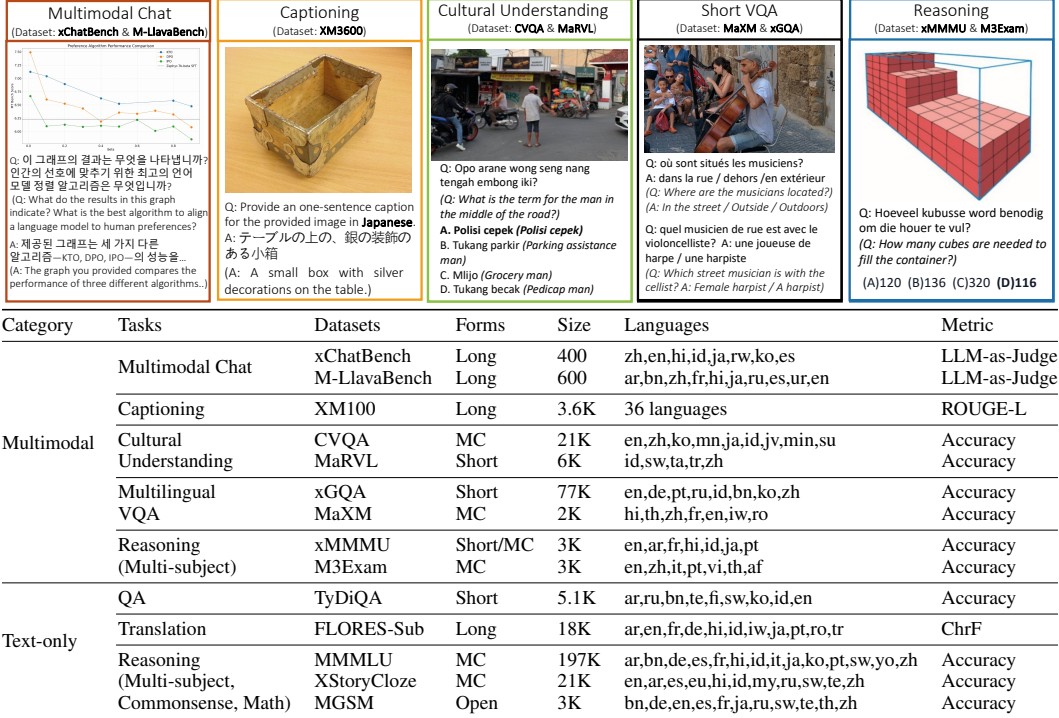

| Category | Tasks | Datasets | Forms | Size | Languages | Metric |
|---|---|---|---|---|---|---|
| Multimodal | Multimodal Chat | xChatBench | Long | 400 | zh,en,hi,id,ja,rw,ko,es | LLM-as-Judge |
| | | M-LlavaBench | Long | 600 | ar,bn,zh,fr,hi,ja,ru,es,ur,en | LLM-as-Judge |
| | Captioning | XM100 | Long | 3.6K | 36 languages | ROUGE-L |
| | Cultural Understanding | CVQA | MC | 21K | en,zh,ko,mn,ja,id,jv,min,su | Accuracy |
| | | MaRVL | Short | 6K | id,sw,ta,tr,zh | Accuracy |
| | Multilingual VQA | xGQA | Short | 77K | en,de,pt,ru,id,bn,ko,zh | Accuracy |
| | | MaXM | MC | 2K | hi,th,zh,fr,en,iw,ro | Accuracy |
| | Reasoning (Multi-subject) | xMMMU | Short/MC | 3K | en,ar,fr,hi,id,ja,pt | Accuracy |
| | | M3Exam | MC | 3K | en,zh,it,pt,vi,th,af | Accuracy |
| Text-only | QA | TyDiQA | Short | 5.1K | ar,ru,bn,te,fi,sw,ko,id,en | Accuracy |
| | Translation | FLORES-Sub | Long | 18K | ar,en,fr,de,hi,id,iw,ja,pt,ro,tr | ChrF |
| | Reasoning (Multi-subject, Commonsense, Math) | MMMLU | MC | 197K | ar,bn,de,es,fr,hi,id,it,ja,ko,pt,sw,yo,zh | Accuracy |
| | | XStoryCloze | MC | 21K | en,ar,es,eu,hi,id,my,ru,sw,te,zh | Accuracy |
| | | MGSM | Open | 3K | bn,de,en,es,fr,ja,ru,sw,te,th,zh | Accuracy |

Figure 4: Overview of PANGEABENCH, which contains 5 multimodal and 3 text tasks covering 14 datasets (including two newly curated xChatBench and xMMMU datasets). The table provides details about the datasets, while the figure shows evaluation examples from five different multimodal eval tasks in our PANGEABENCH.

tasks, enabling a holistic evaluation of PANGEA's performance in cross-lingual, cross-cultural, and multimodal contexts. Each task within PANGEABENCH is designed to probe specific aspects of PANGEA's proficiency, ensuring robust testing across a wide range of scenarios. All tasks in PANGEABENCH are evaluated under a zero-shot setting.

## 3.2 MULTIMODAL TASKS

The multimodal tasks in PANGEABENCH are categorized as follows: Multimodal Chat, Captioning, Cultural Understanding, Multilingual Visual Question Answering (VQA), and Multi-Subject Reasoning. We incorporate these in PANGEABENCH to ensure comprehensive testing of PANGEA's multimodal capabilities. The overview and examples of PANGEABENCH are shown in Figure 4.

**Multimodal Chat.** This task tests the model's ability to engage in natural and dynamic real-world conversations involving both text and images. Multilingual LlavaBench (Rasheed et al., 2025) (M-LlavaBench for short) stands as the only benchmark for evaluating multilingual long-form generation capabilities from MLLMs. Following the evaluation pipeline from Zheng et al. (2023) and Liu et al. (2023b), M-LlavaBench uses a coarse-grained evaluation criteria (e.g., "Please rate the helpfulness, relevance, accuracy, level of details of their responses."). Previous works suggest that employing such coarse-grained evaluation criteria may lead to automatic evaluation results that diverge from how humans would evaluate them (Ye et al., 2023; Kim et al., 2023; Lee et al., 2024; Kim et al., 2024a;b). To assess baselines with a more accurate evaluation pipeline with *fine-grained evaluation criteria* on diverse scenarios, we additionally annotate a new multilingual multimodal generation benchmark called the **xChatBench**, included in the multimodal chat category of PANGEABENCH. A more detailed explanation of the annotation process of xChatBench is included in Appendix E.

**Captioning.** The XM3600 (Thapliyal et al., 2022) dataset was developed to evaluate models' capability in multilingual image captioning. It contains images paired with captions in 36 different languages. However, it includes many similar images and captions. To address this, we clustered

the images based on captions and manually selected 100 representative images (denoted as XM100). This approach enhances the diversity of the samples and accelerates the evaluation process.

**Cultural Understanding.** To assess the model's ability to reason about and understand culturally diverse visual content, we use the CVQA (Romero et al., 2024) and MaRVL (Liu et al., 2021) datasets. These datasets are designed to test the model's performance in reasoning tasks involving culturally relevant imagery and concepts across multiple languages.

**Multilingual VQA.** This task measures the model's proficiency in answering questions about images across multiple languages. The xGQA (Pfeiffer et al., 2022b) and MaXM (Changpinyo et al., 2022) datasets provide a diverse range of visual question-answering challenges in several languages and scripts, addressing cross-lingual visual understanding.

**Multi-Subject Reasoning.** The xMMMU and M3Exam (Zhang et al., 2023) datasets are used to evaluate the model's reasoning abilities across different academic subjects. xMMMU is a machine-translated version of MMMU validation questions, which focuses on multimodal reasoning in multiple subjects. We randomly sample 300 questions from MMMU (Yue et al., 2024a) validation set and employ GPT-4o for the six languages translation. M3Exam challenges the model with real-world educational questions requiring both textual and visual comprehension. Details on how we ensure the translation quality, as well as detailed descriptions of other datasets, can be found in Appendix D.

## 3.3 TEXT-ONLY MULTILINGUAL DATASETS

While multimodal tasks are critical for evaluating the holistic capabilities of models like PANGEA, text-only multilingual tasks provide an equally essential dimension to assess. Most existing multimodal evaluations tend to overlook the importance of text-only evaluation, especially across diverse languages. Including text-only tasks in PANGEABENCH allows us to examine whether the model can perform well in scenarios that require deep linguistic understanding without the aid of visual context, highlighting its performance as a foundation model. We include three tasks QA, Translation, and Reasoning covering five datasets for the text-only evaluations in PANGEABENCH.

Specifically, we include TydiQA (Clark et al., 2020) to test the model's ability to answer questions across 11 typologically diverse languages. We adopt the FLORES (NLLB Team, 2024) dataset to assess machine translation performance. We sample 11 languages (denoted as FLORES-Sub). We use MMMLU (OpenAI, 2024a), a human-translated version of MMLU to test the general language understanding. We use XStoryCloze (Lin et al., 2021) and MGSM (Shi et al., 2022) to test the model's commonsense and mathematical reasoning ability in multilingual contexts respectively.

## 4 EXPERIMENTS

### 4.1 EXPERIMENTAL SETUP

We train PANGEA on PANGEAINS, our multilingual multimodal dataset comprising 6 million samples across 39 languages. The model uses LLaVA-Next as architecture (Liu et al., 2024), Qwen2-7B-Instruct (Yang et al., 2024) as the language model backbone and clip-vit-large-patch14-336 (Radford et al., 2021) as the vision encoder. The training consists of two stages. First, we pretrain the vision-language connector that aligns the outputs of vision encoder to backbone, with the LLaVA LCS-558K[1] (Liu et al., 2023b;a). Then, we perform finetuning on PANGEAINS, where

| Stages | Pretraining | Finetuning |
|---|---|---|
| **Training Data** | | |
| Dataset | LLaVA LCS-558K | PANGEAINS |
| #Samples | 558K | 6M |
| **Model** | | |
| Trainable | Projector (20M) | Full Model (8B) |
| **Training** | | |
| Batch Size | 128 | 128 |
| LR: $\psi_{\text{vision}}$ | $1 \times 10^{-3}$ | $2 \times 10^{-6}$ |
| LR: $\{\theta_{\text{proj}}, \phi_{\text{LLM}}\}$ | $1 \times 10^{-3}$ | $2 \times 10^{-5}$ |
| Epoch | 1 | 1 |
| GPU Hours (H100) | 32 | 1344 |

Table 1: PANGEA's training configurations.

we employ a learning rate of 2e-5, a batch size of 512, coupled with a cosine decay schedule with 0.03 warmup steps. We pretrain and finetune the model for 1 epoch, where pretraining took 4 hours with 8 H100 (32 GPU hours), and finetuning took 168 hours with 8 H100 (1344 GPU hours).

---

[1]https://huggingface.co/datasets/liuhaotian/LLaVA-Pretrain

| Models | AVG (all) | | Multimodal Chat | | | | Cultural Understanding | | | |
| --- | --- | --- | --- | --- | --- | --- | --- | --- | --- | --- |
| | | | xChatBench | | M-LlavaBench | | CVQA | | MaRVL | |
| | en | mul | en | mul | en | mul | en | mul | en | mul |
| Gemini-1.5-Pro | 67.1 | 62.5 | 67.0 | 54.4 | 103.4 | 106.6 | 75.9 | 75.7 | 76.4 | 72.0 |
| GPT4o | 68.6 | 64.6 | 71.0 | 64.4 | 104.6 | 100.4 | 79.1 | 79.4 | 81.4 | 82.1 |
| Llava-1.5-7B | 45.4 | 28.4 | 28.5 | 11.8 | 66.1 | 40.8 | 48.9 | 36.5 | 56.2 | 53.7 |
| Llava-Next-7B | 51.1 | 32.7 | 40.5 | 18.9 | 78.9 | 50.7 | 55.7 | 42.6 | 62.8 | 50.9 |
| Phi-3.5-Vision | 54.0 | 35.0 | 38.5 | 13.2 | 70.8 | 58.0 | 56.3 | 42.3 | 72.1 | 56.5 |
| Cambrian-8B | 50.9 | 36.4 | 27.5 | 11.3 | 78.4 | 61.8 | 59.7 | 47.5 | 75.4 | 61.8 |
| Llava-OV-7B | 59.5 | 41.3 | 51.0 | 28.5 | 89.7 | 55.3 | 65.2 | 53.7 | 72.7 | 57.5 |
| Molmo-7B-D | 55.4 | 34.1 | **49.5** | 21.1 | **95.9** | 13.8 | 59.4 | 48.3 | 65.3 | 54.9 |
| Llama3.2-11B | 57.2 | 41.9 | 49.0 | 27.8 | 93.9 | 58.2 | **70.2** | **61.4** | 64.5 | 58.1 |
| PaliGemma-3B | 37.3 | 25.8 | 6.0 | 3.5 | 32.1 | 31.9 | 52.9 | 42.9 | 56.5 | 52.2 |
| PALO-7B | 46.3 | 32.2 | 27.0 | 11.8 | 68.9 | 71.2 | 50.9 | 39.2 | 63.3 | 54.2 |
| mBLIP mT0-XL | 35.1 | 29.8 | 2.5 | 0.5 | 32.7 | 28.2 | 40.5 | 37.5 | 67.3 | 66.7 |
| mBLIP BLOOMZ | 36.1 | 30.0 | 4.0 | 1.6 | 43.5 | 41.0 | 44.9 | 36.9 | 62.3 | 58.6 |
| PANGEA-7B (Ours) | **59.9** | **52.8** | 46.0 | **35.8** | 84.2 | 89.5 | 64.4 | 57.2 | **87.0** | 79.0 |
| Δ over SoTA Open | +0.4 | +10.9 | -3.5 | +7.3 | -11.7 | +18.3 | -5.8 | -4.2 | +11.6 | +12.3 |

| Models | Captioning | | Short VQA | | | | Multi-subject Reasoning | | | |
| --- | --- | --- | --- | --- | --- | --- | --- | --- | --- | --- |
| | XM100 | | xGQA | | MaXM | | xMMMU | | M3Exam | |
| | en | mul | en | mul | en | mul | en | mul | en | mul |
| Gemini-1.5-Pro | 27.6 | 19.1 | 54.2 | 48.7 | 56.4 | 63.5 | 65.8 | 57.7 | 77.4 | 64.7 |
| GPT4o | 27.7 | 19.1 | 55.8 | 51.0 | 60.7 | 65.4 | 69.1 | 58.3 | 68.0 | 61.0 |
| Llava-1.5-7B | 28.6 | 1.1 | 62.0 | 30.6 | 49.8 | 20.4 | 36.2 | 31.5 | 32.3 | 29 |
| Llava-Next-7B | 29.3 | 9.4 | **64.8** | 37.8 | 54.9 | 21.4 | 36.7 | 34.3 | 36.5 | 28.4 |
| Phi-3.5-Vision | 30.2 | 5.2 | 64.7 | 38.4 | 55.3 | 25.0 | 42.6 | 38.8 | 55.8 | 37.2 |
| Cambrian-8B | 20.6 | 9.9 | 64.6 | 39.8 | **55.3** | 28.7 | 41.8 | 33.2 | 34.7 | 33.4 |
| Llava-OV-7B | **30.6** | 7.0 | 64.4 | 48.2 | 54.9 | 34.8 | 46.3 | 41.0 | 60.4 | **45.8** |
| Molmo-7B-D | 22.1 | 9.1 | 51.5 | 43.0 | 52.9 | 37.5 | 44.5 | 40.4 | 57.1 | 39.1 |
| Llama3.2-11B | 27.6 | 4.5 | 55.6 | 45.4 | **55.3** | 43.9 | **46.5** | 41.4 | 51.8 | 36.6 |
| PaliGemma-3B | 18.7 | 0.8 | 59.7 | 30.5 | 47.9 | 19.9 | 26.3 | 25.2 | 36.0 | 25.6 |
| PALO-7B | 30.4 | 0.8 | 60.5 | 37.8 | 51.4 | 16.3 | 33.1 | 30.5 | 30.8 | 27.8 |
| mBLIP mT0-XL | 31.9 | 3.1 | 44.2 | 39.9 | 44.7 | 36.8 | 29.3 | 30.4 | 22.8 | 25 |
| mBLIP BLOOMZ | 22.5 | 10.3 | 43.3 | 36.9 | 44.7 | 24.8 | 29.2 | 30.8 | 30.3 | 29.5 |
| PANGEA-7B (Ours) | 30.4 | **14.2** | 64.7 | **60.2** | 55.3 | **53.3** | 45.7 | **43.7** | **61.4** | 42.1 |
| Δ over Best Open Model | -0.2 | +3.9 | -0.1 | +12.0 | 0.0 | +9.4 | -0.8 | +2.3 | +1.0 | -3.7 |

Table 2: Overall performance on the multilingual multimodal benchmarks in PANGEABENCH. The best-performing open model on each dataset is in **bold** and the second best is underlined.

For evaluation, we compare PANGEA against several state-of-the-art open source baselines, including English-centric models Llava-1.5-7B (Liu et al., 2023a), Llava-Next-7B (Liu et al., 2024), Phi-3.5-Vision (Abdin et al., 2024), Cambrian-8B (Tong et al., 2024), Llava-OV-7B (Li et al., 2024b), Molmo-7B-D (Deitke et al., 2024) Llama3.2-11B (Dubey et al., 2024) and multilingual models PaliGemma-3B (Beyer et al., 2024), PALO-7B (Rasheed et al., 2025), mBLIP mT0-XL and mBLIP BLOOMZ (Geigle et al., 2024b). We also consider two text-only LLMs baselines Vicuna-1.5-7B (Zheng et al., 2023) and Qwen2-7B-Instruct (Yang et al., 2024), which are the backbones of Llava-Next and our PANGEA respectively. We integrate our multimodal tasks in PANGEABENCH into `lmms-eval` (Li et al., 2024a), a multimodal evaluation package that supports many English multimodal benchmarks. We use `lm-evaluation-harness` (Biderman et al., 2024) to evaluate text-only tasks. We follow the original paper for their best models' prompts in different tasks, and mostly reproduce their original numbers on datasets reported in the original papers.

## 4.2 MULTILINGUAL MULTIMODAL RESULTS

The results in Table 2 provide clear insights into the strengths and remaining challenges of PANGEA-7B in multilingual and multimodal tasks. Key observations from the evaluation include:

| Models | AVG (all) | | FLORES-Sub | | TyDiQA | | XStoryCloze | | MGSM | | MMMLU | |
|---|---|---|---|---|---|---|---|---|---|---|---|---|
| | en | mul | x→en | en→x | en | mul | en | mul | en | mul | en | mul |
| Vicuna-1.5-7B | 52.1 | 38.7 | 55.6 | 42.4 | 59.7 | 52.7 | 78.1 | 57.4 | 17.6 | 6.4 | 49.5 | 34.7 |
| Qwen2-7B-Instruct | 66.6 | **54.5** | **61.8** | **46.0** | 72.2 | **71.2** | **80.3** | **61.9** | 48.8 | 40.4 | **70.1** | **53.1** |
| Llava-1.5-7B | 53.1 | 39.0 | 54.7 | 41.5 | 66.8 | 52.8 | 79.1 | 57.6 | 14.8 | 7.6 | 50.2 | 35.7 |
| Llava-Next-7B | 54.0 | 38.9 | 54.8 | 41.4 | 68.3 | 52.1 | 79.1 | 57.1 | 15.6 | 7.5 | 52.1 | 36.5 |
| Phi-3.5-Vision | 60.7 | 41.7 | 28.5 | 32.5 | **75.9** | 51.3 | 77.9 | 54.8 | **59.2** | 33.1 | 62.0 | 36.7 |
| PALO-7B | 52.0 | 37.5 | 52.9 | 40.4 | 69.4 | 50.8 | 77.4 | 57.2 | 13.6 | 5.8 | 46.7 | 33.4 |
| PANGEA-7B (Ours) | **72.8** | 54.3 | 60.7 | 44.9 | 73.7 | 66.0 | 79.1 | 61.2 | **82.0** | **47.4** | 68.4 | 52.2 |

Table 3: Overall performance on text-only multilingual benchmarks in PANGEABENCH.

**Superior English and Multilingual Performance:** PANGEA-7B outperforms existing open-source models across both English and multilingual tasks. While concurrent multimodal models such as Molmo (Deitke et al., 2024) or Llama 3.2 show strong performance on English datasets, they struggle in multilingual evaluation settings. Particularly in multilingual subsets like xChatBench, M-LlavaBench, and MaRVL, it has achieved substantial gains, highlighting its effectiveness in both cross-lingual and cross-cultural contexts.

**Balanced Cross-Language Capabilities:** Unlike many models that exhibit a significant drop in performance when moving from English to multilingual tasks, PANGEA-7B is relatively consistent. For instance, in Multimodal Chat tasks, the performance gap between English and multilingual remains relatively small, indicating its ability to handle multiple languages effectively.

**Challenges Compared to Proprietary Models:** While PANGEA-7B leads in open-source models, some gaps remain when compared to closed-source models like GPT4o. Additionally, though PANGEA-7B narrows the gap between English and multilingual performance, there is still room for improvement in fully closing this divide across all tasks.

### 4.3 MULTILINGUAL TEXT-ONLY RESULTS

We further evaluate our model in text-only scenarios in Table 3. Interesting findings include:

**Best Text Performance Among Multimodal LLMs:** PANGEA-7B demonstrates the strongest performance among all multimodal LLMs in the text-only tasks consistently outperforming baselines like Llava-Next-7B. This highlights that, despite being trained as a multimodal model, PANGEA-7B maintains superior text understanding and reasoning capabilities compared to other MLLMs.

**Maintained Performance from its Text Backbone.** PANGEA-7B generally maintains or sees slight drops in performance on most text-only benchmarks compared with its text backbone Qwen2-7B-Instruct. Notably, the model shows a significant improvement in MGSM. This improvement is directly attributable to the inclusion of math-related instructions in PANGEAINS, which enhances the model's capability to handle complex multilingual reasoning and mathematical tasks.

## 5 DISCUSSION

Finally, we explore implications of our findings and their potential impact on future developments in the field. We examine the scaling effects of instruction quantity, the persistent role of English data, the relationship between training sample proportions and performance. Through this discussion, we aim to provide a comprehensive understanding of our model and chart a course for future advancements. More discussion on qualitative examples of model behavior on multilingual multimodal chat and challenges in multilingual OCR can be found in Appendix E and Appendix J.

**Scaling Effect of Number of Instructions.** Understanding how the quantity of instructions affects model performance is crucial for optimizing training strategies and resource allocation. Figure 5 reveals a clear scaling effect related to the number of instructions used during training. Performance improvements were consistent as we increased the number of multilingual instructions in PANGEAINS, for both English and multilingual performance. This demonstrates the necessity of scaling multilingual multimodal instruction tuning.

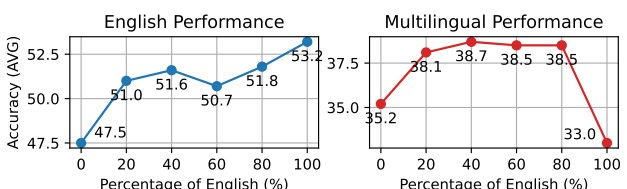

Figure 5: Scaling effect of training samples on English and multilingual scores across datasets.

**Role of English Data.** In multilingual scenarios, English data plays a pivotal role in cross-lingual transfer. To investigate this, we sampled 500K examples from the translated data described in subsection 2.1, ensuring a consistent data distribution. We varied the ratio of English data while keeping the total number of training samples fixed at 500K. For the 17 multilingual languages in the translated subset, we evenly distributed the number of samples across languages.

As shown in Figure 6 , English performance generally improves as the percent of English data increases. Surprisingly, using only multilingual data results in relatively lower multilingual performance. As we introduce more English data, multilingual performance improves, peaking at 38.7% with 40% English. However, performance drops sharply when English data reaches 100%. This suggests that English data aids cross-lingual transfer, however, over-reliance on it harms multilingual performance.

Figure 6: Impact of English training data proportion on English vs. multilingual performance.

**How does the proportion of training samples in a language affect downstream performance?** Is downstream task performance correlated with the number of training samples? Our analysis in Figure 7 revealed the relationship between training sample proportion and downstream performance. While there is a general positive correlation, the impact varies significantly across languages and tasks. For widely spoken languages with rich resources, we observed a near-linear relationship. However, for low-resource languages, even a small increase in proportion yielded disproportionately large performance gains. Interestingly, we also noted instances of positive transfer between typologically similar languages. These findings suggest that strategic allocation of training samples, considering both language prevalence and linguistic similarities, can optimize model performance.

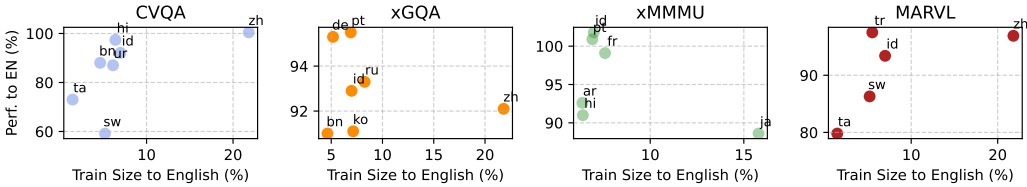

Figure 7: The relationship between training sample size (relative to English) and performance (relative to English) of different languages across four datasets.

# 6   CONCLUSION

In this paper, we introduced PANGEA, a multilingual MLLM designed to bridge linguistic and cultural gaps in visual understanding tasks. By leveraging PANGEAINS, our newly curated 6M multilingual multimodal instruction data samples, we demonstrated significant improvements in cross-lingual and cross-cultural understanding across 39 typologically diverse languages. Our comprehensive evaluation using PANGEABENCH revealed PANGEA's superior performance compared to existing open-source models. We also highlight ongoing challenges in areas such as low-resource language support and multilingual OCR. We fully open-source PANGEA, PANGEAINS, and PANGE-ABENCH to facilitate future research to build open and inclusive MLLMs.

## ACKNOWLEDGMENTS

This work was supported in part by a Carnegie Bosch Institute Fellowship to Xiang Yue, as well as grants from DSTA Singapore, and the Programs for Bridging the gap between R&D and the IDeal society (society 5.0) and Generating Economic and social value (BRIDGE)/Practical Global Research in the AI × Robotics Services, implemented by the Cabinet Office, Government of Japan. The training is supported by the CMU FLAME Center. The authors would like to thank Google Gemini credits for data construction and evaluation. The authors would also like to thank CMU NeuLab colleagues for their constructive comments.

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

# Table of Contents in Appendix

## A  RELATED WORK

**Visual Instruction Tuning.**  Visual instruction tuning is a key technique for enhancing multimodal large language models by aligning visual inputs with textual instructions to improve understanding and generation tasks (Liu et al., 2023b). Traditionally, these instructions are built using English-language data from visual question answering and other datasets (Liu et al., 2023b;a; Xu et al., 2023; Liu et al., 2024; Tong et al., 2024; Beyer et al., 2024; Zhu et al., 2024; Dai et al., 2023; Tong et al., 2024; Li et al., 2024d). Researchers often supplement this with synthetic instruction tuning data, generating large volumes of instructional pairs to possibly cover multiple languages too (Geigle et al., 2024b; Li et al., 2023; 2024d). However, these instruction-tuning datasets have mostly been task-focused and lack conversational capabilities. Further, while translation gives lends to multilingual capabilities, the data remains to be culturally homogeneous. By curating multilingual and multicultural instruction tuning data across various task types, our model is designed to intuitively understand and engage with users from diverse demographics.

| Dataset | # Languages | # of Instances | Multicultural | # of Task Types | Open-Sourced |
|---|---|---|---|---|---|
| MultiInstruct (Xu et al., 2023) | 1 | ~235.0K | ✗ | 310 | ✗ |
| MiniGPT4 (Zhu et al., 2024) | 1 | 5.0K | ✗ | 149 | ✓ |
| LLAVA (Liu et al., 2023b) | 1 | 1.2M | ✗ | >100K | ✓ |
| InstructBLIP (Dai et al., 2023) | 1 | ~1.6M | ✗ | >100K | ✗ |
| M$^3$IT (Li et al., 2023) | 80 | 2.4M | ✗ | 400 | ✓ |
| mBLIP (Geigle et al., 2024a) | 95 | 5.1M | ✗ | 68 | ✓ |
| PALO (Rasheed et al., 2025) | 10 | 2.1M | ✗ | 22 | ✓ |
| Cambrian (Tong et al., 2024) | 1 | 7.1M | ✗ | >1M | ✓ |
| PANGEAINS (Ours) | 39 | 6.2M | ✓ | >1M | ✓ |

Table 4: Comparison of datasets in terms of number of languages, number of instances, whether the dataset is multicultural, number of task types, and open-sourced.

**Multilingual Multimodal LLMs.**  Multilingual MLLMs have evolved from dual-encoder-based models, only capable of understanding and reasoning (Ni et al., 2021; Zeng et al., 2022; Jain et al., 2021), to encoder-decoder models capable of multilingual text generation as well (Shan et al., 2022; Chen et al., 2023; Geigle et al., 2024b). Despite their advancements, these models have remained focused on conventional tasks such as VQA and image captioning. Moreover, most efforts have centered around training with multilingual text, while little attention has been given to curating culturally diverse image datasets. Even for text, despite the focus on multilinguality, few attempts have been made to reflect cultural diversity in instructions and captions. As a result, these models tend to reflect a Western-centric bias. By selecting culturally diverse images from LAION and intentionally integrating this diversity into our instructions and captions, our model aims to serve a wide range of users in an inclusive and equitable manner.

## B PROMPTS USED IN THE DATA CONSTRUCTION

In this appendix, we will list the detailed prompts we used when constructing cultural understanding instruction tuning data described in subsection 2.2.

---

**Cultural Images LLM Scoring Prompt**

You are given an [Alt Text] associated with an image from the web.

**[Alt Text]**: {Alt Text}

Your goal is to:

1. **Evaluate Text Quality:** Rate the following alt text on a scale from 1 to 5 based on its quality in describing the image, assuming the model does not have access to the image:
   - 1 (Very Low Quality): Alt text is vague, irrelevant, misleading, or uses placeholders (e.g., file names).
   - 2 (Low Quality): Alt text is overly simplistic, generic, or provides minimal useful information.
   - 3 (Moderate Quality): Alt text is somewhat descriptive but lacks detail or relevance, with possible redundancy or ambiguity.
   - 4 (High Quality): Alt text is descriptive, clear, concise, and provides sufficient information to understand the image's content.
   - 5 (Very High Quality): Alt text is highly specific, detailed, and relevant, with a clear description that conveys all key aspects of the image.

2. **Subject Classification:** Assign a subject/category to the alt text based on its content. Choose from the following categories:
   - Vehicles and Transportation
   - Cooking and Food
   - People and Everyday Life
   - Sports and Recreation
   - Plants and Animals
   - Objects, Materials, and Clothing
   - Brands and Products
   - Geography, Buildings, and Landmarks
   - Tradition, Art, and History
   - Public Figure and Pop-Culture
   - Others

3. **Country/Region Classification:** Decide if the alt text is closely related to a specific country's culture. For example, if the alt text says, "Tokyo Skytree Photo in March with beautiful cherry blossoms", it's strongly related to Japan. If the alt text is not specifically about a certain culture or country, you can say "No specific country". Even if the alt text is written in their official language, it doesn't mean the caption is specifically about the country (e.g., a product page caption is often unlikely to be country-specific).

**Output:** Provide the final result in the following structured format:

1. **Text Quality Score (1-5):**

2. **Subject Category:**

3. **Country/Region:**

Only generate the final result without any additional descriptions or explanations.

---

---

**Image Recaption Prompts**

We randomly select one recaption prompt from the following:

**PROMPT 1:**
Please describe the image in detail in {language}. The image might be related to the country: "{country}". The topic might be related to: "{category}". The previous short caption of the image is {text}.

**PROMPT 2:**
Analyze this image and provide a comprehensive description in "{language}". Consider that it may be associated with "{country}" and the theme could be related to "{category}". If there is cultural significance, please include it. A brief previous description was: {text}.

**PROMPT 3:**
In "{language}", give a detailed description of what you see in this image. Keep in mind it might be connected to "{country}" and the subject could be about "{category}". If there are culturally relevant details, please include them. An earlier short description stated: {text}.

**PROMPT 4:**
Examine this image closely and describe its contents in "{language}" in a more structured way. The image might have a connection to "{country}" and could be about "{category}". A previous concise caption mentioned: {text}.

**PROMPT 5:**
Using "{language}", provide an in-depth and structured description of this image. It may be related to "{country}" and the topic could be associated with "{category}". A prior brief description was given as: {text}.

---

**Instruction Generation Prompt**

**Task**: Generate two **instruction-response pair** based on the visual content of an image. Choose two task types from the list below to guide the rewriting process:

- Coding & Debugging
- Information Seeking
- Creative Writing
- Critical Reasoning
- Planning & Strategy
- Mathematical Thinking
- Text Revision & Editing
- Data Analysis
- Role Playing & Scenarios
- Brainstorming & Ideation
- Advice Seeking & Problem-Solving
- Learning & Understanding
- Cultural Interpretation

**Guidelines**:

**Instruction**:

- Select two different task types from the list above.
- Make sure the instruction prompts an interpretation or analysis **directly tied to what can be visually observed in the image**, not just general reasoning.
- The instruction should require a response that **uses details from the image**. Avoid generic instructions that can be answered without visual information.

**Response**:

- Provide a **very detailed and structured** response that reflects a clear understanding of the implied visual information.
- Offer multiple perspectives, deep analysis, or step-by-step explanations where applicable.
- Avoid general responses that could be inferred without observing the image. Responses must rely on interpreting the visual content.

**Content**:

- Instructions should be varied, challenging, and explore different advanced aspects of the visual scene.
- Responses must showcase a deep understanding of the image's visual context, using thoughtful insights where applicable.

**Output**:

- Provide the output in JSON format with three keys: "task_type", "instruction" and "response".
- Ensure the instruction and response **do not mention "based on caption"** but instead, refer to the **image** or simply avoid reference to any external description.
- Do not include additional text or explanations beyond what is required.
- Provide both the "instruction" and "response" in {language} but "task_type" in English.

Caption: {caption}

## C RECAPTIONING EXAMPLE FROM LAION-CULTURAL

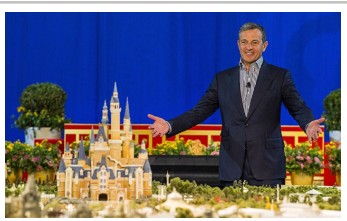

**Original Alt Text:**

*Президент и CEO компании The Walt Disney у макета Шанхайского Диснейлэнда*

*(President and CEO of The Walt Disney Company by a model of Shanghai Disneyland)*

**Caption with High-quality Alt Text:**
На фотографии изображён президент и CEO компании The Walt Disney, стоящий перед макетом Шанхайского Диснейленда. На заднем плане видны красочные цветочные композиции и зелёные кусты, придающие сцене праздничную атмосферу. На переднем плане находится детализированный макет с замком, украшенным множеством башен и шпилей. Президент компании улыбается и разводит руки в стороны, как бы представляя миниатюрный парк перед ним. На нём надет тёмный костюм и полосатая рубашка, что подчёркивает его официальный статус. Фон сцены оформлен синим цветом, что придаёт общей композиции контраст и глубину.

*(The photograph features the President and CEO of The Walt Disney Company standing in front of a model of Shanghai Disneyland. In the background, vibrant floral arrangements and green bushes create a festive atmosphere. In the foreground, a detailed model of the park is displayed, with a castle adorned with numerous towers and spires. The President, smiling and extending his arms as if presenting the miniature park, is dressed in a dark suit and a striped shirt, emphasizing his official status. The backdrop is designed in blue, adding contrast and depth to the overall composition.*

**Caption without Alt Text:**

На изображении мужчина в тёмном костюме, светло-голубой рубашке и без галстука, который стоит перед большим макетом замка. Замок имеет ярко выраженные готические элементы с высокими шпилями, что напоминает замки из сказок Диснея. Мужчина улыбается и раскинул руки в приветственном жесте, демонстрируя замок. На заднем плане сцены можно увидеть зелёные растения, цветы в горшках и красный фон с золотыми элементами, что создаёт праздничную атмосферу. Позади замка можно заметить синий задний фон, который придаёт всей сцене яркость.

*(The image features a man in a dark suit, light blue shirt, and no tie, standing in front of a large model of a castle. The castle has pronounced Gothic elements with tall spires, reminiscent of Disney fairy tale castles. The man is smiling and has his arms spread in a welcoming gesture, showcasing the castle. In the background, green plants, potted flowers, and a red backdrop with gold elements create a festive atmosphere. Behind the castle, a blue background adds brightness to the entire scene.)*

Figure 8: An example from LAION-Cultural illustrating why the filtered informative alt text helps generate a more informative caption. With the high-quality alt text, the model incorporates important details like *"President and CEO of The Walt Disney Company standing in front of a model of Shanghai Disneyland"* into the generated caption.

## D  DATASETS USED IN PANGEABENCH

To comprehensively assess the capabilities of PANGEA across diverse languages, cultures, and task types, we developed PANGEABENCH. We list the details of each dataset included in the PANGEABENCH.

### D.1  MULTIMODAL DATASETS

- **xGQA** (Pfeiffer et al., 2022a): A cross-lingual visual question-answering dataset featuring 9,666 questions in eight languages covering five scripts. The dataset includes 300 unique images from Visual Genome (Krishna et al., 2017). xGQA tests the model's ability to understand and reason about visual content across multiple languages.

- **MaXM** (Changpinyo et al., 2022): A VQA dataset in seven languages and five scripts, with questions and answers in the same language. Images are culturally matched to the target language regions. MaXM specifically addresses the challenge of cultural diversity in multimodal understanding.

- **MaRVL** (Liu et al., 2021): A Multicultural Reasoning over Vision and Language dataset in five languages and three scripts, featuring 4,914 culturally diverse images matched to respective languages. MaRVL focuses on evaluating models' ability to reason about culturally diverse visual concepts.

- **XM100** (Thapliyal et al., 2022): We create a subset of 3600 instances (100 instances per language) from the original XM100 dataset, a large multilingual image captioning dataset comprising 36 languages, with 261,375 captions for 100 unique images per language, culturally matched to each language. XM100 evaluates a model's ability to generate culturally appropriate captions across a wide range of languages. For sampling, we select 100 instances per language, ensuring that all languages share the same set of images for their respective 100 instances. To ensure diversity within our sample, we use Sentence-BERT (Reimers & Gurevych, 2019) to cluster the 3600 English instances from the original dataset into 100 groups, and then select one instance from each group. This method ensures that the sampled instances are as diverse as possible. We evaluate models on this new sample of 3600 instances, which allows for a more time-efficient evaluation while still accurately reflecting the multilingual capabilities of models in diverse contexts.

- **M3Exam** (Zhang et al., 2023): A novel benchmark sourced from real and official human exam questions, featuring 12,317 questions in 9 languages across three educational levels. Approximately 23% of the questions require image processing. M3Exam tests the model's ability to handle complex, multi-step reasoning tasks in an educational context.

- **xMMMU**: MMMU contains multimodal questions from college-level materials across six disciplines and 30 subjects. The dataset features 183 subfields and 30 diverse image types, including charts, diagrams, and chemical structures. We sample 300 questions from the original MMMU validation set and translate them using GPT-4o into xx languages. To ensure the quality, we translated each sampled question multiple times and then back-translated it to English. We select the translation with the highest BLEU score. xMMMU evaluates the model's capacity to understand and reason about specialized academic content across languages and modalities.

### D.2  TEXT-ONLY MULTILINGUAL DATASETS

- **TyDiQA** (Clark et al., 2020): A question answering dataset covering 11 typologically diverse languages with 204K question-answer pairs. Questions are written by native speakers without seeing the answers, ensuring a realistic information-seeking task. TyDiQA is designed to test linguistic diversity and avoid translation artifacts.

- **FLORES** (NLLB Team, 2024): A machine translation benchmark for 200 languages, including many low-resource languages. It consists of 3,001 sentences from 842 web articles, divided into dev, devtest, and test splits. FLORES-200 includes translations from multiple pivot languages and provides script alternatives for some languages, making it a comprehensive test of translation capabilities.

- **MMMLU** (OpenAI, 2024a): A human-translated version of MMLU (Hendrycks et al., 2021), covering 57 subjects across STEM, humanities, social sciences, and more. It ranges in difficulty

from elementary to advanced professional levels, testing both world knowledge and problem-solving ability in a zero-shot and few-shot setting across multiple languages.

- **MGSM** (Shi et al., 2022): Multilingual Grade School Math Benchmark, featuring 250 grade-school math problems translated into 10 languages. Based on GSM8K, it requires multi-step reasoning and tests the model's ability to solve complex mathematical word problems across languages.

This diverse set of datasets in PANGEABENCH allows for a comprehensive evaluation of PANGEA's capabilities across various languages, cultures, modalities, and task types, providing a holistic assessment of its performance in multilingual and multimodal contexts.

# E    EXPLANATION OF XCHATBENCH

**Task Category**    We first divide into 10 task categories, namely *art_explanation*, *bar_chart_interpretation*, *defeasible_reasoning*, *figurative_speech_explanation*, *iq_test*, *ocr*, *graph_interpretation*, *image_humor_understanding*, *science_figure_explanation*, *unusual_images*. The task categories are inspired by existing papers that do not use a free-form generation format (Lu et al., 2024; Yue et al., 2024a; Han et al., 2023; Hessel et al., 2023; Kim et al., 2022).

**Construction Procedure**    To annotate the instances, we mainly follow the procedure of Kim et al. (2024a). Two human annotators first hand-crafted the instances by searching through appropriate images for the task and then hand-crafting each component of the instance. As our motivation for fine-grained evaluation, each instance consists of not only an **instruction**, **reference answer**, but also a unique **evaluation criteria** tailored to each instance (*e.g.*, Does the response effectively explain the humor in the image based on the juxtaposition of a character's portrayal in different scenarios?) and a **description for each score** between 1 and 5 (*e.g.*, score4_description: The response understands the juxtaposition and relates it to the humor involving machine learning models, but may miss some nuances or the related aspect of the humor). During the annotation process, we asked the annotators to not copy-and-paste results from LLM services like ChatGPT or directly from the web. Then, we hire four additional annotators to assess the quality of the instances. Each participant to asked to grade if each instance (1) fits into the devised task category, (2) if the quality of the reference answer is good enough, and (3) if the score rubric is suitable to assess the response. We iteratively ask the annotators who made the instances to revise them if the instance does not satisfy all three criteria. The resulting dataset consists of 50 instructions, reference answers, and evaluation criteria with a corresponding score rubric.

**Translation Procedure**    To assess the multilingual generation capabilities of MLLMs, we translate the hand-crafted 50 instances into 6 different languages, namely Chinese, Hindi, Indonesian, Japanese, Korean, and Spanish. We first use GPT-4o-2024-08-06 to translate the instruction and reference answer of each instance with a naive prompt, "Translate the following sentences into {target_language}. Sentences: {sentences}". Then, the coauthors who are native speakers of each language reviewed the instances and made adjustments if the translated results were unnatural.

**Evaluation Pipeline**    Similar to prior works employing LLM-as-a-Judge, we use GPT-4o-2024-08-06 as the judge model and prompt it in a direct assessment manner. As input, the judge model is given the instruction, the model's response, the reference answer, the evaluation criteria, and the descriptions for each score. As output, the judge generates verbal feedback and an integer score between 1 and 5. For this procedure, we use the `prometheus-eval` library (Kim et al., 2024b) and employ their default hyperparameter setting for evaluation. Lastly, the final score is acquired by averaging the results across the 50 instances for each language. Note that in the main result and breakdown result tables, we normalize the score from 1-5 to 0-100 by $(\text{score} - 1) \times 25$. For the multimodal chat scenarios, we found that many English-centric models tend to respond in English regardless of the query language. This behavior is problematic, as it undermines the fundamental capability of a multilingual model, which should ideally respond in the language of the query. To address this, we implemented a strict evaluation criterion where such responses were penalized and assigned a score of 0. We believe this is crucial, as users may not understand English, and failing to respond in the appropriate language can hinder effective communication and user experience. Thus, for postprocessing, we use langdetect [2] to identify whether the response is written in the given language and change the score to 1 when it is written in a different language, a phenomenon called *language hallucination* (Xue et al., 2021; Pfeiffer et al., 2023).

---

[2]https://pypi.org/project/langdetect/

# F   QUALITATIVE EXAMPLES FROM XCHATBENCH

One important application of MLLMs is to answer users' queries in the wild. Here, we show the outputs of PANGEA for the multimodal chat queries from our xChatBench. The examples included the scoring rubric, query, response from our PANGEA, reference answer, and LLM-as-Judge feedback. As shown in Appendix Figure 9, 10, 11, 12, 13, 14, PANGEA successfully interprets the figures in different tasks and generates fluent and readable in certain languages. These qualitative examples further demonstrate the remarkable visual understanding ability of PANGEA in multilingual contexts. On the other hand, we also identified a few bad cases shown in Figure 15, 16. Despite generating relevant responses to the queries, the model does not capture the key details of the images due to the lack of knowledge, which points out potential improvement directions in the future.

**Score Rubric**

**Does the response comprehensively analyze the humorous elements present in the image featuring an animal with human-like emotions?**

Score 1: The response does not recognize the anthropomorphism or the situational context, offering no insight into the humor.
Score 2: The response identifies the animal's human-like emotions but fails to connect these to the situational context or the cultural nuances that contribute to the humor.
Score 3: The response notes the anthropomorphism and situational context but provides only a basic explanation of how these elements combine to create humor.
Score 4: The response gives a good explanation of the humor, recognizing both the anthropomorphism and situational context, yet it might miss finer cultural nuances.
Score 5: The response offers a detailed analysis, accurately recognizing the anthropomorphism, situational context, and cultural nuances that all contribute to the humor.

**Instruction**

動物が人間のような感情を持つ様子をユーモラスな文脈で描かれた画像を見るとき、その画像に存在する文化的なニュアンス、背景情報、または状況的な文脈を考慮しながら、コメディ効果に寄与する根本的な要素を分析してください。

When looking at an image that humorously portrays animals with human-like emotions, analyze the fundamental elements contributing to the comedic effect, while considering the cultural nuances, background information, or situational context present in the image.

When you send a message to a friend who's online and right after that, they go offline

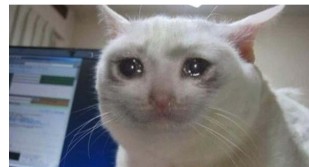

**Response from Pangea**

画像に寄与するコメディ効果にはいくつかの要素があります。第一に、猫は人間のように泣いている表情で描かれており、これはユーモラスな対比です。第二に、猫がスマートフォンを操作している様子は、デジタルコミュニケーションでよく見られる人間の行動を描いています。第三に、猫が「メッセージ」を受け取った後にもうすぐ「オフライン」となりますというこの行動は、猫がメッセージを読んですぐに返事をしようと努力していることを示唆しており、これは人間の行動であり、特に猫はそうしないとされるため、面白い誤解です。これらの要素は、猫がインターネットやメッセージングアプリなどのデジタルコミュニケーションの文脈において、人間と同じように行動し、同じような感情を持っているかのような、ユーモラスで過度化されたシーンを作り出しています。

There are several elements contributing to the comedic effect in the image. First, the cat is depicted with a crying expression, much like a human, which creates a humorous contrast. Second, the cat is operating a smartphone, portraying a behavior often seen in human digital communication. Third, the action of the cat soon going "offline" after receiving a "message" suggests that it is trying to respond immediately, mimicking human behavior, which is amusingly mistaken since cats are not known to do this. These elements together create a humorous, exaggerated scene where the cat acts and feels like a human in the context of internet and messaging apps, which adds to the comedy.

**Reference Answer**

画像には、涙ぐんだ目によって強調された、悲しみや失望の人間のような表情をした猫が描かれています。ユーモアは、人間ではないものに人間の特性を与えるという一般的なコメディ手法である擬人化から生まれます。「オンラインの友達にメッセージを送った直後に、その友達がオフラインになるとき」というキャプションが付けられています。この状況は、デジタルコミュニケーションにおいてよくあるシナリオであり、相手がすぐにオフラインになることで、交流の試みが無視されたり軽視されたりしたと感じることがあります。これは、即時のコミュニケーションの期待と、しばしば暗黙のデジタルエチケットを利用した状況的アイロニーです。\n\nこの状況の文化的な関連性によって、コメディ効果はさらに高まります。多くの人がデジタル時代における会話の突然の終わりを経験しています。画像は猫の表情を巧みに利用して、その瞬間の感情的なインパクトを誇張し、テクノロジーが私たちの社会的交流に与える影響についての共通理解に訴えかけています。インターネット文化の重要な一部であるミーム形式自体が、しばしばテキストと画像を組み合わせて多面的なユーモアを伝える、共感できるコンテンツを広めるために設計されています。したがって、猫の人間化された表情、一般的なオンライン交流へのキャプションの言及、デジタルコミュニケーションにおけるミームの位置を認識することで、この画像が意図するユーモアの全体像を理解することができます。

The comedic effect is further enhanced by the cultural relevance of this situation. Many people have experienced the abrupt end of a conversation in the digital age. The image cleverly uses the cat's expression to exaggerate the emotional impact of that moment, appealing to a shared understanding of how technology affects our social interactions. The meme format itself, an essential part of internet culture, is often designed to combine text and images to convey multi-layered humor and spread relatable content. Therefore, by recognizing the anthropomorphized expression of the cat, the caption's reference to common online interactions, and the role of memes in digital communication, we can fully grasp the humor intended by this image.

**Feedback from GPT-4o**

The response identifies various elements that contribute to the comedic effect in the image, such as the cat displaying a human-like crying expression and interacting with a smartphone. It recognizes the anthropomorphism and outlines a scenario where the cat behaves as if it were in a human situation related to digital communication. However, while there is a description of the elements and the anthropomorphic humor, the analysis could be strengthened by delving deeper into cultural nuances and how they enhance humor, as well as the broader social commentary on digital communication. While the response adequately covers the situational context, it doesn't thoroughly explore the cultural aspects to the same extent as the reference answer.

[RESULT] 4

Figure 9: An xChat example from Japanese subset and our model PANGEA's response. The English part of the instruction, response, and reference answer is additionally added only on the figure to help to understand and was not given nor generated by PANGEA.

**Score Rubric**

**Does the response accurately transcribe all visual information from the image, including specific details such as quantities, weights, underlines, and ink colors?**

Score 1: The response recognizes the presence of text but fails to accurately transcribe any of the materials and their corresponding quantities and weights, for example, quantities are present but weights are not present.
Score 2: The response accurately transcribes the information for one of the materials listed, including quantity and weight, but fails to do so for the other two, for example, 'Card Stock' data is correct but 'Construction Paper' and 'Fabric' data are missing or incorrect.
Score 3: The response accurately transcribes the information for two of the materials listed, including quantities and weights, but fails to do so for the third, for example, 'Card Stock' and 'Construction Paper' are correct, but 'Fabric' is not.
Score 4: The response accurately transcribes the quantities and weights for all materials but misses secondary details such as underlines or ink colors, for example, all weights are correct but underlines and ink colors are not mentioned.
Score 5: The response accurately transcribes all primary information (quantities and weights) and secondary details (underlines, ink colors) for all materials, with no omissions.

**Instruction**

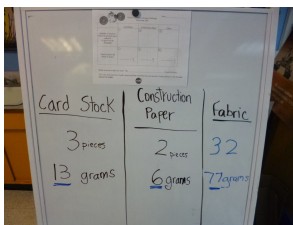

डिज़ाइन चुनौती के दौरान, एक बोर्ड पर विभिन्न सामग्रियों और उनकी संबंधित मात्राएँ और वजन प्रदर्शित किए गए थे    रिकॉर्ड रखने के लिए एक तस्वीर ली गई थी    हालाँकि, छवि थोड़ी धुंधली है और हस्तलेखन शैली और आकार में भिन्न है    छवि की जाँच करें और प्रदर्शित जानकारी का स्पष्ट और सटीक लिप्यंतरण प्रदान करें, जिसमें कार्ड स्टॉक, निर्माण कागज, और कपड़े की मात्राएँ और वजन शामिल हों

During a design challenge, various materials and their respective quantities and weights were displayed on a board. A picture was taken for record-keeping, but the image is slightly blurry, with variations in handwriting style and size. Please examine the image and provide a clear and accurate transcription of the displayed information, including the quantities and weights of card stock, construction paper, and fabric.

**Response from Pangea**

सामग्री प्रदर्शन स्क्रीन पर निम्नलिखित जानकारी है:\n\n| मात्रा | कार्ड स्टॉक | निर्माण कागज | कपड़े |\n|---|---|---|---|\n| मात्रा | 3 | 2 | 32 \n| वजन (ग्राम) | 13 | 6 | 77 |\n\nसाइनेज से, हम निष्कर्ष निकाल सकते हैं कि परीक्षण में कार्ड स्टॉक की तीन टुकड़ों का वजन 13 ग्राम, निर्माण कागज की दो टुकड़ों का वजन 6 ग्राम, और कपड़े की अग्रणी छह-अंक संख्या का वजन 77 ग्राम है    व्यवस्था बताती है कि प्रत्येक विभाग सामग्री के वजन में योगदान को दर्शाता है, लेकिन वजन आकार चर के रूप में व्यक्त किए गए हैं, जो संभवतः कोई माप या वर्ग का प्रतिनिधित्व करता है    सटीक आयाम या सामग्री का उपयोग के बारे में अधिक जानकारी के बिना, यह निर्धारित करना संभव नहीं है कि किस परीक्षण के लिए डेटा प्रस्तुत किया गया है या कौन सी सामग्री को संदर्भित किया गया है

| Quantity  | Card Stock | Construction Paper | Fabric |\n|-----------|------------|--------------------|--------|\n| Quantity  | 3      | 2      | 32    | \n| Weight (grams) | 13    | 6      | 77    |\n\nFrom the signage, we can conclude that three pieces of card stock weigh 13 grams, two pieces of construction paper weigh 6 grams, and the leading six-digit number for the fabric weighs 77 grams. The arrangement suggests that each section represents the material's contribution to the overall weight, but the weight is expressed as a variable, possibly representing some form of measurement or category. Without more precise dimensions or information on the materials used, it is not possible to determine what specific test the data pertains to or which materials are being referenced.

**Reference Answer**

छवि एक व्हाइटबोर्ड दिखाती है जिस पर मार्कर से जानकारी लिखी हुई है    इसमें तीन कॉलम हैं, प्रत्येक एक अलग सामग्री के लिए समर्पित है: कार्ड स्टॉक, निर्माण कागज, और कपड़ा    प्रत्येक सामग्री के लिए, मात्रा (टुकड़ों में) और वजन (ग्राम में) दिए गए हैं    \n\nकार्ड स्टॉक\n\nमात्राः 3 टुकड़े\nवजनः 13 ग्राम\nनिर्माण कागज\n\nमात्राः 2 टुकड़े\nवजनः 6 ग्राम\nकपड़ा\n\nमात्राः 32\nवजनः 77 ग्राम\nवजनों को जोर देने के लिए रेखांकित किया गया लगता है    'कार्ड स्टॉक' और 'निर्माण कागज' के लिए लिखावट काले स्याही में है, और संख्याएँ भी काले रंग में लिखी गई हैं    'कपड़ा' शब्द और इसके संबंधित संख्याएं नीले स्याही में लिखी गई हैं    फॉन्ट शैलियाँ आकस्मिक और आकार में विविध हैं, लेकिन जानकारी फिर भी पठनीय है    व्हाइटबोर्ड के शीर्ष दाएं कोने में एक मुद्रित फॉर्म भी है, लेकिन इस फॉर्म का पाठ उस कोण और दूरी के कारण पठनीय नहीं है जिस पर फोटो लिया गया था

The image shows a whiteboard with information written in marker. There are three columns, each dedicated to a different material: card stock, construction paper, and fabric. For each material, the quantity (in pieces) and weight (in grams) are given.\nCard Stock:\n * Quantity: 3 pieces\n* Weight: 13 grams\n\nConstruction Paper\n* Quantity: 2 pieces\n* Weight: 6 grams\nFabric\n* Quantity: 32\n* Weight: 77 grams\n\nThe weights appear to be underlined for emphasis. The writing for "Card Stock" and "Construction Paper" is in black ink, and the numbers are also written in black. The word "Fabric" and its related numbers are written in blue ink. The font styles are casual and vary in size, but the information is still readable. In the top right corner of the whiteboard, there is a printed form, but the text on the form is not readable due to the angle and distance at which the photo was taken.

**Feedback from GPT-4o**

The response provides an accurate transcription of the primary information, namely the quantities and weights for all three materials: card stock, construction paper, and fabric. However, it does not mention any secondary details such as whether weights are underlined or the specific ink colors used in the original board, which are valuable details as indicated in the score rubric for a perfect score. This lack of mention of secondary details like underlines and ink colors prevents it from receiving a higher score.

[RESULT] 4

Figure 10: An xChat example from Hindi subset and our model PANGEA's response. The English part of the instruction, response, and reference answer is additionally added only on the figure to help to understand and was not given nor generated by PANGEA.

**Score Rubric**

**Does the response effectively explain the historical context of the phrase 'KEEP CALM AND CARRY ON' and analyze the symbolism of the elements within the image?**

Score 1: The response mentions the phrase but fails to provide any historical context or symbolic analysis.
Score 2: The response provides a generic historical context without specific details or a clear explanation of the symbolism.
Score 3: The response gives an accurate historical context or symbolic analysis, but not both. For example, it may mention World War II but not the significance of the crown or color.
Score 4: The response provides a detailed historical context and a basic symbolic analysis. For example, it explains the phrase's origin in World War II and the general significance of the crown, but not the color red or its modern cultural impact.
Score 5: The response offers a comprehensive explanation of both the historical context and the symbolism. For example, it details the phrase's creation during World War II, the authority implied by the crown, the urgency indicated by the color red, and the phrase's modern-day relevance.

**Instruction**

제시된 시각적 메시지를 분석할 때, 어떤 역사적 맥락이 이러한 문구를 발생시켰을 수 있으며, 이 맥락이 이 이미지 내 요소들의 상징성과 사용과 어떻게 연결되는가?

When analyzing the given visual message, what historical context might have generated this phrase, and how is this context connected to the symbolism and usage of the elements within the image?

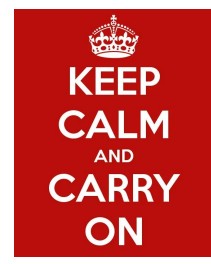

**Response from Pangea-7B**

\"KEEP CALM AND CARRY ON\"이라는 문구는 영국 왕실이 2차 세계대전 중 국가 정신을 높이기 위해 발행한 포스터에서 유래했습니다. 이 문구를 발생시킨 역사적 맥락은 위기 시대에 국민들에게 격려와 안정을 제공하기 위한 것이었습니다. 이 이미지 내 요소들은 전통적인 \"Keep Calm and Carry On\" 메시지를 유지하면서도 현대적인 느낌을 더하여 현대에도 그 목적을 유지하고 있습니다. 굵은, 대문자로 된 텍스트, 굵은 글씨체, 왕관의 상징성은 모두 원래 포스터에 있던 것과 비슷합니다. 이는 문구가 과거의 역사적 맥락에서 온 것으로서 의미가 지속되고 있으며, 현재에도 사용되고 있음을 나타냅니다.

The phrase "KEEP CALM AND CARRY ON" originates from a poster issued by the British royal family during World War II to boost national morale. The historical context that produced this phrase was aimed at providing encouragement and stability to the public during a time of crisis. The elements within this image maintain the traditional "Keep Calm and Carry On" message while adding a modern feel, suggesting that the original purpose remains relevant today. The bold, uppercase text, thick font, and the symbolic crown are all similar to those in the original poster. This indicates that the phrase, rooted in its historical context, continues to hold meaning and is still in use today.

**Reference Answer**

이미지는 밝은 빨간색 배경에 왕실 문장이 얹혀진 \"KEEP CALM AND CARRY ON\"이라는 대담하고 대문자로 된 문구를 묘사하고 있습니다. 이 상징적인 포스터는 제2차 세계 대전 초기에 영국 역사에 뿌리를 두고 있습니다. 이는 1939년 영국 정부에 의해 잠재적인 전시 재난에 대비하여 대중의 사기를 북돋우기 위한 메시지로 원래 제작되었습니다.\n\n\"KEEP CALM AND CARRY ON\"이라는 은유는 다면적입니다. 이는 대중에게 혼란 속에서도 침착함과 정상성을 유지하라고 촉구하는 회복력과 강인함의 메시지를 담고 있습니다. 왕관의 사용은 단순히 장식적인 것이 아니라 군주제를 상징하며, 영국 국민의 집단적 정체성에 대한 호소와 함께 단결과 국가적 자부심을 불러일으킵니다. 메시지 위의 왕관은 이 지시가 군주제와 관련된 권위와 안정성을 가지고 있다는 것을 암시적으로 나타냅니다.\n\n빨간색은 주목을 끌며 경계와 긴급함과 자주 연관되는데, 이는 전시 중에 적절한 감정이었을 것입니다. 그러나 그 자체로 명령형인 문구는 배경의 경고하는 색조와 대조적으로 침착한 숙고의 감각을 심어주려 합니다. 이 대조는 행동의 긴급함이 정신의 평온함에 의해 완화되는 은유의 깊이를 강조합니다.\n\n현대 문화에서 밈과 장식 모티프로서의 문구의 지속성과 부활은 그 은유적 풍부함을 더욱 강조합니다. 이는 전쟁의 맥락뿐 아니라 일상적인 도전에 대한 냉철한 대응을 상징하게 되었으며, 그 원래 목적을 초월한 시대를 초월한 메시지를 담고 있습니다. 여기서 은유는 인생의 어떤 어려움에도 개인의 회복력으로 확장되어 보편적으로 적용 가능한 사고방식을 상징합니다.\n\n본질적으로 이 포스터는 국가의 역사적 회복력을 전달하면서 집단적이고 개인적인 도전에 대한 침착하고 확고한 접근 방식을 제공하는 지속적인 은유를 전달하는 유물입니다.

The image depicts the bold, uppercase phrase "KEEP CALM AND CARRY ON" set against a bright red background with a royal crest. This iconic poster is rooted in British history during the early days of World War II. It was originally created in 1939 by the British government to boost public morale in the face of potential wartime disaster.\nThe metaphor of "KEEP CALM AND CARRY ON" is multifaceted. It conveys a message of resilience and strength, urging the public to maintain calm and normalcy amidst chaos. The use of the crown is not merely decorative but symbolizes the monarchy, evoking unity and national pride within the collective British identity. The crown above the message implicitly suggests that the directive carries the authority and stability associated with the monarchy.\nRed is a color that grabs attention and is often linked to caution and urgency, which would have been appropriate during wartime. However, the commanding phrase, in contrast to the warning-like tone of the background, aims to instill a sense of calm reflection. This contrast highlights the depth of the metaphor, where the urgency of action is tempered by a sense of mental composure.\nIn modern culture, the phrase's persistence and revival as a meme and decorative motif emphasize its metaphorical richness. It has come to symbolize a calm response not only in the context of war but also to everyday challenges, representing a timeless message that transcends its original purpose. Here, the metaphor extends to symbolize personal resilience in the face of life's difficulties, making it universally applicable.\nEssentially, this poster serves as an artifact that communicates the historical resilience of a nation while offering a calm, steadfast approach to both collective and personal challenges.

**Feedback from GPT-4o**

The response begins by providing the historical context of the phrase \"KEEP CALM AND CARRY ON,\" correctly stating its origin during World War II, and explaining its purpose to boost national morale in a time of crisis. This establishes a basic historical framework for understanding the significance of the phrase. \n\nIn terms of symbolic analysis, the response does acknowledge elements such as the bold, uppercase text, bold font, and the crown's symbolism. However, the analysis remains somewhat superficial. It does not delve into the specific symbolic significance of the color red or provide a detailed explanation of how these elements function within the historical and modern contexts. Furthermore, it does touch on the modern relevance of the phrase but only in vague terms, without explaining how the symbolism translates into contemporary culture or addressing the broader implications noted in the reference answer.\n\nOverall, while the response demonstrates an understanding of the historical origins and some elements of symbolism, it lacks the depth and comprehensiveness of the reference answer.

[RESULT] 4

Figure 11: An xChat example from the Korean subset and our model PANGEA's response. The English part of the instruction, response, and reference answer is additionally added only on the figure to help to understand and was not given nor generated by PANGEA.

**Score Rubric**

> **Does the response insightfully analyze the underlying feelings and interactions at the festive gathering?**
>
> Score 1: The response briefly acknowledges the festive setting but fails to delve into the emotions or interactions, for example, noting the Christmas hats but not the expressions or body language of joy and camaraderie.
> Score 2: The response recognizes happiness in the gathering but does not connect it to broader social dynamics, for example, mentioning smiling faces but not the shared amusement or sense of belonging.
> Score 3: The response describes surface-level interactions and some emotions, such as laughter, but lacks depth in exploring the significance of these in the context of human connection, for example, noting the toasting but not the deeper sense of unity it signifies.
> Score 4: The response captures most of the emotional essence and social interactions, like engagement and warmth, but may not fully interpret the broader human experience, for example, recognizing close proximity but not fully discussing the collective mood it creates.
> Score 5: The response provides a comprehensive analysis of the feelings and interactions, deeply understanding the joy, belonging, and human connection present at the event, reflecting on how these elements combine to create a rich, shared human experience.

**Instruction**

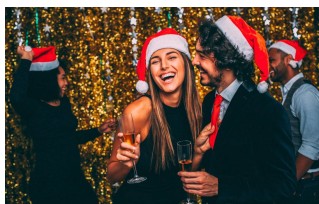

Dalam sebuah pertemuan meriah yang tertangkap, apa yang mungkin menjadi perasaan dan interaksi mendasar antara orang-orang yang hadir, dan bagaimana hal ini mencerminkan emosi manusia yang lebih luas dan dinamika sosial pada acara perayaan?

In a festive gathering captured in an image, what might be the underlying feelings and interactions among the people present, and how does this reflect broader human emotions and social dynamics at celebratory events?

**Response from Pangea-7B**

Berdasarkan gambar, perasaan dan interaksi mendasar di antara orang-orang di sana tampak penuh kebahagiaan dan keterhubungan. Senyum lebar wanita dan gestur mesra pria menunjukkan suasana gembira dan intim yang sering terjadi dalam pertemuan sosial selama perayaan. Interaksi seperti ini mencerminkan emosi manusia universal pada saat mereka merayakan - rasa persahabatan, kesenangan bersama, dan kenikmatan dari pengalaman bersama. Momen-momen semacam itu adalah ciri khas dari acara sosial dan merupakan indikasi hubungan dan empati dalam konteks komunitas atau kelompok. Gambar tersebut menangkap esensi dari dinamika sosial tersebut dengan mengabadikan reaksi spontan manusia dalam lingkungan perayaan yang meriah.

Based on the image, the underlying feelings and interactions among the people seem to be full of happiness and connection. The wide smile of the woman and the affectionate gestures of the man suggest a joyful and intimate atmosphere often found in social gatherings during celebrations. These interactions reflect universal human emotions when celebrating—feelings of friendship, shared enjoyment, and the pleasure of shared experiences. Such moments are typical of social events and are indicative of bonding and empathy within a community or group. The image captures the essence of these social dynamics by immortalizing spontaneous human reactions in a lively festive environment.

**Reference Answer**

Gambar tersebut memancarkan suasana semarak yang khas dari perayaan liburan, dengan individu-individu yang mengenakan topi Natal, menunjukkan adegan persahabatan yang ceria. Tokoh-tokoh utama, seorang pria dan wanita, terlibat dalam momen hiburan bersama, sebagaimana dibuktikan oleh senyum lebar wanita tersebut dan ekspresi halus dan puas dari pria tersebut. Mereka berdua memegang gelas, tanda bersulang sosial dan kenikmatan pesta, menunjukkan semangat perayaan yang ramah dari acara tersebut.\n\nMata wanita itu bersinar dengan tawa, mencerminkan kegembiraan dan kebebasan yang sering dirasakan selama pertemuan semacam itu. Wajahnya yang terbuka dan ekspresif menandakan rasa memiliki dan kenikmatan, sifat-sifat yang selaras dengan suasana kolektif kehangatan dan keceriaan. Keterlibatan pria dengan wanita tersebut melalui kedekatan dan bahasa tubuh menunjukkan kebutuhan manusia akan hubungan dan perayaan bersama orang lain.\n\nDi latar belakang, peserta pesta lainnya sama-sama terhanyut dalam esensi semangat perayaan. Latar belakang yang kabur dengan lampu berkelap-kelip melambangkan kilauan keajaiban liburan, dan rona emas secara keseluruhan memberikan cahaya yang tampaknya meningkatkan kehangatan emosional dari adegan tersebut. Para peserta tidak hanya hadir; mereka adalah kontributor aktif dalam jalinan pengalaman manusia yang ditenun oleh acara semacam itu.\n\nGambar ini menangkap lebih dari sekadar momen pesta; ia merangkum jalinan kaya dari hubungan emosional yang secara universal beresonansi. Ini adalah pengingat akan kemanusiaan bersama yang dapat disoroti oleh acara perayaan, membawa individu-individu bersama dalam ekspresi kolektif kegembiraan, persahabatan, dan keinginan manusia untuk perayaan komunal.

The image radiates the festive spirit typical of holiday celebrations, with individuals wearing Christmas hats, displaying a scene of cheerful camaraderie. The central figures, a man and a woman, are engaged in a moment of shared amusement, as evidenced by the woman's broad smile and the man's soft, contented expression. Both are holding glasses, a sign of social toasting and party enjoyment, reflecting the friendly celebratory spirit of the event.\n\nThe woman's eyes sparkle with laughter, reflecting the joy and freedom often felt during such gatherings. Her open and expressive face signifies a sense of belonging and enjoyment, emotions that align with the collective warmth and cheerfulness of the occasion. The man's engagement with the woman through proximity and body language highlights the human need for connection and celebration with others.\n\nIn the background, other party attendees are similarly absorbed in the festive spirit. The blurred background with twinkling lights symbolizes the magic of the holiday season, and the overall golden hue adds a glow that seems to enhance the emotional warmth of the scene. The participants are not merely present; they are active contributors to the shared human experience woven together by such events.\n\nThis image captures more than just a party moment; it encapsulates the rich tapestry of emotional connections that resonate universally. It is a reminder of the shared humanity that can be highlighted by celebratory events, bringing individuals together in a collective expression of joy, friendship, and the human desire for communal celebration.

**Feedback from GPT-4o**

The response insightfully captures the fundamental feelings and interactions of happiness and connection present at the festive gathering. It discusses the significance of smiles and gestures in conveying joy and camaraderie, and it links these interactions to universal human emotions like friendship and shared pleasure. However, while it does touch on dynamics such as empathy and community, it doesn't fully delve into the broader human experience or collective mood in as much depth as the reference answer does. The reference answer provides a more comprehensive interpretation, connecting these elements to a shared human experience with detailed observation of facial expressions and social cues.

[RESULT] 4

Figure 12: An xChat example from the Indonesian subset and our model PANGEA's response. The English part of the instruction, response, and reference answer is additionally added only on the figure to help to understand and was not given nor generated by PANGEA.

**Score Rubric**

Does this response show a well-rounded preparatory plan for discussing the digestive system in a medical program context?

Score 1: The response does not include any preparatory topics or details relevant to the digestive system.
Score 2: The response mentions one aspect of digestive system study, such as physiology, without additional context or related areas of study.
Score 3: The response includes basic information on several aspects of the digestive system, such as physiology and disease, but lacks depth or the inclusion of diagnostic and treatment approaches.
Score 4: The response provides a good overview, covering multiple aspects like physiology, pathophysiology, diagnostics, and nutrition, but may lack details on pharmacology or interconnected systems.
Score 5: The response thoroughly covers all aspects of the digestive system study, including detailed physiology, pathophysiology, diagnostics, nutrition, pharmacology, and current research, showing an in-depth understanding and preparatory plan.

**Instruction**

Esta es una figura básica de lo que aprendí en la escuela secundaria. He ingresado a un programa médico como estudiante universitario y debería explicar a un profesor lo que me gustaría aprender más durante la universidad. ¿Podrías ayudarme a añadir más detalles sobre lo que debería buscar y examinar antes de la reunión?

This is a basic figure from what I learned in high school. I have entered a medical program as a college student and should explain to a professor what I would like to learn more about during university. Could you help me add more details on what I should look for and examine before the meeting?

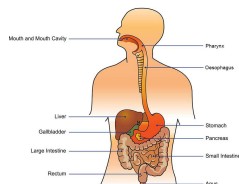

**Response from Pangea-7B**

Basándonos en los conocimientos básicos que probablemente obtuviste en la escuela secundaria y teniendo en cuenta el contexto médico, es posible que quieras profundizar en los siguientes temas antes de tu reunión:\n\n1. **Función y estructura del sistema digestivo**: Explora cómo funciona cada órgano y cómo contribuye al proceso digestivo. Esto puede incluir la función del esfínter anal, el movimiento gástrico, la absorción de nutrientes en el intestino delgado, la secreción de enzimas pancreáticas y suelto del páncreas, la producción de bicarbonato hepático y la función del cálculo biliar.\n\n2. **Afirmaciones y condiciones del sistema digestivo**: Estudia las enfermedades y afecciones comunes relacionadas con la digestión, como úlceras gastrointestinales, enfermedades inflamatorias del tráqueo (IEP), cáncer gástrico, enfermedad de Crohn y síndrome del intestino irritable. Comprende cómo se diagnostican y tratan estas afecciones.\n\n3. **Biología molecular y bioquímica**: Aprende más sobre la base química de los procesos digestivos, como la hidrólisis de carbohidratos, proteínas y grasa; el papel de enzimas específicas y las vías metabólicas implicadas en la digestión.\n\n4. **Fisiología médica**: Examina los procesos fisiológicos, como la secreción gastrointestinal, la regulación de los niveles hormonales que rigen los patrones de alimentación y la función de las vías digestivas en respuesta a la nutrición y la digestión.\n\n5. **Evolución y adaptación biológica**: Considera cómo se ha adaptado el sistema digestivo a diferentes formas de alimentación y cómo esta adaptación ha influido en la evolución humana. También puedes investigar sobre otras especies con sistemas digestivos distintos para ver cómo se han adaptado.\n\n6. **Ciencia forense e historia médica**: Examina cómo el estudio del sistema digestivo se utiliza en la autopsia y cómo ha evolucionado a lo largo de la historia. Esto puede ayudarte a entender las diferencias entre la anatomía y la biología modernas y aquellas de períodos históricos.\n\n7. **Práctica clínica y habilidades de atención al paciente**: Aprende sobre los procedimientos médicos relacionados con la digestión, como endoscopias, exploraciones abdominales y terapia con antibióticos para infecciones bacterianas. También debes familiarizarte con las técnicas de comunicación médica y las consideraciones éticas para la atención al paciente.\n\n8. **Cirugía digestiva y endocrinología**: Conoce las cirugías que se realizan para tratar afecciones digestivas y cómo el sistema digestivo interactúa con el sistema endocrino, particularmente con el sistema gastroenterológico.\n\n9. **Nutrición dietética y alimentación saludable**: Aprende sobre la importancia de una dieta equilibrada y cómo las malnutriciones o desequilibrios dietéticos pueden afectar la función digestiva.\n\nAl explorar estos temas, te prepararás no solo para tus clases, sino también para entender mejor el papel complejo del sistema digestivo en el cuerpo humano y su interconexión con otros sistemas.

Based on the basic knowledge you likely obtained in high school and considering the medical context, you might want to dive deeper into the following topics before your meeting: \n\n1. Function and structure of the digestive system: Explore how each organ works and contributes to the digestive process. This could include the role of the anal sphincter, gastric movement, nutrient absorption in the small intestine, pancreatic enzyme secretion, bile production in the liver, and the function of the gallbladder.\n\n2. Digestive system conditions and diseases: Study common digestive diseases and conditions, such as gastrointestinal ulcers, inflammatory bowel diseases (IBD), gastric cancer, Crohn's disease, and irritable bowel syndrome. Understand how these conditions are diagnosed and treated.\n\n3. Molecular biology and biochemistry: Learn more about the chemical basis of digestive processes, such as the breakdown of carbohydrates, proteins, and fats, the role of specific enzymes, and the metabolic pathways involved in digestion.\n\n4. Medical physiology: Examine physiological processes like gastrointestinal secretion, hormonal regulation that controls feeding patterns, and the digestive system's function in response to nutrition and digestion.\n 5. Evolution and biological adaptation: Consider how the digestive system has adapted to different types of diets and how this has influenced human evolution. You can also research how other species with distinct digestive systems have adapted to their environments.\n\n6. Forensic science and medical history: Examine how the study of the digestive system is used in autopsies and how its understanding has evolved throughout history. This can help you grasp the differences between modern anatomy and biology and those from historical periods.\n\n7. Clinical practice and patient care skills: Learn about medical procedures related to digestion, such as endoscopies, abdominal scans, and antibiotic therapy for bacterial infections. You should also familiarize yourself with medical communication techniques and ethical considerations for patient care.\n\n8. Digestive surgery and endocrinology: Understand the surgeries performed to treat digestive conditions and how the digestive system interacts with the endocrine system, particularly in the context of gastroenterology.\n 9. Nutritional science and healthy eating: Study the importance of a balanced diet and how malnutrition or dietary imbalances can affect digestive function.\n\nBy exploring these topics, you will be prepared not only for your classes but also to better understand the complex role of the digestive system in the human body and its connections to other systems.

**Reference Answer**

Absolutamente, prepararse para discutir sus aspiraciones educativas con un profesor en un programa médico requiere un análisis más profundo del tema. Dado que la imagen se refiere al sistema digestivo humano, esto es lo que podría querer investigar y examinar con más detalle:\n\n1. **Fisiología y Función**: Comprenda cómo funciona cada parte del sistema digestivo, no solo anatómicamente sino a nivel celular y molecular. Explore temas como la digestión enzimática en el estómago y los intestinos, la absorción de nutrientes en el intestino delgado y el papel del microbioma en el intestino grueso.\n\n2. **Fisiopatología**: Investigue enfermedades y trastornos comunes que afectan al sistema digestivo, como el reflujo ácido, la enfermedad inflamatoria intestinal, la cirrosis hepática y los trastornos pancreáticos. Comprender cómo estas enfermedades alteran la fisiología normal puede ser crucial.\n\n3. **Correlaciones Clínicas**: Investigue cómo se presentan los síntomas en varias enfermedades digestivas y qué métodos de diagnóstico se utilizan para identificarlas. Esto podría incluir aprender sobre la endoscopia, colonoscopia, técnicas de imagen y pruebas de laboratorio.\n\n4. **Nutrición**: Dado que el sistema digestivo es integral para la nutrición, profundice en cómo se digieren y procesan los diferentes nutrientes. Puede querer entender el impacto de la dieta en la salud digestiva y cómo cambian las necesidades nutricionales en estados de enfermedad.\n\n5. **Farmacología**: Investigue cómo afectan varios medicamentos al sistema digestivo, incluidos aquellos utilizados para tratar trastornos digestivos. Esto incluye comprender los mecanismos de acción, efectos secundarios y el uso terapéutico de estos medicamentos.\n\n6. **Intervenciones Quirúrgicas**: Familiarícese con los procedimientos quirúrgicos relacionados con el sistema digestivo, como apendicectomías, cirugías bariátricas o trasplantes de hígado, y cuándo son necesarias estas intervenciones.\n\n7. **Investigación y Avances**: Manténgase actualizado sobre la última investigación en gastroenterología, incluidos nuevos tratamientos, avances en la investigación del microbioma intestinal y tecnologías emergentes en diagnósticos de salud digestiva.\n\n8. **Sistemas Interconectados**: Explore cómo el sistema digestivo interactúa con otros sistemas del cuerpo, como el sistema nervioso (el eje intestino-cerebro) y el sistema inmunológico.\n\nAl examinar estos temas, podrá tener una discusión más informada y detallada con su profesor sobre sus objetivos de aprendizaje en su programa médico.

Absolutely, preparing to discuss your educational aspirations with a professor in a medical program requires a more in-depth analysis of the subject. Since the image refers to the human digestive system, here is what you might want to research and examine further:\n\n1. Physiology and Function: Understand how each part of the digestive system works, not only anatomically but also at the cellular and molecular levels. Explore topics like enzymatic digestion in the stomach and intestines, nutrient absorption in the small intestine, and the role of the microbiome in the large intestine.\n\n2. Pathophysiology: Investigate common diseases and disorders affecting the digestive system, such as acid reflux, inflammatory bowel disease, liver cirrhosis, and pancreatic disorders. Understanding how these diseases disrupt normal physiology can be crucial.\n\n3. Clinical Correlations: Research how symptoms present in various digestive diseases and what diagnostic methods are used to identify them. This could include learning about endoscopy, colonoscopy, imaging techniques, and laboratory tests.\n\n4. Nutrition: Since the digestive system is integral to nutrition, dive into how different nutrients are digested and processed. You might want to understand the impact of diet on digestive health and how nutritional needs change in states of illness.\n\n5. Pharmacology: Research how various medications affect the digestive system, including those used to treat digestive disorders. This includes understanding the mechanisms of action, side effects, and therapeutic uses of these drugs.\n\n6. Surgical Interventions: Familiarize yourself with surgical procedures related to the digestive system, such as appendectomies, bariatric surgery, or liver transplants, and when these interventions are necessary.\n\n7. Research and Advances: Stay updated on the latest research in gastroenterology, including new treatments, advances in gut microbiome research, and emerging technologies in digestive health diagnostics.\n\n8. Interconnected Systems: Explore how the digestive system interacts with other body systems, such as the nervous system (the gut-brain axis) and the immune system.\n\nBy examining these topics, you will be able to have a more informed and detailed discussion with your professor about your learning objectives in the medical program.

**Feedback from GPT-4o**

The response attempts to provide a comprehensive preparatory plan by highlighting various topics related to the digestive system. It covers multiple aspects such as the function and structure of the digestive system, common diseases, and certain diagnostic procedures, as well as practical clinical skills and nutrition. However, some key areas necessary for a higher score in the rubric are either vaguely covered or missing. There is no explicit mention of pharmacology, which is an essential part of understanding how medications interact with the digestive system. Additionally, the discussion on interconnected systems, which can include the neurological and immune system interactions with the digestive system, is somewhat lacking in depth. While several areas are addressed, the response does not fully meet the comprehensive criteria for the top score. Overall, the breadth of topics is good, but the depth and inclusion of all necessary topics as per the rubric are limited.

[RESULT] 4

Figure 13: An xChat example from Spanish subset and our model PANGEA's response. The English part of the instruction, response, and reference answer is additionally added only on the figure to help to understand and was not given nor generated by PANGEA.

**Score Rubric**

**Does the response effectively describe the elements that contribute to the mood of the portrait and their interplay to create an emotional experience?**

Score 1: The response identifies basic elements of the portrait but does not relate them to the mood, for example, it might mention the subject's gaze but not how it conveys introspection.
Score 2: The response recognizes one or two elements that contribute to the mood but does not integrate them into a cohesive analysis, for example, noting the lighting or color but not how they work together to evoke calmness.
Score 3: The response describes several elements such as gaze, lighting, and color, and begins to relate them to the mood but lacks detail on their interactive effect, for example, it might suggest tranquility but not delve into the emotional depth created by these elements.
Score 4: The response provides a detailed explanation of the elements and their contribution to the mood, but may not fully capture the holistic emotional experience, for example, it describes the serene atmosphere but not the viewer's personal engagement with the portrait.
Score 5: The response offers a comprehensive explanation of the elements and their synergy, thoroughly describing how they interact to create a holistic emotional experience, reflecting a deep understanding of the portrait's figurative language.

**Instruction**

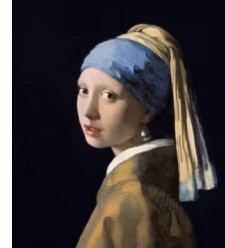

在观察这幅肖像时，哪些特定元素导致了肖像中氛围，它们如何相互作用以创造整体的情感体验？

When observing this portrait, what specific elements contribute to the atmosphere of the portrait, and how do they interact to create an overall emotional experience?

**Response from Pangea-7B**

肖像中营造氛围的特定元素包括女人凝视、她的服装以及光影的运用。她直接且略带怀疑的目光营造出一种亲密而引人深思的感觉，邀请观众进行对话。她的服装，以土色为主，传达出低调优雅和谦逊，符合当时的文化规范。光影的微妙运用，通过对比和柔和过渡，增加了画面的深度和立体感，突出了她的面部特征，特别是嘴唇和项链的光线，与背景的阴影形成鲜明对比。这些元素相互作用，创造了一种既沉思又吸引人的整体情感体验。

Specific elements that create the atmosphere in the portrait include the woman's gaze, her attire, and the use of light and shadow. Her direct yet slightly skeptical look fosters a sense of intimacy and contemplation, inviting the viewer to engage in dialogue. Her clothing, primarily in earthy tones, conveys understated elegance and humility, aligning with cultural norms of the time. The subtle manipulation of light and shadow, through contrast and soft transitions, adds depth and dimensionality to the image, highlighting her facial features—especially the light on her lips and necklace, which starkly contrasts with the background's shadows. These elements interact to create an overall emotional experience that is both contemplative and engaging.

**Reference Answer**

画中人物散发出一种宁静和沉思的氛围，主要通过人物温柔的目光考和柔和的光线传达出来。女人的眼睛微微低垂，目光向一侧，引导观众思考她心中可能在想什么，暗示着内省或安静。画面中的光线轻轻照亮她脸的一侧，投下微妙的阴影，营造出宁静的氛围。这种光影的互动为画作整体的平静和深思感做出了贡献。
此外，色彩的选择增强了这种氛围；蓝色头巾的冷色调和她衣服的温暖土色调实现了平衡和谐的效果，唤起一种平和的感觉。头巾优雅地披在肩上，增加了人物的尊严和反思的气质。孤立的耳环捕捉到一丝光芒，将观众的注意力吸引到她沉思的表情上，加深了肖像的情感共鸣。
此外，人物与深色背景之间的鲜明对比将观众的注意力集中在她身上，进一步增强了图像的亲密和冥想特质。构图的简洁，没有任何分散注意力的元素，允许更深刻的情感连接，邀请观众在个人层面上与人物互动。这种极简主义的方法鼓励观众考虑人物举止的细微差别，并推测她的内心生活，增强了图像的内省氛围。
总体而言，肖像的情感影响是构图、光线、色彩和细节巧妙结合的结果，共同创造出一种强烈而内敛的内心宁静和反思孤独的描绘。

The figure in the painting exudes an atmosphere of tranquility and introspection, primarily conveyed through her gentle gaze and soft lighting. The woman's eyes are slightly lowered, gazing to the side, guiding the viewer to ponder what might be on her mind, suggesting introspection or quietness. The light in the image softly illuminates one side of her face, casting subtle shadows that contribute to a serene atmosphere. This interplay of light and shadow enhances the overall calm and contemplative feeling of the artwork.\n\nAdditionally, the choice of colors reinforces this ambiance; the cool tones of the blue headscarf and the warm earthy tones of her clothing create a balanced and harmonious effect, evoking a sense of peace. The headscarf drapes elegantly over her shoulders, adding dignity and a reflective quality to the figure. The isolated earring captures a glimmer of light, serving as an eye-catching detail that draws attention to her contemplative expression, deepening the emotional resonance of the portrait.\n\nFurthermore, the stark contrast between the figure and the dark background focuses the viewer's attention on her, further enhancing the image's intimate and meditative qualities. The simplicity of the composition, devoid of distracting elements, allows for a deeper emotional connection, inviting viewers to engage with the figure on a personal level. This minimalist approach encourages the audience to consider the nuances of the figure's demeanor and speculate about her inner life, amplifying the introspective atmosphere of the image.\n\nOverall, the emotional impact of the portrait results from a skillful combination of composition, lighting, color, and detail, collectively creating a strong yet subdued depiction of inner peace and reflective solitude.

**Feedback from GPT-4o**

The response effectively identifies several key elements of the portrait, such as the woman's gaze, her attire, and the use of light and shadow, contributing to the overall atmosphere. It adequately explains how each element contributes to the mood: the gaze creates intimacy and thoughtfulness, the attire suggests elegance and humility, and the interplay of light and shadow enhances depth and contrast. There is a notable attempt to explain their interactive effect by discussing how these elements work together to create a contemplative and engaging emotional experience. However, while the response delves into specifics such as the gaze inviting dialogue or the warm earth tones conveying norms, it slightly falls short of capturing the holistic emotional experience or the viewer's personal engagement with the portrait, as a Score 5 response would. The level of detail and the relation of elements are thorough but not as comprehensive as required for the highest score. [RESULT] 4

Figure 14: An xChat example from the Chinese subset and our model PANGEA's response. The English part of the instruction, response, and reference answer is additionally added only on the figure to help to understand and was not given nor generated by PANGEA.

**Score Rubric**

**Does this response show an understanding of the Cepheid variable star light curves and their astronomical significance?**

Score 1: The response does not recognize the graph as a Cepheid variable light curve nor its relevance in astronomy.
Score 2: The response identifies the graph as related to star brightness but fails to connect it to Cepheid variables or their importance in distance measurement.
Score 3: The response correctly identifies the graph as a Cepheid variable light curve but only partially explains its significance in astronomy.
Score 4: The response identifies the graph as a Cepheid variable light curve and explains its importance in distance measurement but does not mention Henrietta Swan Leavitt's contribution.
Score 5: The response accurately identifies the Cepheid variable light curve, explains the period-luminosity relationship, the concept of \"standard candles,\" and acknowledges Henrietta Swan Leavitt's work.

**Instruction**

这是我在天文学课上遇到的一张图表，我不知道该如何解释。你能解释一下它与什么相关以及它传达了什么信息吗？

This is a chart I encountered in my astronomy class, and I'm not sure how to interpret it. Can you explain what it relates to and what information it conveys?

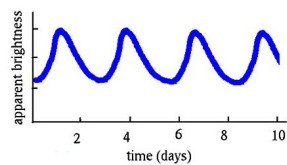

**Response from Pangea-7B**

根据提供的图片，该图表是一个简单的波形图，通常在数学或物理学中用于代表随时间变化的波动现象。在这个情境下，由于图中的横轴是"天"，我们可以推断这个图表可能代表着一个天文学过程。可能的解释是这幅图显示了一个天文事件或天体位置的变化，比如行星或卫星的轨道周期、星系的振荡，或者恒星亮度的变化。这个图表传达了这种现象随时间变化的情况；具体来说，波峰表示该事件处于最大强度，而波谷则表示其最弱的阶段。没有额外的背景信息，我们无法确定具体代表的是什么现象，但它明显是描述了一个随时间变化的天文或数学物理过程。

Based on the provided image, the chart is a simple waveform, commonly used in mathematics or physics to represent fluctuating phenomena over time. In this context, since the horizontal axis is labeled "days," we can infer that this chart likely represents an astronomical process. One possible explanation is that the chart depicts variations in an astronomical event or celestial body's position, such as the orbital period of a planet or satellite, the oscillation of a galaxy, or changes in stellar brightness.

This chart conveys how such phenomena change over time; specifically, the peaks represent the maximum intensity of the event, while the troughs indicate its weakest phase. Without additional background information, we cannot definitively identify what specific phenomenon is being represented, but it clearly describes a time-varying astronomical or mathematical-physical process.

**Reference Answer**

您提供的图表似乎显示了一个天文物体的表观亮度随时间的周期性振荡。这种模式是造父变星的特征。\n\n造父变星是一类由于外层脉动而表现出亮度规律性变化的恒星。图表显示了这些亮度在十天内的变化。造父变星的关键特征是它们的脉动周期与其本征光度直接相关，这种关系被称为周期-光度关系。这种关系在天体物理学中至关重要，因为它允许天文学家确定造父变星的距离，并进而确定它们所在的遥远星系的距离，从而有助于测量宇宙的尺度。\n\n图表旨在传达：\n- 造父变星亮度变化的周期性特征。\n- 亮度峰值之间的规律时间间隔（周期）。\n- 随时间变化的亮度变化幅度。\n\n这些周期性数据用于根据亨利爱塔·斯万·勒维特在20世纪初发现的周期-光度关系计算恒星的绝对星等，从而利用这些恒星作为测量天文距离的"标准烛光"来确定其与地球的距离。

The chart you provided appears to show the periodic oscillation of the apparent brightness of an astronomical object over time. This pattern is characteristic of Cepheid variables.\n\nCepheid variables are a class of stars that exhibit regular changes in brightness due to pulsations in their outer layers. The chart displays changes in brightness over a span of ten days. A key feature of Cepheid variables is that their pulsation period is directly related to their intrinsic luminosity, a relationship known as the period-luminosity relation. This relationship is crucial in astrophysics because it allows astronomers to determine the distance to Cepheid variables and, consequently, the distance to the distant galaxies they reside in, helping to measure the scale of the universe.\nThe chart is intended to convey:\n* The periodic nature of the brightness variations in Cepheid variables.\n* The regular time intervals (periods) between the brightness peaks.\n* The amplitude of brightness changes over time.\n\nThese periodic data are used to calculate the absolute magnitude of stars based on the period-luminosity relation discovered by Henrietta Swan Leavitt in the early 20th century, allowing these stars to serve as "standard candles" for measuring astronomical distances to determine their distance from Earth.

**Feedback from GPT-4o**

The response provided does not identify the graph as a Cepheid variable light curve and instead offers a general interpretation of a waveform graph used in mathematics or physics. It suggests the graph may represent an astronomical process, mentioning possibilities such as orbital periods or brightness variations, but fails to specifically recognize the graph as being related to Cepheid variables. Consequently, it does not address the significance of the period-luminosity relationship in distance measurement nor acknowledge Henrietta Swan Leavitt's contributions. Therefore, the response lacks key elements mentioned in the score rubric.\n\n[RESULT] 1

Figure 15: An xChat example from the Chinese subset and our model PANGEA's response. The English part of the instruction, response, and reference answer is additionally added only on the figure to help to understand and was not given nor generated by PANGEA.

**Score Rubric**

**Does this response show clear communication of the desired artistic elements for a masterpiece to be displayed in the living room?**

Score 1: The response does not specify any of the desired artistic elements for the artwork.
Score 2: The response vaguely mentions abstract elements but lacks detail on style, composition, color, emotional impact, and size.
Score 3: The response includes general descriptions of style and color but does not fully articulate the desired composition, emotional impact, or specific size requirements.
Score 4: The response clearly identifies the desired abstract expressionist style, rich color palette, and emotional impact, with some mention of size and composition preferences.
Score 5: The response provides a detailed and specific description of all desired elements including style, composition, color palette, emotional impact, and size requirements for the artwork.

**Instruction**

저는 제 거실에 전시된 이 작품과 유사한 미술 작품을 구매하고 싶습니다. 여러 걸작을 판매하는 사람에게 이 것을 어떻게 설명해야 할까요? 어떻게 자세히 설명할 수 있을지 잘 모르겠습니다.

I want to buy a piece of art similar to the one displayed in my living room. How should I explain this to someone who sells several masterpieces? I'm not sure how to describe it in detail.

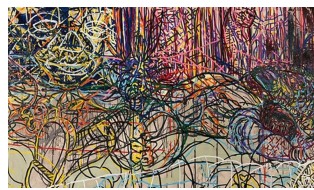

**Response from Pangea-7B**

이미지를 기반으로 이 작품은 현대 미술에서 종종 발견되는 추상적이고 개념적 특성을 가지고 있습니다. 추상화의 한 형태로 볼 수 있으며, 관찰자의 인식과 예술가의 표현을 교차시키는 방식으로 주제와 의미에 대한 보편적인 접근을 선호하는 탁 트인 해석적 표현과 일치합니다. 색상, 선 및 형태를 사용하여 예술가가 주제에 대한 자신의 감정적 또는 정서적 반응을 나타낼 수 있는 방식입니다. 특정한 장르나 주제가 없으므로 관찰자는 자신의 경험과 감정에 따라 자신의 해석과 연결할 수 있습니다.\n\n구매자에게 이러한 정보를 제공하면 예술가와 작업을 더 잘 이해할 수 있으므로 더욱 풍부하고 즐거운 경험을 할 수 있습니다. 또한 이러한 미술 작품은 종종 개인적인 연관성이나 감정적 호소력을 가질 수 있으며, 이것이 거실이라는 공간에서 예술이 어떻게 작용할 수 있는지에 대해 자세히 설명하는 것이 유용할 수 있습니다.

Based on the image, this piece possesses abstract and conceptual characteristics commonly found in contemporary art. It can be seen as a form of abstraction that aligns with open interpretative expressions, preferring universal approaches to themes and meanings that intersect the observer's perception and the artist's expression. The artist uses color, lines, and shapes to convey their emotional or emotional response to the subject. Since there isn't a specific genre or theme, the observer can connect their interpretation based on personal experiences and feelings.\n\nProviding this information to the seller will help you understand the artist and the work better, leading to a richer and more enjoyable experience. Additionally, such art often has personal connections or emotional appeal, making it useful to explain how the artwork might function within the living space.

**Reference Answer**

예술품 딜러나 갤러리 소유자와 이미지에 있는 것과 유사한 작품을 구입하려고 할 때, 관심 있는 스타일적 요소와 주제적 요소를 설명해야 합니다. 다음은 선호도를 설명하는 방법입니다:\n\n1. **스타일**: 에너제틱하고 제스처적인 붓질, 생동감 있는 색상, 즉흥적인 느낌이 특징인 추상 표현주의 스타일을 구현한 작품을 찾고 있다고 언급하세요.\n2. **구성**: 복잡하고 층이 있는 구성의 그림을 찾고 있으며, 아마도 비유적인 요소와 추상적인 형태가 얽혀 있는 작품을 원한다고 명시하세요.\n3. **색상 팔레트**: 디자인의 복잡성 내에서 높은 대조와 조화를 이룰 수 있는 풍부하고 다양한 색상 팔레트에 관심이 있다고 표현하세요.\n4. **감정적 영향**: 강한 감정적 또는 극적인 영향을 주는 작품에 끌린다고 전하며, 아마도 움직임이나 혼돈의 느낌이 있는 작품을 원한다고 전하세요.\n5. **크기**: 거실 공간에 맞고 원하는 대담한 표현을 할 수 있는 그림의 크기를 명시하세요.\n이러한 세부 정보를 제공함으로써, 예술품 판매자는 귀하의 필요를 더 잘 이해하고 거실과 개인 취향에 맞는 걸작으로 안내할 수 있습니다.

When trying to purchase a piece similar to the one in the image from an art dealer or gallery owner, you should describe both the stylistic and thematic elements that interest you. Here's how you can articulate your preferences:\n\n1. Style: Mention that you are looking for a work that embodies the energetic and gestural brushwork characteristic of abstract expressionism, with vibrant colors and an improvisational feel.\n\n2. Composition: Specify that you are seeking a painting with a complex and layered composition, likely featuring a mix of figurative elements and abstract forms.\n\n3. Color Palette: Express your interest in a rich and varied color palette that can create high contrast and harmony within the complexity of the design.\n\n4. Emotional Impact: Convey that you are drawn to pieces that have a strong emotional or dramatic impact, perhaps conveying a sense of movement or chaos.\n\n5. Size: Indicate the size of the painting that would fit well in your living space and allow for the bold expression you desire.\n\nBy providing these details, the art dealer will better understand your needs and guide you to masterpieces that align with your living room and personal taste.

**Feedback from GPT-4o**

The response focuses on abstract and conceptual characteristics of the artwork, mentioning its interpretive nature and emotional resonance. However, it does not provide specifics regarding style, composition, or color palette, nor does it address the emotional impact or size. While the response discusses interpretive qualities and the importance of emotional connection, it lacks clear identification or description of the artistic elements such as the abstract expressionist style, preferred composition, or ideal size, which are crucial for a practical purchase discussion with an art dealer. The information is generally abstract, without articulating detailed preferences that would assist a seller in understanding exactly what is needed for the living room setting.\n\n[RESULT] 2

Figure 16: An xChat example from the Korean subset and our model PANGEA's response. The English part of the instruction, response, and reference answer is additionally added only on the figure to help to understand and was not given nor generated by PANGEA.

# G  LANGUAGES IN PANGEAINS

Table 5 demonstrates the number of instances for each language that we include in PANGEAINS.

| Languages | en | multi | am | ar | bg | bn | cs | de | el | es |
|---|---|---|---|---|---|---|---|---|---|---|
| **Count** | 2554.6 | 4389.5 | 31.7 | 162.8 | 52.7 | 118.4 | 4.6 | 132.2 | 7.3 | 126.6 |
| **Percentage** (%) | 36.8 | 63.2 | 0.5 | 2.3 | 0.8 | 1.7 | 0.1 | 1.9 | 0.1 | 1.8 |
| **Languages** | **fa** | **fr** | **ga** | **hi** | **id** | **ig** | **it** | **iw** | **ja** | **jv** |
| **Count** | 8.4 | 193.8 | 34.7 | 163.5 | 178.4 | 26.1 | 68.8 | 168.3 | 403.3 | 35.2 |
| **Percentage** (%) | 0.1 | 2.8 | 0.5 | 2.4 | 2.6 | 0.4 | 1.0 | 2.4 | 5.8 | 0.5 |
| **Languages** | **ko** | **nl** | **mn** | **ms** | **no** | **pl** | **pt** | **ro** | **ru** | **si** |
| **Count** | 182.5 | 4.5 | 37.6 | 39.2 | 60.7 | 8.2 | 176.5 | 147.2 | 211.0 | 0.6 |
| **Percentage** (%) | 2.6 | 0.1 | 0.5 | 0.6 | 0.9 | 0.1 | 2.5 | 2.1 | 3.0 | 0.1 |
| **Languages** | **su** | **sw** | **ta** | **te** | **th** | **tr** | **uk** | **ur** | **vi** | **zh** |
| **Count** | 33.0 | 132.7 | 37.1 | 64.6 | 160.4 | 140.7 | 7.4 | 156.4 | 315.5 | 557.4 |
| **Percentage** (%) | 0.5 | 1.9 | 0.5 | 0.9 | 2.3 | 2.0 | 0.1 | 2.3 | 4.5 | 8.0 |

Table 5: Language distribution of PANGEAINS. We demonstrate the number of instances (in thousands) for each language, and the percentage of instances for each language in PANGEAINS

## H  TRAINING EXAMPLES

### H.1  MACHINE TRANSLATED INSTRUCTIONS

We include a few machine-translated training examples from PANGEAINS in Figure 17, Figure 18, Figure 19, Figure 20, Figure 21, Figure 22.

### H.2  MULTICULTURAL UNDERSTANDING INSTRUCTIONS

We include a few multicultural understanding instructions from PANGEAINS in Figure 23, Figure 24, Figure 25, Figure 26.

---

**Translated: Cambrian (Russian)**

**Human:**
Средняя заработная плата в час всех сотрудников компании составляет 20 долларов. Какова почасовая ставка Тома?
*(The average hourly wage for all employees in the company is $20. What is Tom's hourly rate?)*

**GPT:**
Невозможно определить почасовую ставку Тома на основе предоставленной информации. Средняя заработная плата в час всех сотрудников компании не обязательно отражает почасовую ставку любого отдельного сотрудника, поскольку индивидуальные оклады могут варьироваться в зависимости от ряда факторов, таких как должность, опыт, образование и производительность. Кроме того, средняя заработная плата в час всех сотрудников компании рассчитывается путем деления общей заработной платы всех сотрудников на количество часов, отработанных всеми сотрудниками, что не обязательно отражает количество часов, отработанных каким-либо отдельным сотрудником.
*(It is not possible to determine Tom's hourly wage based on the information provided. The average hourly wage for all employees of a company does not necessarily reflect the hourly wage of any individual employee, as individual salaries can vary based on a number of factors, such as job title, experience, education, and performance. Additionally, the average hourly wage for all employees of a company is calculated by dividing the total wages of all employees by the number of hours worked by all employees, which does not necessarily reflect the number of hours worked by any individual employee.)*

Figure 17: ***Translated Task***: An example from the Cambrian dataset where it discusses the concept of hourly wages based on a given prompt in Russian.

---

**Translated: ALLaVa-LAION (Turkish)**

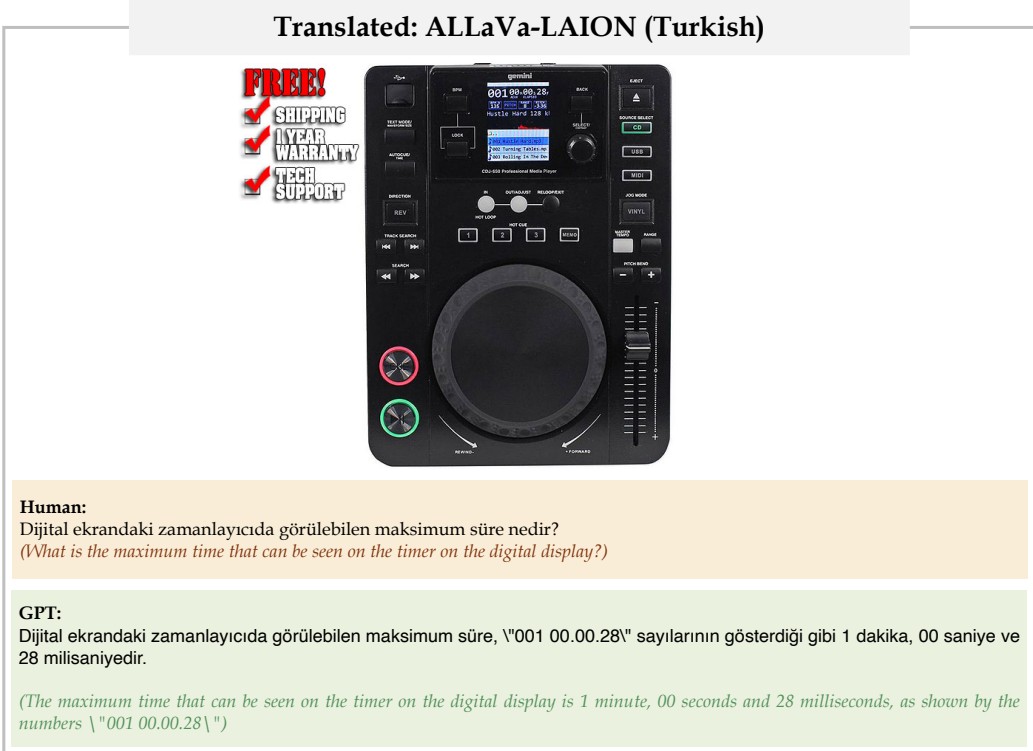

**Human:**
Dijital ekrandaki zamanlayıcıda görülebilen maksimum süre nedir?
*(What is the maximum time that can be seen on the timer on the digital display?)*

**GPT:**
Dijital ekrandaki zamanlayıcıda görülebilen maksimum süre, \"001 00.00.28\" sayılarının gösterdiği gibi 1 dakika, 00 saniye ve 28 milisaniyedir.

*(The maximum time that can be seen on the timer on the digital display is 1 minute, 00 seconds and 28 milliseconds, as shown by the numbers \"001 00.00.28\")*

Figure 18: ***Multimodal Translated Task***: An example from the ALLaVa-LAION dataset where the GPT model answers a prompt in Turkish regarding the maximum time displayed on a digital timer.

**Translated: GPT-4v (Thai)**

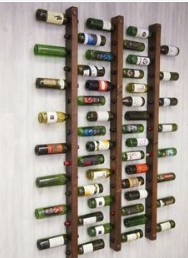

**Human:**
ภาพที่แสดง:
*(Image showing:)*

**GPT:**
ชั้นวางไวน์ติดผนังพร้อมขวดไวน์จำนวนมากเรียงรายกันในแนวตั้ง ชั้นวางทำจากแผ่นไม้แนวนอนหลายชั้น เจาะเป็นรูวงกลมสำหรับวางขวดไวน์ในแนวนอน โดยให้คอขวดหันออกด้านนอก มีขวดไวน์หลากหลายรูปแบบ แต่ละขวดมีฉลากและสีต่างกัน บ่งบอกถึงประเภทและยี่ห้อของไวน์ที่หลากหลาย ขวดต่างๆ ถูกจัดวางอย่างเป็นระเบียบและเต็มชั้นวาง พื้นหลังเป็นผนังสีเรียบ ซึ่งตัดกับสีเข้มของชั้นวางไม้และฉลากไวน์หลากสีสัน รูปแบบของภาพดูตรงไปตรงมาและดูเหมือนจะเป็นภาพถ่ายที่เน้นการใช้งานจริง เพื่อแสดงชั้นวางไวน์และขวดไวน์ภายในอย่างชัดเจนและเรียบง่าย

*(A wall-mounted wine rack with many wine bottles lined up vertically. The rack is made of several horizontal wooden planks, with circular holes cut out to hold the wine bottles horizontally, with the necks of the bottles facing outward. There are various types of wine bottles, each with different labels and colors, indicating the various types and brands of wine. The bottles are arranged neatly and fill the rack. The background is a plain wall, which contrasts with the dark colors of the wooden racks and the colorful wine labels. The layout of the image is straightforward and seems to be a practical photo, to clearly and simply show the wine rack and the wine bottles inside.)*

Figure 19: *Multimodal Translated Task*: An example from the ShareGPT-4v dataset where the model describes an image of a wine rack in Thai, detailing its structure and the arrangement of wine bottles.

**Translated: OpenHermes (Japanese)**

**Human:**
for ループを関数呼び出しに変換して、コードを最小限に抑えるのを手伝ってもらえますか?
*(Can you help me convert the for loop into a function call to minimize the code?)*

**GPT:**
承知しました! コードを最小限に抑えるには、`map` 関数を使用して同じ結果を得ることができます。次に例を示します。\n\n```javascript\nlet arr = [4, 7, 2, 9];\narr.map((elem) => console.log(elem * 2));\n```\n\nJavaScript の `map` 関数を使用すると、配列を反復処理し、指定された関数に従って各要素を変換できます。この場合、各要素に 2 を掛けてから、コンソールに出力しています。\n\n`map` を使用することで、従来の for ループが不要になり、より簡潔な方法で同じ結果を得ることができます。

*(Sure! To minimize code, you can use the `map` function to achieve the same result. For example:\n```javascript\nlet arr = [4, 7, 2, 9];\narr.map((elem) => console.log(elem * 2));\n```\n\nJavaScript's `map` function allows you to iterate through an array and transform each element according to a specified function. In this case, we multiply each element by 2 before printing it to the console.\n\nBy using `map`, we can eliminate the need for a traditional for loop and achieve the same result in a more concise way.)*

Figure 20: *Text-only Translated Task*: An example from the OpenHermes2.5 dataset translated into Japanese.

**Translated: NuminaMath (Spanish)**

**Human:**
Definir una terna ordenada $(A, B, C)$ de conjuntos como mínimamente intersecantes como se describe: $|A \cap B| = 1$, $|B \cap C| = 1$, $|C \cap A| = 1$, y $A \cap B \cap C = \emptyset$. Determinar el número de tales ternas ordenadas donde cada conjunto es un subconjunto de $\{1,2,3,4,5,6,7,8\}$. Cada conjunto debe incluir ahora exactamente un elemento que no se comparte con los otros conjuntos, y ningún conjunto puede estar vacío. Calcular el resto cuando el número de estas ternas se divide entre $1000$.

*(Define an ordered triple $(A, B, C)$ of sets to be minimally intersecting as follows: $|A \cap B| = 1$, $|B \cap C| = 1$, $|C \cap A| = 1$, and $A \cap B \cap C = \emptyset$. Determine the number of such ordered triples where each set is a subset of $\{1,2,3,4,5,6,7,8\}$. Each set must now include exactly one element that is not shared with the other sets, and no set can be empty. Compute the remainder when the number of such triples is divided by $1000$.)*

**GPT:**
1. Elegir $x$, $y$, $z$ como identificadores únicos para las intersecciones por pares ($A\cap B$, $B\cap C$, y $C\cap A$). Hay $8 \cdot 7 \cdot 6 = 336$ maneras de elegir valores distintos para $x$, $y$, y $z$ ya que el conjunto $\{1,2,3,4,5,6,7,8\}$ contiene $8$ números.\n\n2. Cada conjunto ($A$, $B$, $C$) debe incluir exactamente un elemento único aparte de $x$, $y$, $z$. Esto significa seleccionar un elemento único de los $5$ números restantes para cada conjunto de manera que ningún elemento se reutilice: \n  - Elegir un elemento único para $A$ (no $x$ o $y$) de los $5$ elementos restantes: $5$ opciones.\n  - Elegir otro elemento único para $B$ (no $y$ o $z$) de los $4$ elementos restantes: $4$ opciones.\n  - Elegir otro elemento único para $C$ (no $x$ o $z$) de los $3$ elementos restantes: $3$ opciones.\n\n3. Los $2$ elementos restantes pueden ir a cualquier conjunto o a ninguno, por lo que hay $4$ opciones para cada uno de estos $2$ elementos (cualquiera de los tres conjuntos o ninguno): $4^2 = 16$.\n\nCombinando estas opciones,\n\[ 336 \times 5 \times 4 \times 3 \times 16 = 107520 \]\nPor lo tanto, el resto al dividir entre $1000$ es $\boxed{520}$.\n   \n".

*(1. Choose $x$, $y$, $z$ as unique identifiers for the pairwise intersections ($A\cap B$, $B\cap C$, and $C\cap A$). There are $8 \cdot 7 \cdot 6 = 336$ ways to choose distinct values for $x$, $y$, and $z$ since the set $\{1,2,3,4,5,6,7,8\}$ contains $8$ numbers.\n\n2. Each set ($A$, $B$, $C$) must contain exactly one unique element other than $x$, $y$, $z$. This means selecting a unique element from the remaining $5$ numbers for each set such that no element is reused: \n- Choose a unique element for $A$ (not $x$ or $y$) from the remaining $5$ elements: $5$ options.\n- Choose another unique element for $B$ (not $y$ or $z$) from the remaining $4$ elements: $4$ options.\n- Choose another unique element for $C$ (not $x$ or $z$) from the remaining $3$ elements: $3$ options.\n\n3. The remaining $2$ elements can go into any set or none, so there are $4$ options for each of these $2$ elements (any of the three sets or none): $4^2 = 16$.\n\nCombining these options,\n\[ 336 \times 5 \times 4 \times 3 \times 16 = 107520 \]\nThus, the remainder when divided by $1000$ is $\boxed{520}$.\n \n")*

Figure 21: ***Text-only Translated Task***: An example from the NumininaMath dataset translated into Spanish.

## Translated: Code-Feedback (German)

**Human:**
Gegeben ein Array von Ganzzahlen, bestimmen Sie den Maximal- und Minimalwert des Arrays. Das Array kann doppelte Werte haben und kann eine beliebige Länge zwischen 1 und 10^6 haben. Außerdem kann das Array unsortiert sein.
*(Given an array of integers, determine the maximum and minimum values of the array. The array can have duplicate values and can have any length between 1 and 10^6. Additionally, the array can be unsorted.)*

**GPT:**
## Plan\n1. Initialisieren Sie die Variablen min_val und max_val mit dem ersten Element des Arrays.\n2. Iterieren Sie durch das Array und aktualisieren Sie min_val und max_val nach Bedarf.\n3. Geben Sie min_val und max_val zurück
*(## Plan\n1. Initialize the variables min_val and max_val with the first element of the array.\n2. Iterate through the array and update min_val and max_val as needed.\n3. Return min_val and max_val)*

**Human**:
Berechnen Sie den Umfang und die Fläche eines Dreiecks mit der gegebenen Seitenlänge. Seitenlängen: 6, 8, 10. Das Programm sollte auch prüfen, ob die angegebenen Seiten ein gültiges Dreieck bilden. Außerdem sollte das Programm Fälle behandeln, in denen die Seiten kein gültiges Dreieck bilden, indem es entsprechende Fehlermeldungen ausgibt

*(Calculate the perimeter and area of a triangle with the given side length. Side lengths: 6, 8, 10. The program should also check whether the given sides form a valid triangle. In addition, the program should handle cases where the sides do not form a valid triangle by giving appropriate error messages)*

**GPT:**
Um den Umfang und die Fläche eines Dreiecks mit den gegebenen Seitenlängen (6, 8, 10) zu berechnen, können wir die Formel von Heron verwenden. Bevor wir jedoch mit der Berechnung fortfahren, müssen wir prüfen, ob die gegebenen Seiten ein gültiges Dreieck bilden.\n\nUm festzustellen, ob die gegebenen Seiten ein gültiges Dreieck bilden, können wir den Satz der Dreiecksungleichung anwenden, der besagt, dass die Summe zweier Seiten eines Dreiecks größer sein muss als die dritte Seite.\n\nSchreiben wir ein Programm in Python, das den Umfang und die Fläche eines Dreiecks mit den gegebenen Seitenlängen berechnet und prüft, ob die Seiten ein gültiges Dreieck bilden:\n\n```python\nimport math\n\ndef calculate_perimeter(a, b, c):\n    return a + b + c\n\ndef calculate_area(a, b, c):\n    s = (a + b + c) / 2  # Berechnen Sie den halben Umfang\n    area = math.sqrt(s * (s - a) * (s - b) * (s - c))  # Verwenden Sie die Formel von Heron\n    return area\n\ndef is_valid_triangle(a, b, c):\n    if a + b > c and b + c > a and a + c > b:\n        return True\n    return False\n\n# Geben Sie die Seitenlängen des Dreiecks ein\na = 6\nb = 8\nc = 10\n\n# Prüfen Sie, ob die Seiten ein gültiges Dreieck bilden\nif is_valid_triangle(a, b, c):\n    perimeter = calculate_perimeter(a, b, c)\n    area = calculate_area(a, b, c)\n\n    print(\"Umfang:\", perimeter)\n    print(\"Fläche:\", area)\nelse:\n    print(\"Die angegebenen Seiten bilden kein gültiges Dreieck.\")\n```\n\nWenn Sie dieses Programm ausführen, wird Folgendes ausgegeben:\n\n```\nUmfang: 24\nFläche: 24.0\n```\n\nDa die gegebenen Seitenlängen (6, 8, 10) den Satz der Dreiecksungleichung erfüllen, können wir den Umfang und die Fläche des Dreiecks berechnen. Der Umfang beträgt 24 Einheiten und die Fläche 24 Flächeneinheiten."
*(To calculate the perimeter and area of a triangle with given side lengths (6, 8, 10), we can use Heron's formula. However, before we proceed with the calculation, we need to check if the given sides form a valid triangle.\n\nTo determine if the given sides form a valid triangle, we can apply the triangle inequality theorem, which states that the sum of two sides of a triangle must be greater than the third side.\n\nLet's write a program in Python that calculates the perimeter and area of a triangle with the given side lengths and checks if the sides form a valid triangle:\n\n```python\nimport math\n\ndef calculate_perimeter(a, b, c):\n return a + b + c\n\ndef calculate_area(a, b, c):\n s = (a + b + c) / 2 # Calculate half the perimeter\n area = math.sqrt(s * (s - a) * (s - b) * (s - c)) # Use Heron's formula\n return area\n\ndef is_valid_triangle(a, b, c):\n if a + b > c and b + c > a and a + c > b:\n return True\n return False\n\n# Enter the side lengths of the triangle\na = 6\nb = 8\nc = 10\n\n# Check if the sides form a valid triangle\nif is_valid_triangle(a, b, c):\n perimeter = calculate_perimeter(a, b, c)\n area = calculate_area(a, b, c)\n\n print(\"Perimeter:\", perimeter)\n print(\"Area:\", area)\nelse:\n print(\"The given sides do not form a valid triangle.\")\n```\n\nWhen you run this program, the following is output:\n\n```\nPerimeter: 24\nArea: 24.0\n```\n\nSince the given If the side lengths (6, 8, 10) satisfy the triangle inequality, we can calculate the perimeter and area of the triangle. The perimeter is 24 units and the area is 24 units.")*

Figure 22: ***Text-only Translated Task***: An example from the Code-Feedback dataset translated into German.

---

**Multicultural Understanding: LAION-Culture (Portugese)**

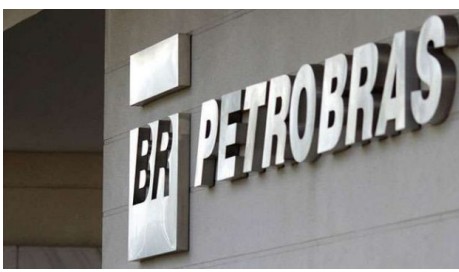

**Human:**
Descreva os elementos visuais que compõem o logotipo da Petrobras na imagem.
*(Describe the visual elements that make up the Petrobras logo in the image.)*

**GPT:**
O logotipo da Petrobras na imagem é composto por letras metálicas prateadas e brilhantes, fixadas em uma parede de concreto cinza. A palavra 'PETROBRAS' está escrita em letras maiúsculas e maiores, enquanto a sigla 'BR' aparece em letras menores, em um quadrado separado. As letras são tridimensionais, o que lhes confere uma aparência moderna e destacada. O contraste entre o brilho das letras e a superfície opaca da parede cria um efeito visual interessante, destacando a presença da marca.
*(The Petrobras logo in the image is made up of shiny silver metallic letters, fixed to a gray concrete wall. The word 'PETROBRAS' is written in larger capital letters, while the acronym 'BR' appears in smaller letters, in a separate square. The letters are three-dimensional, giving them a modern, standout look. The contrast between the brightness of the letters and the opaque surface of the wall creates an interesting visual effect, highlighting the brand's presence.)*

**Human:**
Analise como o design do logotipo da Petrobras reflete a identidade e os valores da empresa.
*(Analyze how the Petrobras logo design reflects the company's identity and values.)*

**GPT:**
O design do logotipo da Petrobras reflete a identidade e os valores da empresa de várias maneiras. Primeiramente, as letras metálicas prateadas e brilhantes sugerem inovação e modernidade, valores essenciais para uma empresa de petróleo e gás natural. A tridimensionalidade das letras confere uma sensação de solidez e estabilidade, reforçando a ideia de que a Petrobras é uma empresa forte e confiável. Além disso, o contraste entre o brilho das letras e a parede de concreto cinza pode simbolizar a integração da empresa com o ambiente urbano e industrial. A separação da sigla 'BR' em um quadrado distinto destaca a importância da identidade brasileira para a empresa, reforçando seu papel como uma entidade nacional. Portanto, o design do logotipo não só representa a identidade visual da Petrobras, mas também comunica seus valores e sua conexão com o Brasil.
*(The Petrobras logo design reflects the company's identity and values in many ways. Firstly, the shiny silver metallic letters suggest innovation and modernity, essential values for an oil and natural gas company. The three-dimensionality of the letters gives a feeling of solidity and stability, reinforcing the idea that Petrobras is a strong and reliable company. Furthermore, the contrast between the brightness of the letters and the gray concrete wall can symbolize the company's integration with the urban and industrial environment. The separation of the acronym 'BR' into a distinct square highlights the importance of Brazilian identity for the company, reinforcing its role as a national entity. Therefore, the logo design not only represents Petrobras' visual identity, but also communicates its values and its connection with Brazil.)*

Figure 23: ***Multicultural Understanding***: An example from the LAION-Culture dataset where the GPT model describes and analyzes the Petrobras logo in Portuguese.

## Multicultural Understanding: LAION-Culture (Bulgarian)

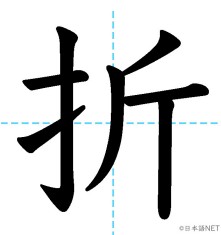

**Human:**
Какво е значението на японския йероглиф на снимката?
*(What is the meaning of the Japanese character in the picture)*

**GPT:**
Йероглифът на снимката представлява символа \"折\", който се чете \"ори\" на японски. Този йероглиф се използва за означаване на думата \"ори\", която може да има различни значения в зависимост от контекста. В случая, йероглифът е показан с черно мастило върху бяла хартия, със сини пунктирани линии, които разделят символа на секции, за да се покаже правилната последователност на ударите при писането. Това е важен елемент от японската калиграфия, която се нарича \"шодо\".
*(The hieroglyph in the photo is the character \"折\" which is read \"ori\" in Japanese. This hieroglyph is used to denote the word \"ori\", which can have different meanings depending on the context. In this case, the hieroglyph is shown in black ink on white paper, with blue dotted lines dividing the symbol into sections to show the correct sequence of strokes when writing. This is an important element of Japanese calligraphy, which is called ``shodo''.)*

**Human**:
Какво значение имат сините пунктирани линии в японската калиграфия?.
(What is the meaning of blue dotted lines in Japanese calligraphy?)

**GPT:**
Сините пунктирани линии в японската калиграфия имат специфично значение и предназначение. Те са използвани за обозначаване на последователността и посоката на движенията на четката при писане на йероглифи. В случая, тези линии разделят йероглифа \"折\" на секции, за да се улесни процесът на писане и да се осигури, че всеки удар ще бъде извършен правилно. Това е част от традицията на японската калиграфия, която се нарича \"шодо\", и е съществена за изучаването и практикуването на този изкуство. Сините линии не само помагат за правилното писане, но и допринасят за естетическото качество на финалния произведение
(The blue dotted lines in Japanese calligraphy have a specific meaning and purpose. They were used to indicate the sequence and direction of brush movements when writing hieroglyphs. In this case, these lines divide the character \"折\" into sections to facilitate the writing process and ensure that each stroke will be executed correctly.  It is part of the tradition of Japanese calligraphy, which is called ``shōdo,'' and is essential to the study and practice of this art. Blue lines not only help with correct writing, but also contribute to the aesthetic quality of the final work)

Figure 24: *Multicultural Understanding*: Example from the LAION-Culture dataset where the model interprets the meaning of a Japanese character and its representation in calligraphy, described in Bulgarian.

## Multicultural Understanding: LAION-Culture (Hebrew)

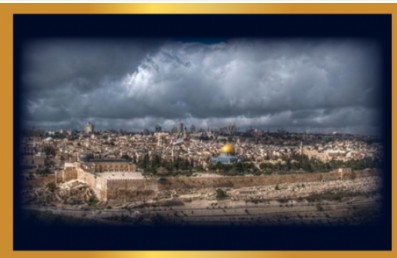

**Human:**
Analyze the cultural significance of the prominent landmarks visible in the image of Jerusalem. Answer it in Hebrew.

**GPT:**
התמונה מציגה שלושה מורכבים רב-תווים של ירושלים: חומות העיר העתיקה, כיפת הסלע המוזהבת, ועצמה העיר העתיקה. חומ
ות העיר העתיקה, שמקובלות כמורכב בנוי בין השנים 1535 ל-1538 על ידי הסולטן העות'מאני סלימאן המפיק, מסמלות את ההיס
טוריה העשירה של העיר ואת המשמעות התרבותית והדתית שלה כמקום של חשיבות רבה ביהדות, נוצרות ואסלאם. כיפת הסלע
המוזהבה, בנויה בשנת 691 לספירה, עומדת כסמל נוצץ של האסלאם ומייצגת את המשמעות הרוחנית העמוקה של המקום הזה ב
תקופה האיסלאמית. העיר העתיקה בעצמה, עם רחובותיה הצרים ובתיה האבן, מעיינת בין השאר במורשת התרבותית העשירה ש
ל ירושלים, בה מתמזגות השפות, הדתות והתרבויות בצורה ייחדית. הנוף הזה, עם העצים והרחובות החיים, מעין בחיי העיר ובמ
שמעות התרבותית וההיסטורית של ירושלים כמרכז רוחני ותיירותי.
*(The image shows three multi-character complexes of Jerusalem: the walls of the Old City, the golden Dome of the Rock, and the Old City itself. The Old City Walls, which are accepted as a complex built between 1535 and 1538 by the Ottoman Sultan Suleiman the Magnificent, symbolize the city's rich history and its cultural and religious significance as a place of great importance in Judaism, Christianity and Islam. The golden Dome of the Rock, built in 691 AD, stands as a shining symbol of Islam and represents the deep spiritual significance of this place in the Islamic period. The old city itself, with its narrow streets and stone houses, reflects, among other things, the rich cultural heritage of Jerusalem, where the languages, religions and cultures merge in a unique way. This landscape, with the trees and living streets, is a source of the life of the city and the cultural and historical significance of Jerusalem as a spiritual and touristic center.")*

**Human:**
Imagine you are planning a guided tour of Jerusalem based on the landmarks visible in the image. Outline the key points you would cover in your tour. Answer it in Hebrew.

**GPT:**
בהסכמה לסיור מודרך בירושלים בהתבסס על המקומות המובילים שמופיעים בתמונה, הייתי מתחיל בחומות העיר העתיקה, שמ
סמלות את הגבול ההיסטורי של העיר. הייתי מספר על ההיסטוריה של החומות, על סלימאן המפיק ועל המשמעות התרבותית של
הן. משם, הייתי מזיז את הקבוצה לכיפת הסלע המוזהבת, והייתי מספר על הבנייה שלה בשנת 691 ועל המשמעות הרוחנית של ה
מקום באסלאם. הייתי מדגיש את הארכיטקטורה הייחודית והמשמעות הסמלית של הכיפה. לבסוף, הייתי מוביל את הקבוצה
לרחובות הצרים של העיר העתיקה, והייתי מספר על החיי היומיומי, על המזג התרבותי העשיר ועל המשמעות התיירותית של האז
ור. הייתי מדגיש את השילוב הייחדי של שפות, דתות ותרבויות בתוך העיר, ואיך זה משפיע על החוויה התיירותית. הייתי מסיים ע
ם הצגת הנוף המרהיב והשלווה של העיר, והמשמעות הרוחנית וההיסטורית של ירושלים כמרכז חשוב בעולם."
*(Agreeing to a guided tour of Jerusalem based on the leading places that appear in the picture, I would start with the walls of the old city, which symbolize the historical border of the city. I would tell about the history of the walls, about Suleiman the producer and their cultural significance. From there, I would move the group to the golden Dome of the Rock, and I would tell about its construction in 691 and the spiritual significance of the place in Islam. I would emphasize the unique architecture and the symbolic meaning of the dome. Finally, I would lead the group to the narrow streets of the old city, and I would tell about the daily life, the rich cultural temperament and the tourist significance of the area. I would emphasize the unique combination of languages, religions and cultures within the city, and how this affects the tourist experience. I would conclude by presenting the spectacular and peaceful view of the city, and the spiritual and historical significance of Jerusalem as an important center in the world.")*

Figure 25: *Multicultural Understanding*: LAION-Culture sample in Hebrew.

## Multicultural Understanding: LAION-Culture (Tamil)

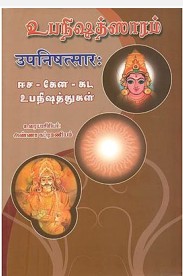

**Human:**

வெளிப்பாட்டு படத்தில் உள்ள புத்தகத்தின் கவர்ச்சியான கவர்ச்சிகளை விவரிக்கவும்
*(Describe the charms of the book in expressive imagery.)*

**GPT:**

புத்தகத்தின் கவர்ச்சிகள் இந்திய பாரம்பரியத்தை, கலையை, வரலாற்றை ஒருங்கிணைக்கும். புத்தகத்தின் மேற்பரப்பில் 'உபநிஷத் சாரம்: Upanishad Saram' என்ற தலைப்பு தன்னார்வலியாக எழுதப்பட்டுள்ளது. மையப்பகுதியில் ஒரு சூரிய ஒளியை சுற்றியுள்ள வட்டங்கள் காட்டும் சிற்பம் உள்ளது, இது உபநிஷத்களில் கொண்டிருக்கும் ஆன்மீக ஒளியுதர்வு அல்லது ஞானத்தின் ஆதாரமாக இருக்கலாம். வலதுபுறத்தில் ஒரு ஆள் வரைபடம் உள்ளது, இது உபநிஷத்களில் கொண்டிருக்கும் ஞானம் மற்றும் ஆன்மீக அறிவுகளைக் குறிப்பிடுகிறது. இந்த புத்தகம் உபநிஷத்களின் ஆன்மீக அறிவு மற்றும் இறைவன் அல்லது இறையியற்றின் ஆதாரத்தைப் பிரதிபலிக்கும்.

*(The attractions of the book combine Indian heritage, art and history. The title 'Upanishad Saram: Upanishad Saram' is voluntarily written on the surface of the book. In the center there is a sculpture showing circles surrounding a sun, which may be a source of spiritual illumination or wisdom in the Upanishads. On the right is a human diagram, representing the wisdom and spiritual knowledge contained in the Upanishads. This book reflects the spiritual knowledge of the Upanishads and the source of the Lord or God.)*

**Human**:

புத்தகத்தின் கவர்ச்சியான கவர்ச்சிகளைப் பெற்றோர் ஒரு குறுகிய கதை உருவாக்கவும்..
(Create a short story for parents of the book's attractive charms.)

**GPT**:

ஒரு ஆழ்ந்த காலம் முன்னால், ஒரு சிறு கிராமத்தில், ஒரு பெரிய ஆசாரி ஒரு உபநிஷத் சாரம் என்ற புத்தகத்தை எடுத்துக்கொண்டார். அந்த புத்தகத்தில் உபநிஷத்களின் ஆன்மீக அறிவு மற்றும் இறைவன் அல்லது இறையியற்றின் ஆதாரம் காட்டப்பட்டுள்ளது. ஆசாரி இந்த புத்தகத்தை எடுத்துக்கொண்டு கிராமத்தின் மக்களுக்கு அறிவு மற்றும் ஆன்மீக அறிவு பகிர்ந்தார். கிராமவாசிகள் இந்த அறிவுடன் ஆன்மீக ஒளியுதர்வை அனுபவிக்க ஆரம்பித்தனர்.

(A long time ago, in a small village, a great priest picked up a book called an Upanishad Saram. In that book the spiritual knowledge of the Upanishads and the source of the Lord or God is shown. Asari took this book and imparted knowledge and spiritual knowledge to the people of the village. The villagers began to experience spiritual enlightenment with this knowledge.)

Figure 26: *Multicultural Understanding*: LAION-Culture sample in Tamil.

# I    BREAKDOWN RESULTS OF DIFFERENT LANGUAGES ON PANGEABENCH

## I.1    XCHAT

We show the performance of different models on the xChat benchmark in Table 6.

| Models | English | Multi | Spanish | Hindi | Indonesian | Japanese | Korean | Chinese |
|---|---|---|---|---|---|---|---|---|
| Gemini-1.5-Pro | 71.0 | 65.6 | 66.0 | 62.0 | 65.5 | 68.0 | 66.5 | 65.5 |
| GPT4o | 67.0 | 65.1 | 66.0 | 64.0 | 65.0 | 66.5 | 67.5 | 61.5 |
| Llava-1.5-7B | 22.5 | 16.7 | 22.5 | 3.5 | 18.0 | 23.0 | 12.0 | 21.0 |
| Llava-Next-7B | 40.5 | 20.4 | 33.0 | 1.5 | 19.0 | 25.0 | 15.0 | 29.0 |
| Phi-3.5-Vision | 38.5 | 21.1 | 37.0 | 11.5 | 10.5 | 31.0 | 12.5 | 24.0 |
| Cambrian-8B | 27.5 | 15.8 | 22.5 | 4.0 | 20.0 | 20.0 | 10.5 | 18.0 |
| Llava-OV-7B | 51.0 | 33.1 | 45.5 | 6.5 | 42.0 | 36.5 | 26.0 | 42.0 |
| Molmo-7B-D | 49.5 | 34.7 | 45.0 | 19.5 | 36.5 | 36.0 | 35.0 | 46.0 |
| Llama3.2-11B | 49.0 | 31.3 | 42.5 | 19.5 | 45.0 | 26.0 | 21.0 | 43.0 |
| PaliGemma-3B | 6.0 | 3.8 | 4.5 | 0.5 | 6.5 | 6.5 | 2.0 | 3.0 |
| PALO-7B | 27.0 | 16.2 | 23.0 | 3.0 | 19.0 | 20.0 | 13.5 | 18.5 |
| mBLIP mT0-XL | 2.5 | 0.5 | 0.0 | 0.0 | 0.5 | 2.0 | 0.5 | 0.0 |
| mBLIP BLOOMZ-7B | 4.0 | 1.7 | 2.0 | 2.5 | 2.5 | 0.0 | 0.0 | 3.0 |
| PANGEA-7B (Ours) | 46.0 | 35.8 | 43.5 | 23.5 | 34.5 | 39.0 | 33.5 | 40.5 |

Table 6: Comparison of models on the xChat dataset across different languages.

## I.2    MULTILINGUAL LLAVABENCH

We show the performance of different models on the Multilingual LLaVABench benchmark in Table 7.

| Models | English | Multi | Arabic | Bengali | Chinese | French | Hindi | Japanese | Russian | Spanish | Urdu |
|---|---|---|---|---|---|---|---|---|---|---|---|
| Gemini-1.5-Pro | 103.4 | 106.6 | 112.9 | 117.1 | 104.1 | 115.5 | 106.2 | 118.1 | 95.7 | 88.2 | 101.6 |
| GPT4o | 104.6 | 100.4 | 98.3 | 111.9 | 96.5 | 101.1 | 99.7 | 104.0 | 88.5 | 100.9 | 102.5 |
| Llava-1.5-7B | 66.1 | 40.8 | 26.4 | 11.9 | 50.7 | 63.8 | 23.2 | 70.0 | 46.5 | 59.2 | 15.4 |
| Llava-Next-7B | 78.9 | 50.7 | 24.9 | 11.2 | 72.8 | 91.4 | 18.0 | 70.1 | 71.8 | 82.9 | 13.4 |
| Phi-3.5-Vision | 70.8 | 58.0 | 50.1 | 35.1 | 69.2 | 86.0 | 35.9 | 63.0 | 67.6 | 75.6 | 39.3 |
| Cambrian-8B | 78.4 | 61.8 | 54.1 | 35.4 | 80.9 | 87.3 | 44.2 | 64.4 | 76.4 | 90.3 | 23.3 |
| Llava-OV-7B | 89.7 | 55.3 | 45.5 | 33.8 | 90.0 | 89.4 | 35.3 | 70.3 | 44.7 | 75.5 | 13.3 |
| Molmo-7B-D | 95.9 | 13.8 | 10.1 | 4.2 | 0.3 | 59.6 | 5.5 | 6.0 | 8.7 | 29.5 | 0.0 |
| Llama3.2-11B | 93.9 | 58.2 | 39.4 | 48.1 | 47.2 | 85.6 | 67.8 | 53.7 | 68.5 | 77.8 | 35.3 |
| PaliGemma-3B | 32.1 | 31.9 | 37.3 | 38.2 | 29.1 | 30.0 | 35.8 | 33.4 | 26.1 | 32.3 | 25.1 |
| PALO-7B | 68.9 | 71.2 | 79.1 | 54.6 | 71.5 | 83.9 | 61.9 | 66.6 | 80.9 | 74.4 | 68.2 |
| mBLIP mT0-XL | 32.7 | 28.2 | 33.7 | 26.2 | 3.6 | 39.8 | 26.9 | 26.8 | 34.1 | 36.9 | 26.0 |
| mBLIP BLOOMZ-7B | 43.5 | 41.0 | 48.1 | 44.1 | 30.6 | 53.3 | 39.1 | 29.8 | 38.1 | 51.5 | 34.0 |
| PANGEA-7B (Ours) | 84.2 | 89.5 | 91.0 | 94.9 | 94.4 | 93.8 | 84.9 | 92.8 | 91.2 | 87.4 | 75.5 |

Table 7: Comparison of models on the Multilingual LLaVABench benchmark across different languages.

## I.3    CVQA

We show the performance of different models on the CVQA benchmark in Table 8 and Table 9.

## I.4    MARVL

We show the performance of different models on the MaRVL benchmark in Table 10.

## I.5    XM100

We show the performance of different models on the XM100 benchmark in Table 11.

| Models | ar-es | br-pt | bu-bg | ch-es | ch-zh | co-es | ec-es | eg-ar | et-am | et-or |
|---|---|---|---|---|---|---|---|---|---|---|
| Llava-1.5-7B | 37.8 | 51.1 | 35.6 | 42.4 | 44.4 | 50.6 | 48.6 | 31.5 | 27.8 | 31.8 |
| Llava-Next-7B | 52.5 | 62.3 | 41.5 | 59.0 | 51.1 | 54.8 | 50.8 | 33.5 | 29.5 | 36.9 |
| Phi-3.5-Vision | 54.0 | 57.2 | 36.9 | 57.7 | 51.1 | 52.3 | 50.1 | 38.4 | 27.8 | 32.2 |
| Cambrian-8B | 59.6 | 60.6 | 42.0 | 64.5 | 59.5 | 57.7 | 56.1 | 40.9 | 27.8 | 25.7 |
| Llava-OV-7B | 64.5 | 69.7 | 49.6 | 67.1 | 69.1 | 66.8 | 65.5 | 47.8 | 32.5 | 41.1 |
| Molmo-7B-D | 61.1 | 69.0 | 54.9 | 60.7 | 66.2 | 58.5 | 54.9 | 56.7 | 58.1 | 60.7 |
| Llama3.2-11B | 69.1 | 74.6 | 64.2 | 70.5 | 73.6 | 69.3 | 66.9 | 68.5 | 68.4 | 63.1 |
| PaliGemma-3B | 48.7 | 53.9 | 39.1 | 53.4 | 53.7 | 50.6 | 45.3 | 40.4 | 24.8 | 28.0 |
| PALO-7B | 50.9 | 56.7 | 36.7 | 55.1 | 45.3 | 48.5 | 46.4 | 28.6 | 19.2 | 32.7 |
| mBLIP BLOOMZ-7B | 45.3 | 51.4 | 30.5 | 45.3 | 51.1 | 46.9 | 44.8 | 35.9 | 23.9 | 25.7 |
| mBLIP mT0-XL | 40.8 | 44.4 | 38.0 | 44.9 | 39.9 | 41.9 | 42.5 | 31.0 | 35.9 | 26.6 |
| PANGEA-7B (Ours) | 68.3 | 72.9 | 53.9 | 70.5 | 74.0 | 64.7 | 63.5 | 49.3 | 36.3 | 35.5 |

| Models | fr-br | in-bn | in-ta | in-te | ind-id | ind-jv | ind-mi | ind-sv | ir-ir | ja-jp |
|---|---|---|---|---|---|---|---|---|---|---|
| Llava-1.5-7B | 29.4 | 31.1 | 29.8 | 28.0 | 41.7 | 32.0 | 32.7 | 33.5 | 42.6 | 37.4 |
| Llava-Next-7B | 27.4 | 31.1 | 28.8 | 28.0 | 42.2 | 38.7 | 40.2 | 35.5 | 42.6 | 32.5 |
| Phi-3.5-Vision | 29.3 | 39.0 | 40.0 | 36.8 | 45.0 | 38.2 | 38.2 | 30.8 | 39.6 | 39.7 |
| Cambrian-8B | 31.6 | 47.2 | 38.1 | 44.0 | 50.2 | 43.8 | 39.4 | 45.5 | 47.9 | 40.9 |
| Llava-OV-7B | 34.3 | 56.3 | 43.9 | 46.5 | 58.0 | 45.8 | 45.4 | 40.5 | 50.6 | 49.8 |
| Molmo-7B-D | 44.2 | 61.9 | 61.2 | 58.5 | 52.9 | 53.9 | 54.6 | 55.0 | 64.4 | 42.9 |
| Llama3.2-11B | 49.4 | 76.9 | 80.4 | 80.5 | 65.8 | 60.6 | 68.9 | 64.0 | 76.4 | 54.2 |
| PaliGemma-3B | 29.9 | 46.2 | 46.0 | 43.5 | 45.4 | 41.4 | 39.8 | 33.0 | 34.4 | 43.3 |
| PALO-7B | 29.1 | 37.8 | 31.2 | 25.0 | 41.3 | 32.3 | 32.3 | 32.0 | 42.9 | 30.5 |
| mBLIP BLOOMZ-7B | 26.7 | 41.9 | 40.0 | 42.0 | 41.9 | 35.4 | 35.1 | 32.0 | 29.4 | 31.0 |
| mBLIP mT0-XL | 23.5 | 36.4 | 44.2 | 39.0 | 37.4 | 37.4 | 34.7 | 31.0 | 35.3 | 30.0 |
| PANGEA-7B (Ours) | 34.6 | 59.1 | 51.9 | 54.5 | 62.1 | 49.5 | 47.8 | 53.0 | 56.4 | 48.3 |

| Models | ke-sw | ma-my | me-es | mo-mg | ni-ig | no-ng | pk-ur | ph-fi | ro-ro | ru-ru |
|---|---|---|---|---|---|---|---|---|---|---|
| Llava-1.5-7B | 34.4 | 42.2 | 42.4 | 26.9 | 34.5 | 47.5 | 26.4 | 43.8 | 47.0 | 51.0 |
| Llava-Next-7B | 46.2 | 45.7 | 51.4 | 33.3 | 35.0 | 56.9 | 36.6 | 46.8 | 52.3 | 53.5 |
| Phi-3.5-Vision | 46.0 | 45.1 | 46.3 | 31.9 | 33.3 | 50.0 | 35.2 | 41.4 | 47.4 | 50.5 |
| Cambrian-8B | 50.5 | 52.1 | 56.7 | 34.6 | 36.0 | 53.5 | 48.6 | 47.3 | 52.0 | 61.5 |
| Llava-OV-7B | 46.5 | 55.6 | 59.4 | 35.9 | 33.5 | 62.5 | 58.3 | 56.2 | 60.3 | 75.5 |
| Molmo-7B-D | 73.3 | 54.6 | 53.6 | 51.9 | 53.0 | 54.8 | 67.1 | 57.6 | 63.6 | 61.5 |
| Llama3.2-11B | 79.1 | 72.1 | 66.6 | 54.5 | 61.5 | 66.9 | 78.7 | 70.0 | 76.8 | 74.5 |
| PaliGemma-3B | 44.0 | 44.1 | 47.4 | 29.2 | 32.0 | 52.2 | 44.9 | 39.9 | 50.3 | 53.5 |
| PALO-7B | 35.9 | 42.5 | 44.3 | 28.8 | 29.5 | 49.2 | 44.4 | 39.4 | 46.0 | 47.0 |
| mBLIP BLOOMZ-7B | 37.0 | 42.5 | 44.8 | 28.8 | 33.0 | 49.2 | 47.7 | 31.5 | 46.0 | 34.0 |
| mBLIP mT0-XL | 45.1 | 40.6 | 44.9 | 29.2 | 30.5 | 42.8 | 40.3 | 32.0 | 43.7 | 42.0 |
| PANGEA-7B (Ours) | 64.1 | 59.7 | 62.2 | 42.3 | 46.0 | 64.5 | 66.2 | 58.6 | 64.6 | 74.0 |

| Models | rw-ki | sg-zh | sk-ko | sp-es | sr-si | ur-es | macro | | | |
|---|---|---|---|---|---|---|---|---|---|---|
| Llava-1.5-7B | 31.1 | 44.3 | 44.5 | 56.9 | 24.9 | 37.8 | 38.7 | | | |
| Llava-Next-7B | 34.5 | 44.8 | 43.4 | 63.5 | 29.8 | 41.0 | 42.6 | | | |
| Phi-3.5-Vision | 31.1 | 43.9 | 55.2 | 62.4 | 28.0 | 43.3 | 42.4 | | | |
| Cambrian-8B | 31.9 | 54.7 | 54.5 | 70.4 | 36.4 | 45.7 | 47.5 | | | |
| Llava-OV-7B | 35.3 | 70.3 | 65.2 | 79.9 | 31.6 | 47.3 | 53.8 | | | |
| Molmo-7B-D | 57.4 | 69.3 | 65.2 | 70.1 | 68.0 | 50.8 | 59.4 | | | |
| Llama3.2-11B | 57.9 | 80.7 | 73.8 | 81.4 | 72.4 | 52.4 | 70.1 | | | |
| PaliGemma-3B | 27.2 | 48.6 | 61.0 | 60.1 | 31.6 | 39.4 | 43.0 | | | |
| PALO-7B | 28.9 | 45.8 | 44.5 | 64.8 | 28.0 | 39.4 | 39.3 | | | |
| mBLIP BLOOMZ-7B | 29.4 | 47.6 | 33.1 | 56.6 | 28.0 | 39.4 | 36.9 | | | |
| mBLIP mT0-XL | 33.2 | 36.8 | 38.3 | 53.5 | 31.1 | 39.1 | 37.6 | | | |
| PANGEA-7B (Ours) | 35.7 | 65.6 | 70.7 | 72.6 | 39.1 | 49.8 | 57.2 | | | |

Table 8: Comparison of models on CVQA across different country-language pairs (in local languages). Includes Macro-Acc.

## I.6   xGQA

We show the performance of different models on the xGQA benchmark in Table 12.

## I.7   MAXM

We show the performance of different models on the MAXM benchmark in Table 13.

| Models | ar-es | br-pt | bu-bg | ch-es | ch-zh | co-es | ec-es | eg-ar | et-am | et-or |
|---|---|---|---|---|---|---|---|---|---|---|
| Llava-1.5-7B | 56.2 | 61.6 | 52.3 | 60.2 | 54.0 | 55.6 | 55.5 | 50.2 | 51.3 | 53.3 |
| Llava-Next-7B | 53.9 | 61.3 | 50.9 | 59.8 | 58.8 | 60.2 | 52.8 | 54.7 | 52.9 | 58.9 |
| Phi-3.5-Vision | 59.2 | 61.9 | 54.9 | 64.1 | 58.2 | 59.3 | 57.5 | 50.7 | 54.7 | 58.4 |
| Cambrian-8B | 57.7 | 66.5 | 56.1 | 65.4 | 64.3 | 59.3 | 60.2 | 56.7 | 60.3 | 56.5 |
| Llava-OV-7B | 63.0 | 73.9 | 59.3 | 65.8 | 68.8 | 65.1 | 63.3 | 62.1 | 59.8 | 59.3 |
| Molmo-7B-D | 57.7 | 65.8 | 45.6 | 63.7 | 68.5 | 57.3 | 55.0 | 43.8 | 31.6 | 38.8 |
| Llama3.2-11B | 66.8 | 72.9 | 54.4 | 72.6 | 72.0 | 66.4 | 65.2 | 56.7 | 41.9 | 32.2 |
| PaliGemma-3B | 51.7 | 59.5 | 49.3 | 51.7 | 54.9 | 54.8 | 47.2 | 51.2 | 52.6 | 51.4 |
| PALO-7B | 50.2 | 57.0 | 48.8 | 53.4 | 52.1 | 51.9 | 53.0 | 48.3 | 47.0 | 52.3 |
| mBLIP mT0-XL | 38.1 | 45.4 | 39.1 | 42.7 | 43.7 | 41.1 | 40.9 | 42.9 | 34.2 | 42.1 |
| mBLIP BLOOMZ-7B | 46.0 | 51.4 | 41.5 | 44.4 | 48.9 | 49.0 | 45.0 | 45.3 | 38.9 | 46.3 |
| Pangea-7B (Ours) | 67.2 | 72.9 | 60.1 | 68.8 | 67.2 | 64.7 | 61.6 | 59.1 | 60.7 | 56.0 |

| Models | fr-br | in-bn | in-ta | in-te | ind-id | ind-jv | ind-mi | ind-sv | ir-ir | ja-jp |
|---|---|---|---|---|---|---|---|---|---|---|
| Llava-1.5-7B | 37.3 | 52.1 | 61.4 | 63.5 | 47.8 | 50.8 | 49.0 | 44.0 | 61.3 | 41.9 |
| Llava-Next-7B | 37.5 | 60.8 | 61.4 | 60.5 | 48.5 | 48.1 | 51.4 | 49.0 | 66.6 | 40.9 |
| Phi-3.5-Vision | 41.7 | 58.7 | 60.5 | 60.0 | 51.7 | 45.5 | 51.4 | 47.5 | 62.6 | 41.4 |
| Cambrian-8B | 40.7 | 68.5 | 65.6 | 63.0 | 55.1 | 50.2 | 58.2 | 56.0 | 66.6 | 42.4 |
| Llava-OV-7B | 44.2 | 69.6 | 72.0 | 70.5 | 59.0 | 55.9 | 59.4 | 58.5 | 76.4 | 47.3 |
| Molmo-7B-D | 29.6 | 47.9 | 36.4 | 41.5 | 50.5 | 45.1 | 43.4 | 39.5 | 43.6 | 44.8 |
| Llama3.2-11B | 36.3 | 62.9 | 66.4 | 66.5 | 63.6 | 48.8 | 58.2 | 54.0 | 57.4 | 58.1 |
| PaliGemma-3B | 37.3 | 59.1 | 66.0 | 62.5 | 49.3 | 48.1 | 43.4 | 46.0 | 58.3 | 44.8 |
| PALO-7B | 36.8 | 52.4 | 53.5 | 56.5 | 45.1 | 45.8 | 44.2 | 42.0 | 55.6 | 37.4 |
| mBLIP mT0-XL | 30.4 | 43.0 | 46.0 | 41.0 | 38.1 | 39.1 | 38.6 | 32.5 | 37.4 | 34.0 |
| mBLIP BLOOMZ-7B | 34.6 | 43.4 | 52.6 | 49.5 | 41.0 | 44.8 | 38.2 | 30.5 | 42.3 | 36.5 |
| Pangea-7B (Ours) | 45.2 | 67.1 | 71.0 | 68.0 | 60.4 | 57.2 | 56.9 | 56.0 | 72.7 | 45.8 |

| Models | ke-sw | ma-my | me-es | mo-mg | ni-ig | no-ng | pk-ur | ph-fi | ro-ro | ru-ru |
|---|---|---|---|---|---|---|---|---|---|---|
| Llava-1.5-7B | 68.9 | 52.1 | 47.9 | 45.8 | 51.0 | 58.5 | 63.9 | 52.7 | 55.6 | 59.0 |
| Llava-Next-7B | 71.1 | 54.9 | 51.1 | 44.2 | 53.0 | 57.2 | 67.1 | 56.7 | 62.6 | 58.5 |
| Phi-3.5-Vision | 72.9 | 57.1 | 46.3 | 50.7 | 53.0 | 56.2 | 60.6 | 57.6 | 61.9 | 58.5 |
| Cambrian-8B | 74.4 | 61.9 | 56.7 | 48.7 | 56.5 | 60.5 | 73.1 | 60.1 | 66.6 | 61.5 |
| Llava-OV-7B | 79.1 | 65.1 | 63.2 | 52.6 | 57.5 | 64.2 | 75.0 | 64.0 | 72.5 | 72.5 |
| Molmo-7B-D | 47.6 | 51.7 | 55.1 | 35.9 | 36.0 | 49.2 | 46.8 | 43.3 | 52.0 | 63.5 |
| Llama3.2-11B | 61.5 | 69.2 | 64.7 | 41.0 | 39.5 | 65.9 | 65.7 | 66.0 | 75.5 | 74.5 |
| PaliGemma-3B | 59.7 | 54.9 | 51.7 | 43.4 | 46.0 | 55.2 | 67.6 | 48.8 | 60.9 | 56.0 |
| PALO-7B | 65.9 | 49.2 | 53.4 | 42.9 | 49.0 | 54.5 | 60.6 | 52.7 | 55.0 | 53.5 |
| mBLIP mT0-XL | 50.2 | 41.6 | 34.7 | 33.9 | 39.5 | 43.1 | 45.4 | 36.9 | 43.7 | 41.0 |
| mBLIP BLOOMZ-7B | 54.6 | 45.7 | 39.3 | 38.1 | 45.0 | 47.2 | 60.6 | 36.9 | 50.3 | 44.0 |
| Pangea-7B (Ours) | 77.2 | 62.5 | 61.6 | 52.9 | 59.5 | 64.9 | 72.2 | 64.0 | 71.9 | 68.5 |

| Models | rw-ki | sg-zh | sk-ko | sp-es | sr-si | ur-es | macro |
|---|---|---|---|---|---|---|---|
| Llava-1.5-7B | 51.1 | 60.8 | 56.9 | 66.0 | 58.7 | 42.5 | 54.2 |
| Llava-Next-7B | 52.8 | 62.3 | 60.0 | 67.6 | 59.1 | 38.7 | 55.7 |
| Phi-3.5-Vision | 52.3 | 59.4 | 66.5 | 66.7 | 61.3 | 46.3 | 56.3 |
| Cambrian-8B | 56.2 | 66.0 | 63.1 | 71.7 | 63.1 | 47.0 | 59.7 |
| Llava-OV-7B | 55.7 | 73.6 | 67.9 | 80.2 | 72.9 | 48.9 | 65.2 |
| Molmo-7B-D | 34.9 | 66.0 | 56.9 | 66.7 | 31.6 | 44.8 | 48.3 |
| Llama3.2-11B | 40.4 | 73.6 | 73.1 | 83.3 | 51.1 | 56.2 | 61.2 |
| PaliGemma-3B | 44.7 | 59.4 | 58.3 | 61.0 | 62.2 | 40.6 | 52.9 |
| PALO-7B | 51.9 | 56.1 | 55.9 | 62.9 | 54.2 | 42.2 | 50.9 |
| mBLIP mT0-XL | 38.3 | 43.9 | 41.4 | 51.9 | 48.0 | 34.9 | 40.5 |
| mBLIP BLOOMZ-7B | 45.1 | 53.8 | 46.9 | 58.5 | 46.7 | 34.0 | 44.9 |
| Pangea-7B (Ours) | 56.6 | 71.7 | 66.6 | 75.2 | 70.6 | 52.7 | 64.4 |

Table 9: Comparison of models on CVQA across different country-language pairs (in English). Includes Macro-Acc.

| Models | English | Multi | Indonesian | Swahili | Tamil | Turkish | Chinese |
|---|---|---|---|---|---|---|---|
| GPT4o | 81.8 | 82.3 | 81.9 | 80.8 | 80.2 | 86.4 | 82.1 |
| Gemini-1.5-Pro | 76.4 | 72.0 | 71.2 | 67.8 | 70.0 | 75.4 | 75.8 |
| Llava-1.5-7B | 56.2 | 53.7 | 56.1 | 49.8 | 49.7 | 55.4 | 57.5 |
| Llava-Next-7B | 62.8 | 50.9 | 52.2 | 50.6 | 50.5 | 50.4 | 50.6 |
| Phi-3.5-Vision | 72.1 | 56.5 | 58.6 | 51.4 | 52.0 | 58.6 | 61.7 |
| Cambrian-8B | 75.4 | 61.8 | 64.7 | 53.6 | 56.7 | 65.2 | 68.9 |
| Llava-OV-7B | 72.7 | 57.5 | 60.9 | 51.2 | 51.9 | 63.5 | 60.0 |
| Molmo-7B-D | 65.3 | 54.9 | 61.1 | 49.6 | 49.6 | 52.2 | 62.2 |
| Llama3.2-11B | 64.5 | 58.1 | 62.7 | 52.4 | 54.0 | 61.6 | 59.5 |
| PaliGemma-3b | 56.5 | 52.2 | 53.4 | 49.6 | 50.5 | 56.3 | 51.3 |
| PALO-7B | 63.3 | 54.2 | 58.3 | 50.6 | 51.9 | 54.9 | 55.3 |
| mBLIP mT0-XL | 67.3 | 66.7 | 64.9 | 64.8 | 69.7 | 68.1 | 65.9 |
| mBLIP BLOOMZ-7B | 62.3 | 58.6 | 59.1 | 56.2 | 60.3 | 57.7 | 59.7 |
| PANGEA-7B | 87.0 | 79.0 | 81.3 | 75.1 | 69.4 | 84.8 | 84.3 |

Table 10: Comparison of models on the MaRVL dataset across different languages.

## I.8 XMMMU

We show the performance of different models on the xMMMU benchmark in Table 14.

## I.9 M3EXAM

We show the performance of different models on the M3Exam benchmark in Table 15.

## I.10 TYDIQA

We show the performance of different models on the TyDiQA benchmark in Table 16.

## I.11 XSTORYCLOZE

We show the performance of different models on the XStoryCloze benchmark in Table 17.

## I.12 MGSM

We show the performance of different models on the MGSM benchmark in Table 18.

## I.13 MMMLU

We show the performance of different models on the MMMLU benchmark in Table 19.

## J  A PRELIMINARY EXPLORATION OF CONSTRUCTING MULTILINGUAL OCR INSTRUCTIONS

Optical Character Recognition (OCR) is a critical capability for multimodal LLMs, enabling them to interpret and process textual information embedded within images. However, most existing OCR training datasets are predominantly English-centric, which limits the models' performance in non-English contexts. To address this gap, we have curated a comprehensive set of 500K multilingual OCR training samples from web user interfaces, spanning 10 languages, with 50K examples per language, sourced from web user interfaces. Webpages naturally serve as image-rich environments containing abundant text, and by capturing screenshots of websites from various countries in different languages, we were able to gather a substantial number of OCR images.

We utilize URLs from the CC-News-Multilingual[3] dataset (Hamborg et al., 2017) to obtain a diverse set of multilingual web pages. Using Playwright[4], we render each website and automatically capture screenshots under various device settings and resolutions to achieve a wide range of image dimensions and aspect ratios. Each screenshot includes a red bounding box that highlights a specific element targeted for OCR extraction. We focus on ten languages for this dataset: English, Chinese, Japanese, Korean, Indonesian, Hindi, Spanish, French, Portuguese, and Arabic. We totally have 1M samples (50K for each language).

We employed the same model architecture as PANGEA but trained it exclusively on these OCR images, reserving a portion of the data as a test set. As shown in Figure 27 , the results indicate that improving multilingual OCR performance is feasible with an increase in training samples. However, the OCR accuracy for non-Latin scripts (e.g., Chinese, Japanese, and Korean) remains lower than for Latin-based languages. Looking ahead, we aim to further expand the multilingual OCR training dataset to include more languages and integrate this data into PANGEAINS.

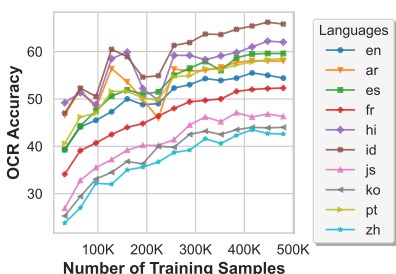

Figure 27: A preliminary exploration of multilingual OCR.

---

[3]https://huggingface.co/datasets/intfloat/multilingual_cc_news
[4]https://github.com/microsoft/playwright

| Models | English | Multi | Arabic | Bengali | Czech | Danish | German | Greek |
|---|---|---|---|---|---|---|---|---|
| Gemini-1.5-Pro | 27.6 | 19.1 | 1.7 | 7.5 | 25.9 | 32.8 | 27.6 | 5.0 |
| GPT4o | 27.7 | 19.1 | 15.8 | 13.5 | 21.1 | 25.3 | 19.3 | 21.1 |
| Llava-1.5-7B | 28.6 | 1.1 | 0.0 | 0.0 | 2.1 | 1.0 | 3.1 | 0.0 |
| Llava-Next-7B | 29.3 | 9.4 | 5.6 | 0.1 | 12.1 | 15.7 | 14.4 | 4.2 |
| Phi-3.5-Vision | 30.2 | 5.2 | 0.4 | 2.4 | 16.6 | 16.2 | 0.0 | 20.7 |
| Cambrian-8B | 20.6 | 9.9 | 1.4 | 6.6 | 7.4 | 15.1 | 15.5 | 4.4 |
| Llava-OV-7B | 30.6 | 7.0 | 0.2 | 0.6 | 5.2 | 16.8 | 14.0 | 0.4 |
| Molmo-7B-D | 22.1 | 9.1 | 5.4 | 7.9 | 5.7 | 13.8 | 12.2 | 4.2 |
| Llama3.2-11B | 27.6 | 4.5 | 0.0 | 0.0 | 1.5 | 11.8 | 4.6 | 1.2 |
| PaliGemma-3B | 18.7 | 0.8 | 0.0 | 0.0 | 1.1 | 3.1 | 2.7 | 0.0 |
| PALO-7B | 30.4 | 0.8 | 0.0 | 0.0 | 2.0 | 1.0 | 2.7 | 0.0 |
| mBLIP mT0-XL | 31.9 | 3.1 | 3.2 | 1.6 | 3.7 | 2.1 | 2.9 | 3.1 |
| mBLIP BLOOMZ | 22.5 | 10.3 | 9.5 | 6.4 | 11.5 | 15.9 | 14.5 | 10.9 |
| PANGEA-7B (Ours) | 30.4 | 14.2 | 18.1 | 16.4 | 16.2 | 20.7 | 20.6 | 11.2 |

| Models | Spanish | Persian | Finnish | Filipino | French | Hebrew | Hindi | Croatian |
|---|---|---|---|---|---|---|---|---|
| Gemini-1.5-Pro | 39.5 | 4.2 | 29.0 | 28.7 | 42.4 | 4.3 | 2.2 | 33.8 |
| GPT4o | 28.3 | 26.6 | 13.1 | 26.4 | 23.1 | 20.4 | 17.0 | 19.4 |
| Llava-1.5-7B | 3.7 | 0.0 | 0.4 | 1.1 | 2.0 | 0.1 | 0.0 | 0.3 |
| Llava-Next-7B | 23.6 | 9.4 | 5.5 | 9.3 | 23.0 | 2.7 | 10.2 | 7.5 |
| Phi-3.5-Vision | 20.7 | 0.0 | 1.0 | 1.7 | 21.2 | 0.3 | 0.0 | 0.5 |
| Cambrian-8B | 18.6 | 9.6 | 5.1 | 19.6 | 18.3 | 5.8 | 6.8 | 7.2 |
| Llava-OV-7B | 24.9 | 3.8 | 1.5 | 4.2 | 22.0 | 0.0 | 4.4 | 7.2 |
| Molmo-7B-D | 19.8 | 11.3 | 3.1 | 13.0 | 19.8 | 8.3 | 9.4 | 6.9 |
| Llama3.2-11B | 10.2 | 0.0 | 2.4 | 8.4 | 12.0 | 0.0 | 0.2 | 0.7 |
| PaliGemma-3B | 0.7 | 0.0 | 0.1 | 0.1 | 0.6 | 0.0 | 0.0 | 1.3 |
| PALO-7B | 1.5 | 0.0 | 0.4 | 0.9 | 2.1 | 0.0 | 0.0 | 0.2 |
| mBLIP mT0-XL | 8.3 | 5.5 | 1.7 | 2.8 | 6.4 | 4.0 | 1.8 | 0.9 |
| mBLIP BLOOMZ | 18.9 | 13.8 | 4.8 | 7.7 | 19.1 | 7.5 | 10.1 | 3.2 |
| PANGEA-7B (Ours) | 26.2 | 19.3 | 3.8 | 18.9 | 26.7 | 18.2 | 17.4 | 10.8 |

| Models | Hungarian | Indonesian | Italian | Japanese | Korean | Maori | Dutch | Norwegian |
|---|---|---|---|---|---|---|---|---|
| Gemini-1.5-Pro | 37.2 | 55.4 | 27.6 | 1.2 | 8.2 | 3.8 | 27.7 | 36.7 |
| GPT4o | 21.8 | 28.4 | 21.0 | 0.0 | 11.1 | 26.8 | 26.4 | 24.7 |
| Llava-1.5-7B | 3.3 | 0.9 | 4.3 | 0.0 | 0.0 | 0.2 | 2.9 | 3.7 |
| Llava-Next-7B | 9.3 | 14.7 | 17.6 | 4.2 | 5.2 | 9.2 | 23.8 | 16.3 |
| Phi-3.5-Vision | 3.4 | 3.2 | 17.5 | 1.6 | 0.3 | 0.2 | 17.2 | 14.1 |
| Cambrian-8B | 6.6 | 15.7 | 15.5 | 7.2 | 2.0 | 3.2 | 20.3 | 16.0 |
| Llava-OV-7B | 3.6 | 16.4 | 12.8 | 0.6 | 0.0 | 1.7 | 24.7 | 13.9 |
| Molmo-7B-D | 3.5 | 17.2 | 17.8 | 5.2 | 2.4 | 7.5 | 15.7 | 13.8 |
| Llama3.2-11B | 12.7 | 1.2 | 16.0 | 0.0 | 0.0 | 9.3 | 22.0 | 1.1 |
| PaliGemma-3B | 2.0 | 0.2 | 1.8 | 0.0 | 0.0 | 4.0 | 2.6 | 2.3 |
| PALO-7B | 3.4 | 1.1 | 3.2 | 0.0 | 0.0 | 0.1 | 3.5 | 0.7 |
| mBLIP mT0-XL | 2.8 | 6.0 | 2.8 | 0.3 | 2.1 | 1.5 | 3.4 | 3.1 |
| mBLIP BLOOMZ | 11.8 | 16.0 | 16.5 | 0.0 | 4.5 | 0.1 | 18.2 | 14.5 |
| PANGEA-7B (Ours) | 7.7 | 27.9 | 22.9 | 2.1 | 8.1 | 0.7 | 26.6 | 24.9 |

| Models | Polish | Portuguese | Quechua | Romanian | Russian | Swedish | Swahili | Telugu |
|---|---|---|---|---|---|---|---|---|
| Gemini-1.5-Pro | 35.5 | 35.7 | 0.7 | 31.2 | 32.4 | 37.8 | 10.7 | 0.0 |
| GPT4o | 22.2 | 28.0 | 4.4 | 19.1 | 20.7 | 26.0 | 20.0 | 12.5 |
| Llava-1.5-7B | 0.8 | 2.5 | 0.0 | 1.6 | 0.5 | 2.0 | 0.1 | 0.0 |
| Llava-Next-7B | 13.5 | 21.3 | 0.0 | 11.5 | 13.5 | 16.0 | 3.2 | 0.0 |
| Phi-3.5-Vision | 1.0 | 21.0 | 0.4 | 3.2 | 0.7 | 12.5 | 0.4 | 0.0 |
| Cambrian-8B | 9.3 | 17.5 | 0.0 | 13.4 | 11.3 | 17.9 | 3.7 | 2.3 |
| Llava-OV-7B | 7.4 | 24.6 | 0.0 | 6.8 | 5.5 | 15.0 | 2.0 | 0.0 |
| Molmo-7B-D | 8.2 | 16.2 | 0.6 | 11.6 | 12.3 | 14.1 | 3.8 | 0.4 |
| Llama3.2-11B | 1.0 | 18.6 | 0.0 | 10.1 | 0.6 | 7.4 | 5.8 | 0.0 |
| PaliGemma-3B | 0.9 | 1.3 | 0.1 | 0.8 | 0.0 | 2.0 | 0.0 | 0.0 |
| PALO-7B | 0.8 | 1.7 | 0.0 | 1.1 | 0.5 | 0.9 | 0.2 | 0.0 |
| mBLIP mT0-XL | 3.5 | 5.8 | 0.2 | 2.3 | 3.1 | 3.7 | 3.8 | 2.7 |
| mBLIP BLOOMZ | 11.8 | 16.5 | 0.1 | 13.7 | 14.5 | 14.5 | 8.4 | 3.0 |
| PANGEA-7B (Ours) | 16.2 | 28.1 | 0.0 | 21.4 | 20.9 | 19.4 | 18.7 | 0.1 |

| Models | Thai | Turkish | Ukrainian | Vietnamese | Chinese |
|---|---|---|---|---|---|
| Gemini-1.5-Pro | 0.0 | 0.9 | 0.0 | 0.0 | 0.9 |
| GPT4o | 0.0 | 17.6 | 16.9 | 30.9 | 0.4 |
| Llava-1.5-7B | 0.0 | 0.0 | 0.0 | 0.0 | 0.0 |
| Llava-Next-7B | 0.0 | 0.0 | 0.3 | 0.0 | 6.3 |
| Phi-3.5-Vision | 0.5 | 1.9 | 0.0 | 2.2 | 0.0 |
| Cambrian-8B | 0.4 | 9.3 | 5.9 | 17.8 | 11.3 |
| Llava-OV-7B | 0.0 | 0.0 | 0.0 | 0.0 | 2.9 |
| Molmo-7B-D | 0.0 | 0.0 | 0.0 | 0.0 | 0.0 |
| Llama3.2-11B | 0.0 | 0.0 | 0.0 | 0.0 | 2.9 |
| PaliGemma-3B | 0.5 | 0.0 | 0.0 | 0.2 | 0.0 |
| PALO-7B | 0.2 | 0.0 | 0.0 | 0.1 | 0.0 |
| mBLIP mT0-XL | 0.0 | 3.9 | 2.0 | 7.1 | 0.0 |
| mBLIP BLOOMZ | 0.5 | 1.9 | 0.0 | 2.2 | 0.0 |
| PANGEA-7B (Ours) | 0.0 | 0.0 | 0.3 | 0.0 | 4.9 |

Table 11: Comparison of models on the XM100 dataset across different languages.

| Models | English | Multi | Bengali | German | Indonesian | Korean | Portuguese | Russian | Chinese |
|---|---|---|---|---|---|---|---|---|---|
| Gemini-1.5-Pro | 54.2 | 48.7 | 49.4 | 50.2 | 48.6 | 46.4 | 51.2 | 44.8 | 50.2 |
| GPT4o | 55.8 | 51.0 | 49.4 | 52.6 | 50.4 | 51.0 | 52.2 | 50.0 | 51.4 |
| Llava-1.5-7B | 62.0 | 30.7 | 15.6 | 28.4 | 33.4 | 38.2 | 27.5 | 33.1 | 38.4 |
| Llava-Next-7B | 64.8 | 37.8 | 11.5 | 41.5 | 37.3 | 42.5 | 39.8 | 43.5 | 48.2 |
| Phi-3.5-Vision | 64.7 | 38.4 | 7.7 | 51.4 | 36.0 | 36.3 | 49.6 | 46.2 | 41.4 |
| Cambrian-8B | 64.6 | 39.8 | 32.3 | 44.6 | 36.0 | 43.6 | 41.6 | 44.2 | 36.2 |
| Llava-OV-7B | 64.4 | 48.2 | 41.8 | 49.2 | 48.8 | 45.3 | 52.4 | 54.0 | 45.9 |
| Molmo-7B-D | 51.5 | 43.0 | 25.6 | 45.9 | 44.9 | 44.2 | 46.5 | 45.6 | 48.1 |
| Llama3.2-11B | 55.6 | 45.4 | 42.9 | 46.7 | 46.2 | 44.5 | 46.5 | 44.7 | 46.1 |
| PaliGemma-3B | 59.7 | 30.5 | 13.3 | 44.5 | 21.3 | 22.8 | 34.7 | 35.8 | 41.2 |
| PALO-7B | 60.5 | 37.8 | 42.2 | 39.1 | 36.8 | 41.7 | 31.7 | 27.0 | 46.5 |
| mBLIP mT0-XL | 44.2 | 39.9 | 39.1 | 41.1 | 39.1 | 39.7 | 40.7 | 40.2 | 39.4 |
| mBLIP BLOOMZ-7B | 43.3 | 36.9 | 37.7 | 36.3 | 39.3 | 28.5 | 40.7 | 36.6 | 39.1 |
| PANGEA-7B (Ours) | 64.7 | 60.2 | 58.9 | 61.6 | 60.1 | 58.9 | 61.8 | 60.4 | 59.6 |

Table 12: Comparison of models on the xGQA dataset across different languages

| Models | English | Multi | French | Hindi | Hebrew | Romanian | Thai | Chinese |
|---|---|---|---|---|---|---|---|---|
| Gemini-1.5-Pro | 56.4 | 63.5 | 60.2 | 66.5 | 65.7 | 57.4 | 73.9 | 57.4 |
| GPT4o | 60.7 | 65.4 | 59.8 | 68.8 | 70.0 | 61.3 | 76.5 | 56.3 |
| Llava-1.5-7B | 49.8 | 20.4 | 32.2 | 17.3 | 12.9 | 15.1 | 17.2 | 27.8 |
| Llava-Next-7B | 54.9 | 21.4 | 33.7 | 16.2 | 10.7 | 15.5 | 18.3 | 33.9 |
| Phi-3.5-Vision | 55.3 | 25.0 | 38.3 | 31.9 | 17.5 | 10.9 | 24.3 | 27.4 |
| Cambrian-8B | 55.3 | 28.7 | 41.7 | 23.8 | 17.1 | 32.0 | 25.7 | 31.8 |
| Llava-OV-7B | 54.9 | 34.8 | 37.9 | 31.9 | 17.8 | 30.2 | 53.0 | 37.9 |
| Molmo-7B-D | 52.9 | 37.5 | 45.5 | 33.5 | 30.7 | 28.9 | 46.3 | 40.4 |
| Llama3.2-11B | 55.3 | 43.9 | 48.1 | 50.4 | 41.8 | 36.6 | 56.7 | 30.0 |
| PaliGemma-3B | 47.9 | 19.9 | 8.0 | 36.5 | 19.3 | 13.4 | 31.3 | 10.8 |
| PALO-7B | 51.4 | 16.3 | 33.7 | 15.8 | 12.1 | 11.3 | 14.6 | 10.5 |
| mBLIP mT0-XL | 44.7 | 36.8 | 36.0 | 42.7 | 28.9 | 30.3 | 56.3 | 26.4 |
| mBLIP BLOOMZ-7B | 44.7 | 24.8 | 33.0 | 47.3 | 8.9 | 16.9 | 9.7 | 33.2 |
| PANGEA-7B (Ours) | 55.3 | 53.3 | 43.6 | 53.5 | 59.3 | 45.8 | 67.2 | 50.2 |

Table 13: Comparison of models on the MAXM dataset across different languages.

| Models | English | Multi | Arabic | French | Hindi | Indonesian | Japanese | Portuguese |
|---|---|---|---|---|---|---|---|---|
| Gemini-1.5-Pro (0801) | 65.8 | 57.7 | 57.7 | 58.1 | 55.5 | 60.2 | 55.0 | 59.6 |
| GPT4o (0513) | 69.1 | 58.3 | 56.7 | 58.1 | 58.1 | 59.9 | 58.0 | 58.9 |
| Llava-1.5-7B | 36.2 | 31.5 | 29.5 | 34.9 | 27.5 | 31.6 | 32.0 | 33.7 |
| Llava-Next-7B | 36.7 | 34.3 | 30.5 | 35.6 | 30.9 | 37.0 | 34.9 | 37.0 |
| Phi-3.5-Vision | 42.6 | 38.8 | 35.6 | 44.0 | 30.9 | 36.7 | 37.9 | 47.8 |
| Cambrian-8B | 41.8 | 33.2 | 32.6 | 34.6 | 30.9 | 31.3 | 33.5 | 36.0 |
| Llava-OV-7B | 46.3 | 41.0 | 41.6 | 43.0 | 34.7 | 43.4 | 40.1 | 43.4 |
| Molmo-7B-D | 42.9 | 40.4 | 40.6 | 42.6 | 32.6 | 40.7 | 43.9 | 42.1 |
| Llama3.2-11B | 39.2 | 34.0 | 33.6 | 39.6 | 32.3 | 36.7 | 29.0 | 33.0 |
| PaliGemma-3B | 26.3 | 25.2 | 29.2 | 23.8 | 21.6 | 24.2 | 24.5 | 27.6 |
| PALO-7B | 33.1 | 30.5 | 30.5 | 33.2 | 28.9 | 34.0 | 27.1 | 33.3 |
| mBLIP mT0-XL | 29.3 | 30.4 | 30.2 | 33.2 | 28.2 | 26.9 | 31.6 | 32.3 |
| mBLIP BLOOMZ-7B | 29.2 | 30.8 | 28.5 | 33.9 | 27.8 | 33.3 | 31.6 | 29.6 |
| PANGEA-7B (Ours) | 45.7 | 43.7 | 42.3 | 45.3 | 41.6 | 46.5 | 40.5 | 46.1 |

Table 14: Comparison of models on the xMMMU dataset across different languages.

| Models | English | Multi | Afrikaans | Chinese | Italian | Portuguese | Thai | Vietnamese |
|---|---|---|---|---|---|---|---|---|
| Gemini-1.5-Pro | 77.4 | 64.7 | 80.4 | 74.1 | 76.3 | 61.8 | 49.9 | 46.0 |
| GPT4o | 68.0 | 61.0 | 73.0 | 68.0 | 67.0 | 58.0 | 52.0 | 48.3 |
| Llava-1.5-7B | 32.3 | 29.0 | 28.2 | 24.3 | 40.1 | 28.2 | 23.7 | 29.3 |
| Llava-Next-7B | 36.5 | 28.4 | 28.2 | 25.4 | 37.8 | 27.0 | 23.7 | 28.4 |
| Phi-3.5-Vision | 55.8 | 37.2 | 44.2 | 40.8 | 51.4 | 40.3 | 25.2 | 21.6 |
| Cambrian-8B | 34.7 | 33.4 | 36.8 | 34.2 | 45.2 | 30.3 | 28.9 | 25.0 |
| Llava-OV-7B | 60.4 | 45.8 | 50.3 | 58.0 | 57.2 | 43.8 | 30.9 | 34.5 |
| Molmo-7B-D | 57.1 | 39.1 | 35.6 | 56.4 | 49.4 | 40.2 | 27.4 | 25.9 |
| Llama3.2-11B | 51.8 | 36.6 | 42.3 | 46.4 | 45.8 | 28.4 | 26.4 | 30.2 |
| PaliGemma-3B | 36.0 | 25.6 | 26.4 | 24.7 | 32.2 | 24.3 | 27.2 | 19.0 |
| PALO-7B | 30.8 | 27.8 | 31.9 | 22.1 | 36.9 | 32.3 | 22.7 | 20.7 |
| mBLIP mT0-XL | 22.8 | 25.0 | 16.0 | 25.6 | 33.7 | 21.2 | 22.4 | 31.0 |
| mBLIP BLOOMZ-7B | 30.3 | 29.5 | 28.2 | 29.8 | 37.3 | 28.3 | 22.9 | 30.2 |
| PANGEA-7B (Ours) | 61.4 | 42.1 | 52.1 | 49.2 | 54.9 | 43.3 | 32.9 | 19.8 |

Table 15: Comparison of models on the M3Exam dataset across different languages.

| Models | English | Multi | Arabic | Bengali | Finnish | Indonesian | Korean | Russian | Swahili | Telugu |
|---|---|---|---|---|---|---|---|---|---|---|
| Vicuna-1.5-7B | 59.7 | 52.7 | 32.3 | 68.1 | 63.0 | 72.6 | 58.8 | 57.6 | 51.3 | 18.1 |
| Qwen2-7B-Instruct | 72.2 | 71.2 | 67.6 | 75.9 | 67.1 | 78.0 | 64.9 | 67.2 | 75.3 | 73.8 |
| Llava-1.5-7B | 66.8 | 52.8 | 61.8 | 33.4 | 60.2 | 72.8 | 63.3 | 55.0 | 55.0 | 20.6 |
| Llava-Next-7B | 68.3 | 52.1 | 64.5 | 24.9 | 63.0 | 74.3 | 61.9 | 58.4 | 53.1 | 17.0 |
| Phi-3.5-Vision | 75.9 | 51.3 | 63.1 | 24.8 | 57.3 | 70.6 | 60.2 | 57.5 | 48.7 | 28.3 |
| PALO-7B | 69.4 | 50.8 | 60.9 | 46.0 | 61.8 | 70.6 | 56.8 | 56.7 | 42.5 | 10.8 |
| PANGEA-7B (Ours) | 73.7 | 66.0 | 55.5 | 65.3 | 66.3 | 74.5 | 69.4 | 60.1 | 76.6 | 60.0 |

Table 16: Comparison of models on the TyDiQA dataset across different languages.

| Models | English | Multi | Arabic | Spanish | Basque | Hindi | Ind. | Burmese | Russian | Swahili | Telugu | Chinese |
|---|---|---|---|---|---|---|---|---|---|---|---|---|
| Vicuna-1.5-7B | 78.1 | 57.4 | 52.7 | 69.4 | 50.8 | 54.5 | 61.0 | 48.4 | 66.5 | 52.1 | 54.5 | 63.5 |
| Qwen2-7B-Instruct | 80.3 | 61.9 | 64.0 | 71.6 | 51.6 | 59.6 | 68.5 | 50.7 | 72.7 | 53.2 | 55.3 | 72.1 |
| Llava-1.5-7B | 79.1 | 57.6 | 52.7 | 69.2 | 50.9 | 54.9 | 62.6 | 49.0 | 65.9 | 51.7 | 55.8 | 63.9 |
| Llava-Next-7B | 79.1 | 57.1 | 51.7 | 68.8 | 50.3 | 54.5 | 62.0 | 46.7 | 65.5 | 52.1 | 55.2 | 63.8 |
| Phi-3.5-Vision | 77.9 | 54.8 | 53.7 | 67.2 | 50.4 | 54.9 | 51.7 | 47.8 | 61.3 | 49.3 | 52.5 | 59.5 |
| PALO-7B | 77.4 | 57.2 | 56.5 | 68.4 | 49.8 | 58.6 | 58.5 | 47.4 | 65.6 | 51.2 | 53.1 | 62.8 |
| PANGEA-7B (Ours) | 79.1 | 61.2 | 60.5 | 67.8 | 50.0 | 61.8 | 66.4 | 48.7 | 69.4 | 58.9 | 60.4 | 68.2 |

Table 17: Comparison of models on the XStoryCloze dataset across different languages.

| Models | English | Multi | Bengali | German | Spanish | French | Japanese | Russian | Swahili | Telugu | Thai | Chinese |
|---|---|---|---|---|---|---|---|---|---|---|---|---|
| Vicuna-1.5-7B | 17.6 | 6.4 | 0.0 | 14.4 | 9.6 | 14.4 | 2.8 | 10.8 | 3.6 | 0.0 | 2.0 | 14.8 |
| Qwen2-7B-Instruct | 48.8 | 40.4 | 0.0 | 67.2 | 67.6 | 68.8 | 11.2 | 71.2 | 10.8 | 2.4 | 45.6 | 59.2 |
| Llava-1.5-7B | 14.8 | 7.6 | 0.0 | 15.2 | 10.8 | 18.0 | 2.8 | 11.2 | 0.4 | 0.0 | 1.6 | 15.6 |
| Llava-Next-7B | 15.6 | 7.5 | 0.0 | 13.6 | 13.2 | 16.0 | 1.6 | 12.8 | 2.0 | 0.0 | 1.6 | 14.0 |
| Phi-3.5-Vision | 59.2 | 33.1 | 0.0 | 64.0 | 59.6 | 58.0 | 20.0 | 54.0 | 4.0 | 0.0 | 18.8 | 52.4 |
| PALO-7B | 13.6 | 5.8 | 0.0 | 11.6 | 9.6 | 13.2 | 1.6 | 8.8 | 0.4 | 0.0 | 0.0 | 12.4 |
| PANGEA-7B (Ours) | 82.0 | 47.4 | 0.0 | 68.4 | 74.8 | 63.2 | 22.0 | 68.0 | 54.0 | 5.6 | 49.6 | 68.0 |

Table 18: Comparison of models on the MGSM dataset across different languages.

| Models | English | Multi | Arabic | Bengali | Portuguese | Chinese | French | German |
|---|---|---|---|---|---|---|---|---|
| Vicuna-1.5-7B | 49.5 | 34.7 | 30.3 | 28.5 | 39.6 | 36.9 | 40.4 | 39.8 |
| Qwen2-7B-Instruct | 70.1 | 53.1 | 51.0 | 43.4 | 60.7 | 63.8 | 61.5 | 57.7 |
| Llava-1.5-7B | 50.2 | 34.9 | 29.7 | 28.5 | 40.3 | 36.8 | 40.1 | 39.8 |
| Llava-Next-7B | 52.1 | 35.6 | 30.0 | 28.8 | 40.7 | 37.3 | 41.4 | 41.4 |
| Phi-3.5-Vision | 62.0 | 39.1 | 34.9 | 27.9 | 47.6 | 41.5 | 49.2 | 45.8 |
| PALO-7B | 46.7 | 32.6 | 30.3 | 29.5 | 36.0 | 34.2 | 36.9 | 35.8 |
| PANGEA-7B (Ours) | 68.4 | 52.2 | 49.3 | 44.4 | 58.9 | 60.5 | 58.9 | 56.7 |
| **Models** | **Hindi** | **Indonesian** | **Italian** | **Japanese** | **Korean** | **Spanish** | **Swahili** | **Yoruba** |
| Vicuna-1.5-7B | 29.8 | 36.5 | 39.5 | 35.9 | 34.1 | 40.3 | 27.9 | 26.8 |
| Qwen2-7B-Instruct | 45.7 | 57.1 | 60.8 | 58.0 | 54.6 | 61.9 | 36.0 | 31.8 |
| Llava-1.5-7B | 29.2 | 37.1 | 41.0 | 35.1 | 34.1 | 41.6 | 28.0 | 27.3 |
| Llava-Next-7B | 29.6 | 37.5 | 41.2 | 36.0 | 34.2 | 42.7 | 28.5 | 28.7 |
| Phi-3.5-Vision | 32.9 | 38.3 | 47.0 | 40.0 | 36.6 | 49.6 | 28.9 | 27.8 |
| PALO-7B | 29.6 | 33.7 | 36.4 | 32.7 | 30.6 | 37.0 | 26.4 | 27.1 |
| PANGEA-7B (Ours) | 45.7 | 55.4 | 58.8 | 55.3 | 52.7 | 59.7 | 42.8 | 31.3 |

Table 19: Comparison of models on the MMMLU dataset across different languages.

