# OpenReview forum: "Pangea: A Fully Open Multilingual Multimodal LLM for 39 Languages"
_ICLR.cc/2025/Conference — ICLR 2025 Poster_

### Official Review · Reviewer_pV19 · 2024-10-28

**Soundness:** 4
**Presentation:** 3
**Contribution:** 4
**Rating:** 8
**Confidence:** 4

**Summary:**

The paper introduces PANGEA, a multilingual multimodal large language model (MLLM) addressing the limitations of English-centric data in existing models. The model was trained on a dataset called PANGEAINS, which includes 6 million instruction samples across 39 languages. PANGEINS combines high-quality English instructions, machine-translated content, and culturally relevant tasks. The authors also developed PANGEABENCH, an evaluation suite to assess the model’s multilingual and multimodal capabilities, including tasks like chat, image captioning, and visual question answering. PANGEA outperforms existing open-source models, especially in tasks requiring cross-cultural understanding, though it still lags behind proprietary models like GPT-4 in areas like complex reasoning. Fully open-sourced along with its dataset and evaluation suite, PANGEA aims to promote further research in inclusive multilingual LLMs, addressing challenges like data scarcity, cross-lingual transfer, and multilingual OCR.

**Strengths:**

1. This paper is well-written, offering comprehensive details on the curated instruction-tuning datasets, benchmarks, and evaluation results. The authors present their work with clarity and thoroughness.
2. The meticulously curated datasets for both training and evaluation are of significant practical value. They will undoubtedly make a substantial contribution to the multilingual research community, providing a robust foundation for future studies in this field.

**Weaknesses:**

1. The paper lacks some crucial details about the model and training process. Specifically, it would be beneficial to provide more information about the architecture, including the type of vision encoder used and the final total number of parameters. Additionally, details regarding the training infrastructure and duration are absent.
2. Some evaluation setup details are missing, such as the number of shots for each dataset, which makes the results difficult to reproduce. For instance, the results for Qwen2-7B-instruct are significantly worse than those reported in the official release (https://qwenlm.github.io/blog/qwen2/#multilingual-capability-of-instruction-tuned-models). What is the reason?
3. Although the authors claim the model supports 39 languages, it's unclear which specific languages have shown improvement and to what extent compared to the backbone model. This is particularly uncertain for languages that comprise only a small portion of the PANGEAINS dataset.

**Questions:**

In Figure 7, the model's poor performance on English OCR is counterintuitive. What might explain this unexpected result?

---

> ### Author Response · Authors · 2024-11-23
>
> Thanks to the reviewer for recognizing the **significant contributions** of our work, including the **PANGEA model**, the **PANGEAINS dataset**, and the **PANGEABENCH evaluation suite**. We are glad the reviewer appreciates the **comprehensiveness** of our datasets, benchmarks, and evaluation results, as well as the **practical value** they offer to the **multilingual research community**.
>
> > The paper lacks some crucial details about the model and training process.
>
> Thank you for pointing this out! We added **more details on the training specifics** in the revised version in **Section 4.1**. Specifically:
> - We used the **LLaVA-NeXT framework**, with **Qwen2-7B-Instruct** as the language backbone and **clip-vit-large-patch14-336** as the vision encoder.
> - The training consists of **two stages**:
>   1. **Pretraining the connector** on the LLaVA pretraining dataset, which took around **4 hours with 8 × H100 GPUs (32 GPU hours)**.
>   2. **Finetuning** for **1 epoch on PANGEAINS**, which took approximately **168 hours with 8 × H100 GPUs (1344 GPU hours)**.
>
> > Some evaluation setup details are missing, such as the number of shots for each dataset. For instance, the results for Qwen2-7B-Instruct are significantly worse than those reported in the official release ([https://qwenlm.github.io/blog/qwen2/#multilingual-capability-of-instruction-tuned-models](https://qwenlm.github.io/blog/qwen2/#multilingual-capability-of-instruction-tuned-models)). What is the reason?
>
> Thank you for pointing this out! We added **more details of the evaluation setup** in **Section 3.1** in the revised version. For all the evaluation, we used a **zero-shot setting** for all evaluation benchmarks.
>
> In the **Qwen official release**, results were demonstrated for **MMMLU** and **MGSM**.
> **Key differences:**
> - For **MMMLU**, our reported performance is **higher** because we used the **OpenAI version of MMMLU** ([link](https://huggingface.co/datasets/openai/MMMLU)), which was **not available** during the Qwen official release.
> - For **MGSM**, we were unable to reproduce their results under the **zero-shot setting**. However, under the **5-shot setting**, our performance closely aligns with theirs:
>
> | Model              | MGSM EN (0-shot) | MGSM Multi (0-shot) | MGSM EN (5-shot) | MGSM Multi (5-shot) |
> |--------------------|------------------|---------------------|------------------|---------------------|
> | **Pangea-7B**      | **82.0**         | **47.3**            | **81.6**         | **54.9**            |
> | **Qwen2-7B-Instruct** | 48.8             | 40.4                | 79.2             | 50.7                |
>
> We evaluated Qwen2-7B-Instruct's performance under the **5-shot setting**:
> - **English:** 79.2
> - **Multilingual:** 50.7
> - **Overall:** 53.3 (close to their reported overall performance of **57.0**).
>
> > It's unclear which specific languages have shown improvement and to what extent compared to the backbone model.
>
> In the **text-only benchmarks**, we evaluated the performance of both **Qwen2-7B-Instruct** (the backbone model) and **Pangea**. In the appendix, we included a **language-wise breakdown of results** in **Tables 16, 17, and 18**. Specifically, **Pangea** shows **improvements** over **Qwen2-7B-Instruct** in the following languages:
> - **Korean**
> - **Swahili**
> - **Hindi**
> - **German**
> - **Japanese**
> - **Thai**
>
> > In Figure 7, what might explain the model's unexpected poor performance on English OCR?
>
> It is important to note that this is a **preliminary exploration** of constructing **OCR instruction tuning data**, which aims to provide **insights for future research** on OCR training. For testing the model's OCR capabilities, we reserved a **small portion of the OCR instruction tuning dataset** curated from **multilingual web screenshots**. Upon **manual inspection**, we found that the **English training data** for web OCR tasks had **more complex structures and contexts** compared to data in other languages. This **increased complexity** in English OCR data likely explains the model's **unexpected poor performance** on English OCR tasks.

---

> > ### Comment · Reviewer_pV19 · 2024-11-26
> >
> > Thank you for the clarification and additional experimental details. I would like to increase the score.

---

> > > ### Author Response · Authors · 2024-11-26
> > >
> > > Thank you for the positive feedback! We are happy that our response addressed your concerns! Your invaluable feedback was really helpful in boosting the quality of this work!
> > >
> > > Thanks!

---

### Official Review · Reviewer_GEq8 · 2024-11-04

**Soundness:** 2
**Presentation:** 2
**Contribution:** 3
**Rating:** 5
**Confidence:** 3

**Summary:**

PANGEAINS is a multilingual and multicultural dataset comprising 6 million samples across 39 languages, developed for instruction tuning. This dataset emphasizes linguistic and cultural diversity and implements strategies such as machine translation, cultural understanding guidelines, and high-quality image caption generation to address challenges in multilingual multimodal learning. By using machine translation, English instructions are extended into other languages, and culturally biased issues are resolved by curating images from the LAION-Multi dataset to capture cultural contexts.

PANGEABENCH is a comprehensive evaluation tool designed to assess PANGEA's performance across various languages and tasks. It includes diverse benchmarks that evaluate capabilities in cross-lingual and cross-cultural contexts. The multimodal tasks encompass natural conversation, image captioning, cultural understanding, multilingual visual question answering, and multi-subject reasoning, while text-only tasks are also included to examine the model's linguistic understanding.

Experimental results show that PANGEA-7B outperforms existing models in both English and multilingual tasks, particularly demonstrating significant improvements in cultural understanding tasks. The performance gap between English and multilingual tasks is small, indicating strong multilingual processing capabilities, although there remains a performance disparity compared to closed-source models. Additionally, PANGEA-7B exhibits the strongest performance in text-only tasks, with notable enhancements in handling complex multilingual reasoning and mathematical tasks attributed to the inclusion of math-related instructions.

The discussion explores the scaling effect of instruction quantity, the role of English data, the relationship between training sample proportions and performance, real-world applications of multimodal chat, and preliminary explorations of multilingual OCR. These insights aim to provide a comprehensive understanding of the model and outline future advancement directions. The development of PANGEAINS and PANGEABENCH is expected to significantly contribute to enhancing the performance of multilingual and multimodal models.

**Strengths:**

Originality
The paper presents approaches to multilingual and multimodal learning through the development of the PANGEAINS dataset and the PANGEABENCH evaluation framework. By focusing on both linguistic and cultural diversity, it addresses gaps in existing datasets that often overlook these critical aspects. The integration of machine translation and culturally relevant guidelines for data curation showcases an innovative methodology that enhances the relevance and applicability of the dataset. This originality is further emphasized by the emphasis on cultural understanding in multimodal tasks, which is a relatively underexplored area in the field.

Quality
The quality of the research is evident in the rigorous methodology employed for dataset creation and model evaluation. The paper provides a thorough explanation of the processes involved in curating the PANGEAINS dataset, including the use of high-quality image captioning and the implementation of robust evaluation metrics. The empirical results presented are well-supported, demonstrating PANGEA's superior performance in various tasks. Additionally, the use of established benchmarks and the careful selection of evaluation metrics contribute to the overall robustness of the findings.

Clarity
The paper is well-structured and clearly organized, making it accessible to readers across different backgrounds. Each section is logically connected, and the authors effectively communicate complex ideas through concise language and well-defined terminology. The inclusion of visual aids, such as figures and tables, enhances understanding by illustrating key concepts and results. Overall, the clarity of presentation allows for easy comprehension of the methodologies and findings.

Significance
The significance of the paper lies in its potential impact on the field of multilingual and multimodal learning. By providing a comprehensive dataset and a structured evaluation framework, the authors contribute valuable resources that can facilitate further research and development in this area. The focus on cultural diversity and understanding not only enriches the training of models but also promotes more inclusive and applicable AI systems. The findings underscore the importance of addressing cultural nuances in AI applications, which is essential for creating systems that are effective and relevant in a global context.

**Weaknesses:**

Limitations of the Dataset
The PANGEAINS dataset comprises samples from 39 languages; however, there exists a potential imbalance in both the quantity and quality of data for specific languages. This issue is particularly pronounced for low-resource languages that the authors do not explicitly address. The scarcity of data for these languages can lead to significant challenges in training models that are capable of performing effectively across diverse linguistic contexts. Insufficient representation may result in suboptimal performance, as the models may lack the necessary exposure to the varied linguistic structures and cultural nuances inherent in these languages.

Moreover, the quality of the data is as crucial as its quantity. If the available data for certain languages is of lower quality—whether due to inaccuracies in translation, lack of contextual relevance, or inadequate cultural representation—this can further exacerbate performance issues. Such disparities in data quality can hinder the model's ability to generalize effectively across languages, potentially leading to biased outputs and diminished reliability in multilingual applications.

Consequently, these imbalances may impede the overall generalization capabilities of multilingual processing, as the model’s performance may be skewed towards languages with richer datasets. Thus, it is imperative for future iterations of the dataset to ensure a more equitable distribution of data quality and quantity across all included languages, particularly for those that are underrepresented, in order to enhance the robustness and applicability of multilingual models.

-Depth of Cultural Context
In evaluating cultural understanding, the discussion regarding whether the PANGEAINS dataset adequately reflects diverse cultural backgrounds is somewhat lacking. Cultural differences are complex and nuanced, extending beyond mere linguistic expressions or visual representations to encompass social contexts, customs, values, and other interrelated elements. These factors shape the meanings within specific cultural contexts and are essential for multilingual multimodal models to accurately understand and process cultural nuances. However, the current dataset primarily focuses on the matching of images and captions, which may not sufficiently capture the depth of cultural context.

For instance, metaphorical expressions or symbolic meanings in one culture may be interpreted entirely differently in another. Capturing these subtle distinctions necessitates a more in-depth analysis that transcends simple dataset curation. It would be beneficial to incorporate expert consultations on cultural backgrounds and conduct comprehensive research on each culture to inform data collection efforts. Such an approach would enhance the model's ability to understand meanings across various cultural contexts and generate more nuanced responses.

- Limitations of Experimental Design
From the perspective of experimental design, there is a notable lack of in-depth analysis concerning specific tasks or languages. For example, in tasks such as multilingual visual question answering (VQA) or multi-subject reasoning, a detailed exploration of the factors influencing model performance is missing. Without such analysis, it becomes challenging to clearly understand the model's strengths and weaknesses, ultimately limiting the ability to establish future improvement directions.

Particularly, without a thorough examination of various determinants of performance—such as the characteristics of specific languages, dataset imbalances, or hyperparameter adjustments during training—understanding why a model excels or falters in certain tasks becomes difficult. Therefore, it is crucial that the experimental design explicitly defines the specific conditions and variables for each task, along with comparative analyses of performance across different environments. This approach would not only enhance model performance but also deepen the overall understanding of multilingual and multimodal systems.

**Questions:**

Data Representation: What strategies can be implemented to ensure a more balanced representation of low-resource languages within the PANGEAINS dataset?

Quality Assessment: How can we establish rigorous quality assessment protocols to evaluate the accuracy and relevance of data for each language in the dataset?

Cultural Context: What methodologies can be employed to enrich the dataset with culturally relevant examples and ensure that cultural nuances are adequately captured across all languages?

Data Augmentation: What role can data augmentation techniques play in enhancing the dataset, particularly for underrepresented languages, to improve model performance?


As a reviewer, I would like to provide detailed technical advice regarding the concerns particularly focusing on the potential imbalances in data quantity and quality across different languages, especially for low-resource languages. Perhaps I (not a robot) missed the methods described in your manuscript. Please let me know and provide additional information if possible.

1. Addressing Imbalances in Quantity and Quality
- Data Augmentation Techniques:

Synonym Replacement: Identify and replace words with their synonyms to generate paraphrased sentences that maintain the original meaning.
Back Translation: Translate sentences into another language and then back to the original language to create varied sentence structures while preserving semantic content.
Noise Injection: Introduce small amounts of noise (e.g., typos, grammatical variations) to simulate real-world language use and increase dataset diversity.

- Collaboration with Native Speakers:

Engage with native speakers or linguists proficient in low-resource languages to enhance the quality of your dataset. This could involve Crowdsourcing Platforms with native speakers for data validation and enrichment.

- Detailed Data Distribution Analysis:

Provide a comprehensive analysis of the data distribution across the 39 languages included in the dataset. This should include:
Quality Metrics: If applicable, include metrics that assess the quality of the data per language (e.g., linguistic diversity, error rates).

Visual Representations: Use graphs or charts to visually represent the distribution and highlight any significant imbalances.

2. Enhancing Quality of Cultural Representation

When collecting data, consider the cultural context of each language. This can be achieved by:
Ethnographic Research: Integrate ethnographic methods to understand cultural nuances that might not be reflected in traditional data collection.

Focus Groups: Conduct focus groups with speakers of the languages to gather insights on cultural expressions and values that should be represented in the dataset.

Examples of Cultural Representation: Provide specific examples of how the dataset captures cultural differences. Detail how various cultural elements (e.g., idioms, customs, social norms) are represented in the data. Case studies that illustrate the inclusion of culturally relevant examples. An analysis of how the dataset addresses or fails to address complex cultural concepts.



3. insufficient in-depth analysis related to specific tasks and languages, particularly in areas such as multilingual visual question answering (VQA) and multi-subject reasoning.

Below are some actionable suggestions to enhance your analysis:

- Conducting Ablation Studies
Ablation Studies on Language Groups: Perform ablation studies to assess how different language groups affect model performance. This involves systematically removing specific languages or language features from the dataset to observe changes in performance metrics. Analyze which languages contribute the most to overall model effectiveness and identify any languages that may cause degradation in performance.

Ablation Studies on Task Types: Similarly, conduct ablation studies focusing on different task types (e.g., VQA versus multi-subject reasoning). By isolating the performance impacts of each task type, you can gain insights into which tasks are more challenging for your model and why.

Expert Collaboration: How can collaboration with linguistic and cultural experts be structured to inform the data collection process, ensuring a more comprehensive understanding of low-resource languages?

Evaluation Metrics: What specific evaluation metrics can be developed to assess the model's performance across different languages, particularly focusing on those with limited data?

Feedback Mechanisms: How can feedback mechanisms be incorporated into the dataset's development process to continuously improve data quality and representation based on model performance?

Cross-Linguistic Comparisons: What methods can be utilized to conduct cross-linguistic comparisons that identify specific challenges faced by the model when processing low-resource languages?

---

> ### Author Response · Authors · 2024-11-23
>
> We appreciate the reviewer’s feedback! We would like to address the following concerns:
>
> > There exists a potential imbalance in both the quantity and quality of data for specific languages. What strategies can be implemented to ensure a more balanced representation of low-resource languages within the PANGEAINS dataset?
>
> We acknowledge that there might be an **imbalance in the quality and quantity of data** for lower-resource languages. When curating **PangeaIns**, we aimed for **balanced sampling**, and we believe that **abilities gained from training on high-resource languages can be transferred** to lower-resource languages. For example, we achieved **similar or higher accuracy** on two low-resource languages, **Swahili and Tamil**, on **MaRVL**, as shown in **Table 9** in the paper.
>
> > How can we establish rigorous quality assessment protocols to evaluate the accuracy and relevance of data for each language in the dataset?
>
> To ensure the **high quality** of the training data, we performed:
> 1. **Careful selection** of existing datasets and **comprehensive filtering** to ensure only high-quality data is included.
> 2. **Manual quality checks** for the images and texts of each existing corpus before incorporation into PangeaIns.
> 3. Input from **multilingual experts** to inspect the quality of multilingual data.
>
> For translations:
> - We examined various open-source translation models, such as **NLLB-3B**, but found through manual inspections that they struggled with **complex instruction-following** and **context-switching** tasks, especially in **specialized domains** like code generation and mathematical reasoning.
> - Therefore, we opted for the **proprietary Gemini 1.5 Pro model**, which demonstrated **slightly better performance** in small-scale human evaluations compared to **GPT4o**.
>
> For culturally relevant data:
> - We manually inspected a **small subset of outputs** in all 39 languages to ensure high quality.
> - Once satisfied with the general quality, we curated the **culturally relevant subset** of PangeaIns and **filtered out lower-quality data** using a **large language model (Llama-3.1-8B-Instruct)**.
>
> > In evaluating cultural understanding, the discussion regarding whether the PANGEAINS dataset adequately reflects diverse cultural backgrounds is somewhat lacking. What methodologies can be employed to enrich the dataset with culturally relevant examples and ensure that cultural nuances are adequately captured across all languages?
>
> In **PangeaBench**, we evaluated **Pangea** on two benchmarks:
> 1. **CVQA**
> 2. **MaRVL**
>
> These benchmarks test model performance in **reasoning tasks involving culturally relevant imagery and concepts** across multiple languages. We believe that **high scores on these benchmarks** reflect:
> - **Good cultural understanding capabilities** of Pangea.
> - The effectiveness of **PangeaIns** in providing **diverse and culturally rich training data**, which enables the model to perform well in cultural understanding tasks.
>
> > From the perspective of experimental design, there is a notable lack of in-depth analysis concerning specific tasks or languages. For example, in tasks such as multilingual visual question answering (VQA) or multi-subject reasoning, a detailed exploration of the factors influencing model performance is missing.
>
> In **Section 5** and **Figure 6**, we performed an **in-depth analysis** on the impact of the **proportion of training samples in a language** on downstream performance. Additionally, we explored **factors influencing model performance** on **multimodal chat** in **Section 5** and **Appendix E**.
>
> > What role can data augmentation techniques play in enhancing the dataset, particularly for underrepresented languages, to improve model performance?
>
> We utilized **data augmentation techniques** to enhance data in **lower-resource languages** such as **Telugu** and **Swahili**. Data augmentation is particularly valuable in cases where data for specific languages is limited. For instance:
> - We used **machine translation** and **culturally relevant curation** to address resource insufficiencies.
> - These efforts led to **improved performance** in underrepresented languages. **Table 9** demonstrates an example of such improvements.

---

> > ### Author Response · Authors · 2024-11-26
> >
> > Dear Reviewer ```GEq8```,
> >
> > As today is the last day to update the paper PDF, we would like to learn if our response addresses your concerns and questions, and we invite any additional feedback or thoughts for improving our paper. If you feel that our responses resolve the issues raised, we would be grateful if you could consider reflecting this in the evaluation. We would be happy to address any further concerns or questions.

---

### Official Review · Reviewer_rNZk · 2024-11-04

**Soundness:** 4
**Presentation:** 3
**Contribution:** 4
**Rating:** 8
**Confidence:** 4

**Summary:**

This paper introduces Pangea, a multilingual multi-modal LLM together with its multi-modal instruction fine-tuning dataset PangeaIns and a comprehensive benchmark PangeaBench:
- PangeaIns includes 6M examples from 39 languages and consists of (1) high-quality English instructions, (2) machine-translated instructions, (3) culturally relevant multimodal tasks
- PangeaBench, a multimodal evaluation suite consists of 14 diverse datasets covering 47 languages, including a human-curated multimodal chat benchmark xChat

Pangea outperforms other open-source models by 7.3 points on English tasks and 18.0 points in multilingual tasks. Finally, the paper discusses the impact of English data, the necessity of language-specific datasets, and the limitations in multimodal chat and multilingual OCR.

**Strengths:**

A. This paper includes a very significant set of contributions which will be made public together with the code:
1. PangeaIns with 6M instructions from 39 languages that include (1) high-quality English instructions, (2) machine-translated instructions, (3) culturally relevant multimodal tasks
2. PangeaBench, a multimodal evaluation suite consists of 14 diverse datasets covering 47 languages
3. Pangea an MLLM covering 39 languages outperforms existing open-source models by 7.3 points on English tasks and 18.0 points in multilingual tasks.

B. The paper presents a detailed multilingual multi-modal data generation pipeline showcasing the limitation of open-source machine translation models, and the use of LLMs to enrich the multilingual data where the LLM score and filter images based on cultural informativeness. Then remaining data is enriched with re-generating descriptions and creating complex instructions.

C. PangeaBench covering 47 languages in 14 datasets introduces a comprehensive evaluation for multilingual multimodal LLMs. Notably, the new evaluation dataset xChat is a human-curated benchmark where reference answers and scoring rubrics are annotated for each query. This enables a better use of LLM-as-a-judge for scoring other models' generations using a scale of 1-5 for each query.

**Weaknesses:**

There are only two minor weaknesses mainly focusing on the need for clarity and the presentation:

A. In Section 2.1, the paper mentions a post-processing pipeline for noisy translations, however, details are missing. This might be quite important for practitioners to build similar models. Therefore, I suggest authors to include more details there.

B. In Section 4.1, it has been mentioned that the model is based on Llava-Next architecture but this needs to be detailed. For example, does the training include multiple stages such as adapter alignment and SFT, or is only SFT has been used? If so what datasets are used in which stage?

**Questions:**

A. For text-only performance, how is the chat performance compared to the other models? Some evaluation benchmarks such as MTBench, ArenaHardAuto or AlphacaEval can be used for this comparison.

B. Have you checked if there is any data leakage from the benchmarks that has been tested?

---

> ### Author Response · Authors · 2024-11-23
>
> Thanks to the reviewer for highlighting the **significant contributions** of our work, including the **PangeaIns dataset**, **PangeaBench evaluation suite**, and the **Pangea model**, which together provide robust multilingual multimodal resources and achieve **superior performance across diverse tasks**!
>
> > In Section 2.1, the paper mentions a post-processing pipeline for noisy translations; however, details are missing.
>
> Thank you for pointing this out! We will release all the code for curating **PangeaIns**, including that for post-processing translations. For example, in the post-processing step, we checked whether the **number of turns in conversations remains the same** post-translation, and we verified whether the **ratio between the length of pre- and post-translation** is reasonable.
>
> > In Section 4.1, it has been mentioned that the model is based on LLaVA-NeXT architecture, but this needs to be detailed.
>
> Thank you for pointing this out! We added **more details on the training specifics** in the revised version in **Section 4.1**. Specifically:
> - We used the **LLaVA-NeXT framework**, with **Qwen2-7B-Instruct** as the language backbone and **clip-vit-large-patch14-336** as the vision encoder.
> - The training consists of **two stages**:
>   1. **Pretraining the connector** on the LLaVA pretraining dataset, which took around **4 hours with 8 × H100 GPUs (32 GPU hours)**.
>   2. **Finetuning** for **1 epoch on PANGEAINS**, which took approximately **168 hours with 8 × H100 GPUs (1344 GPU hours)**.
>
> > For text-only performance, how is the chat performance compared to other models? Some evaluation benchmarks such as MTBench, ArenaHardAuto, or AlpacaEval can be used for this comparison.
>
> In **PangeaBench**, we didn’t include text-only chat evaluation benchmarks. Instead, we included **two multilingual multimodal chat benchmarks**:
> - **Multilingual LLaVA Bench**
> - **XChat**
>
> We believe these benchmarks provide good indicators of Pangea’s chat capabilities. Additionally, we evaluated **Pangea** and **LLaVA-NeXT** on **MT-Bench**. The results are as follows:
>
> | Language | **Pangea** | **LLaVA-NeXT-7B** |
> |----------|------------|-------------------|
> | en       | **7.59**   | 5.78             |
> | fr       | **6.84**   | 4.77             |
> | ja       | **6.38**   | 3.91             |
>
> > Have you checked if there is any data leakage from the benchmarks that have been tested?
>
> Yes! We ensured that **no data from PangeaBench** was included in PangeaIns. However, it's worth noting that **PangeaIns** does include the **training splits** of **NLVR2** and **GQA-ru** datasets. In PangeaBench:
> - We evaluated model performance on the **evaluation splits** of these datasets:
>   - **NLVR2's evaluation split** was used for the English performance of the multilingual-only dataset MaRVL.
>   - **GQA-ru's evaluation split** was used for GQA-ru.
>
> It is important to note that this is a **common practice** in instruction tuning, as seen in [1, 2].
>
> [1] Haotian Liu, Chunyuan Li, Qingyang Wu, and Yong Jae Lee. Visual instruction tuning. In NeurIPS, 2023. [URL](https://arxiv.org/abs/2304.08485)
> [2] Shengbang Tong, Ellis Brown, Penghao Wu, Sanghyun Woo, Manoj Middepogu, Sai Charitha Akula, Jihan Yang, Shusheng Yang, Adithya Iyer, Xichen Pan, et al. Cambrian-1: A fully open, vision-centric exploration of multimodal LLMs. ArXiv preprint, abs/2406.16860, 2024. [URL](https://arxiv.org/abs/2406.16860)

---

> > ### Comment · Reviewer_rNZk · 2024-11-25
> >
> > Thanks authors for their response including the required details and additional results. I will keep my positive score.

---

> > > ### Author Response · Authors · 2024-11-26
> > >
> > > Dear Reviewer ```rNZk```,
> > >
> > > We would like to thank you again for the positive feedback. Your invaluable comments really helped us further boost the quality of the paper!
> > >
> > > Thanks!

---

### Official Review · Reviewer_e2zk · 2024-11-07

**Soundness:** 3
**Presentation:** 3
**Contribution:** 3
**Rating:** 6
**Confidence:** 4

**Summary:**

This paper describes PANGEA model and data, with a focus on multilingual multimodal settings. 	PANGEAINS is an instruction dataset that spans 39 languages that covers the different languages through translation and the culture references aspects of the data. PANGEABENCH is an evaluation suite consists of 14 datasets covering 47 languages. PANGEA, a model based on the LLaVA-Next architecture and trained on the PANGEAINS data, showed strong performance on several tasks comparing to strong baseline models.

**Strengths:**

1. This paper addressed an important research problem on multilingual multimodal learning, by providing a large instruction dataset and evaluation suite that covers multiple languages with the corresponding culture reference annotations. This data could be of interest to a broad audience in the research community.
2. The paper is well written and easy to follow. The proposed PANGEAINS instruction data and the PANGEABENCH evaluation suite were described with good amount of details. The authors further provided examples of the data which is helpful in understanding the dataset proposed.
3. The authors conducted comprehensive experiments of the proposed model and data, by comparing it to several strong baseline models and evaluating on a diverse set of tasks. The performance of the model seems to be pretty strong from the evaluation results.

**Weaknesses:**

1. It could be helpful to a section to discuss related multilingual multimodal datasets (e.g. M3IT etc), and show how PANGEA compares to them. This could further illustrate the significance of the proposed data here.
2. Although the authors highlighted the introduction of PANGEA, the discussion on the modeling aspect of PANGEA is light. It seems to be a directly application of a LLaVA-Next based model trained using the new data proposed. It could help to further discuss the training specifics.
3. The proposed method (model and data) is only evaluated on one eval set, which is the proposed PANGEABENCH. It could be helpful to evaluate on additional evaluation suits to see how the improvement from training on PANGEAINS would generalize.

**Questions:**

1. Would the other open source models used as the baseline in this paper could also benefit from training on the PANGEAINS data?
2. How to validate if there is potential overfitting of training on the PANGEAINS and evaluating on PANGEABENCH?

---

> ### Author Response · Authors · 2024-11-23
>
> Thanks to the reviewer for the positive feedback on our paper addressing **multilingual multimodal learning**, highlighting the well-detailed **PANGEAINS dataset**, **PANGEABENCH evaluation suite**, and the **comprehensive experiments** demonstrating strong model performance.
>
> > It could be helpful to add a section to discuss related multilingual multimodal datasets (e.g., M3IT, etc.), and show how PANGEA compares to them.
>
> In the revised version, we added **a table comparing other multimodal instruction tuning datasets to PANGEAINS** in **Table 4** and included more discussion of other multilingual multimodal datasets in the Appendix A Related Works section.
> | Dataset          | # Languages | # of Instances | Multicultural | # of Task Types | Open-Sourced |
> |------------------|-------------|----------------|---------------|-----------------|--------------|
> | MultiInstruct    | 1           | ~235.0K        | ✗             | 310             | ✗            |
> | MiniGPT4         | 1           | 5.0K           | ✗             | 149             | ✓            |
> | LLAVA            | 1           | 1.2M           | ✗             | >100K           | ✓            |
> | InstructBLIP     | 1           | ~1.6M          | ✗             | >100K           | ✗            |
> | M³IT             | 80          | 2.4M           | ✗             | 400             | ✓            |
> | mBLIP            | 95          | 5.1M           | ✗             | 68              | ✓            |
> | PALO             | 10          | 2.1M           | ✗             | 22              | ✓            |
> | Cambrian         | 1           | 7.1M           | ✗             | >1M             | ✓            |
> | PANGEAINS (Ours) | 39          | 6.2M           | ✓             | >1M             | ✓            |
>
> > Although the authors highlighted the introduction of PANGEA, the discussion on the modeling aspect of PANGEA is light.
>
> Thank you for pointing this out! We added **more details on the training specifics** in the revised version in **Section 4.1**. Specifically:
> - We used the **LLaVA-NeXT framework**, with **Qwen2-7B-Instruct** as the language backbone and **clip-vit-large-patch14-336** as the vision encoder.
> - The training consists of **two stages**:
>   1. **Pretraining the connector** on the LLaVA pretraining dataset, which took around **4 hours with 8 × H100 GPUs (32 GPU hours)**.
>   2. **Finetuning** for **1 epoch on PANGEAINS**, which took approximately **168 hours with 8 × H100 GPUs (1344 GPU hours)**.
>
> > The proposed method (model and data) is only evaluated on one eval set, which is the proposed PANGEABENCH. It could be helpful to evaluate on additional evaluation suites.
>
> **PANGEABENCH is not a single eval set.** Instead, it consists of **14 unique evaluation datasets**, covering a wide range of languages and tasks, including:
> - **Multimodal chat**
> - **Image captioning**
> - **Cultural understanding**
> - **Multilingual visual question answering**
> - **Multi-subject reasoning**
> - **Text-only evaluation**
>
> Among the evaluation datasets in PANGEABENCH, **12 out of 14 datasets are out-of-domain.** Thus, we believe **PANGEABENCH provides a comprehensive evaluation** of PANGEA's capabilities.
>
> > Would the other open-source models used as the baseline in this paper also benefit from training on the PANGEAINS data?
>
> Yes! We believe that training on **PANGEAINS**, a **high-quality instruction tuning multimodal dataset spanning 39 languages**, could improve the performance of other models. We chose **LLaVA-NeXT** as our model backbone due to its **simplicity**, **effectiveness**, and **widespread adoption**.
>
> > How to validate if there is potential overfitting of training on PANGEAINS and evaluating on PANGEABENCH?
>
> It's important to note that **PANGEAINS was curated with a primary focus on providing high-quality multilingual and multimodal instructions**, rather than being tailored specifically for performance on **PANGEABENCH**.
> In fact:
> - Only **2.2% of the samples in PANGEAINS are in-domain training examples** related to PANGEABENCH.
> - We also conducted a **data contamination check** before training the model.
> The strong performance observed on **PANGEABENCH reflects the robust capabilities developed** through training on the diverse and comprehensive data in PANGEAINS, rather than overfitting to the benchmark dataset.

---

> > ### Author Response · Authors · 2024-11-26
> >
> > Dear Reviewer ```e2zk```,
> >
> > As today is the last day to update the paper PDF, we would like to learn if our response addresses your concerns and questions, and we invite any additional feedback or thoughts for improving our paper. If you feel that our responses resolve the issues raised, we would be grateful if you could consider reflecting this in the evaluation. We would be happy to address any further concerns or questions.
> >
> > Thank you again for your time and effort!

---

### Author Response · Authors · 2024-11-25
**Friendly Reminder**

Dear reviewers, this is a friendly reminder that the discussion period concludes on November 26, and thus we would be grateful for your feedback at your earliest convenience. Your insights are invaluable to us. If you require any additional information or have questions, please do not hesitate to let us know.
Thank you very much for your time and consideration!

---

### Meta-Review · Area_Chair_VD9E · 2024-12-20

**Metareview:**

This paper introduces PANGEA, a multilingual multimodal LLM, and two resources: PANGEAINS (a 6M instruction dataset covering 39 languages) and PANGEABENCH (an evaluation suite spanning 47 languages). The authors claim significant improvements over existing open-source models on both English and multilingual tasks.

Strengths: This work emphasizes the importance of culturally aware training data and evaluation, with rigorous evaluation across multiple languages and tasks.

Weaknesses: Some architectural and training details were initially missing (though addressed in rebuttal), these should be added to the final version of the paper. Some reviewers also raise concerns about the potential data imbalances and limited discussion/handling of low-resource languages.

Despite some limitations, the work makes substantial contributions with multilingual and multimodal resources. I recommend accepting this paper.

**Additional Comments On Reviewer Discussion:**

Despite that Reviewer GEq8's comment seems to be auto-generated, the remaining comments from other reviewers are detailed and instructive. The authors provided more details in terms of architecture and training during the discussion period, which helped clarify several technical questions raised during the initial review.

---

### Decision · Program_Chairs · 2025-01-22

Accept (Poster)